# TELESCOPE: IMPROVING ZERO SHOT DETECTION OF LLM GENERATED CONTENT WITH TOKEN REPETITION

## ABSTRACT

Distinguishing Large Language Model (LLM) generated text from human writing is a critical and difficult challenge. While LLMs are trained to write like humans, we hypothesize that this training leaves an indelible mark. LLMs develop a particularly strong aversion to token repetition very early in training. This bias persists as a "Vestigial Heuristic" (a developmental artifact) that is activated in LLM-generated text, separating LLM from human writing. To probe this phenomenon, we introduce Telescope Perplexity, a metric that evaluates the token repetition of the model, $P(s_i|s_{1:i})$. Our empirical investigation reveals that the Telescope Perplexity signature emerges early in pre-training, and Telescope Perplexity empirically enables highly effective zero-shot LLM detection. We show state-of-the-art or competitive performance across diverse datasets (including modern evaluation sets we introduce), reference models, and perturbation schemes with greater efficiency than other methods.

## 1 INTRODUCTION

### 1.1 BACKGROUND AND MOTIVATIONS

Large Language Models (LLMs) captured the attention of the general public in 2022 when OpenAI released ChatGPT. This step forward showed the world how well deep learning models could learn to understand and respond to human text (Ouyang et al. (2022)). In the time since ChatGPT released, the industry around Artificial Intelligence has been thrown into the cultural zeitgeist, while the capabilities of large language models have improved significantly. Language models have continued to become smarter (Hoffmann et al. (2022)), faster (Dao et al. (2022)), and cheaper (Cai et al. (2024)). They are now being used constantly for tasks from real time translation to helping explain difficult topics and problems for students. However, LLM's have also been used for deploying spam bots (Liyanage et al. (2024)), disseminating fake news online (Sallami et al. (2024)), and writing essays for students (Jelson et al. (2025)). Humans simply cannot distinguish between LLM and human written text (Radivojevic et al. (2024)), so algorithmic methods are needed to help control missuses of LLMs, and ideally these methods will continue to be effective and keep pace with new model releases. For this reason, we focus on **Zero-Shot Detection**. Zero-shot methods perform without the need to continuously update their training data, making them ideal for handling the constant churn of model releases.

### 1.2 INTRODUCING TELESCOPE PERPLEXITY AND THE VESTIGIAL HEURISTIC

It is well understood that neural networks often learn simpler, local patterns before mastering long-range dependencies (Choshen et al. (2021); Belrose et al. (2024)). This suggests a compelling question:

> *Do optimization pressures during early LLM pre-training instill*
> *fundamental statistical biases that are never unlearned later in training?*

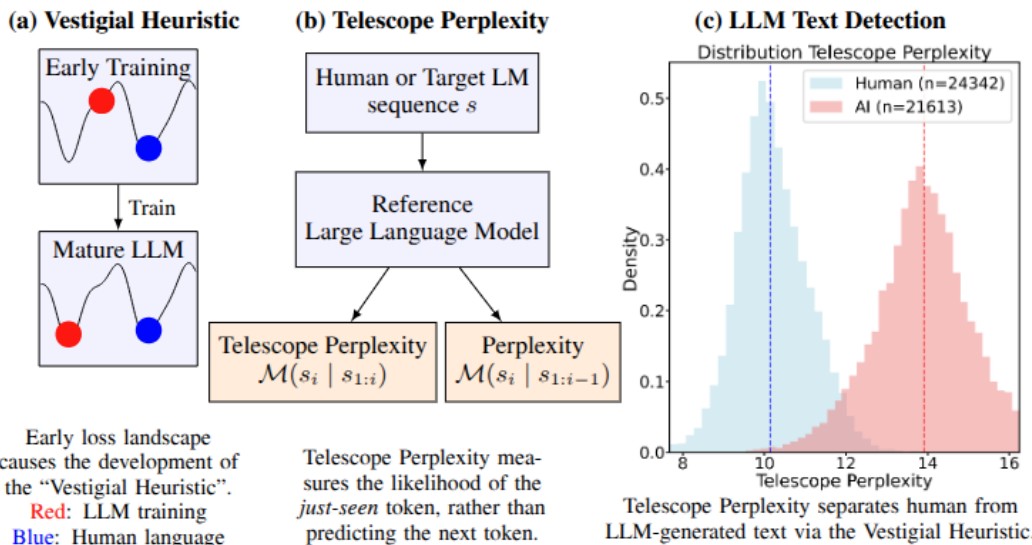

Figure 1: An overview of the "Vestigial Heuristic" hypothesis and the Telescope probe.

We know from previous work by Belrose et al. (2024) that most language models start out in early training as bigram models, directly predicting the next token from the one or maybe two tokens that precede it. We hypothesize that early in training, while they are developing these bigram models, LLMs learn an important heuristic that aids them in modeling text as a bigram model: tokens very rarely repeat themselves. Therefore, these early stage bigram models will consistently have a uniquely strong aversion to token repetition as they learn that tokens very rarely repeat themselves in natural language.

A couple key results from previous work show that what these bigram models learn persists throughout later stages of training, and that most LLMs will develop the same bigram model as one another even though they are trained through different datasets and architectures (Choshen et al. (2021), Belrose et al. (2024)). Since this heuristic persists over time, even though it is often not entirely necessary to vehemently reject token repetitions at early stages of training. We call this a **Vestigial Heuristic** because it was developed early on in training to aid the initial bigram model but is not strictly needed by the model in later stages of training. Both the target and reference model will have a "Vestigial Heuristic" that prevents token repetition, so the text produced by the target model will activate the neurons responsible for the "Vestigial Heuristic" more strongly than in human text, which does not have the same "Vestigial Heuristic".

To investigate this hypothesis and its power in separating human written and LLM generated text, we propose Telescope Perplexity. While the formulation is strikingly simple, we argue its power lies in probing the specific hypothesized aversion to repetition that forms very early in training.

Telescope Perplexity measures the reference language model's learned likelihood of outputting the last token it processed given its context up to that point. For a language model $\mathcal{M}$ and a token sequence $\vec{s} = (s_1, ..., s_L)$ of length $L$, the Telescope Perplexity is defined as:

$$\text{Telescope PPL}_{\mathcal{M}}(\vec{s}) = -\frac{1}{L} \sum_{i=1}^{L} \log \mathcal{M}(s_i \mid s_{1:i}) \tag{1}$$

$\mathcal{M}(s_i \mid s_{1:i})$ is the probability assigned by the model $\mathcal{M}$ to token $s_i$ given the sequence $s_1...s_i$. This provides a targeted look into the model's token repetition likelihood, whereas standard Perplexity works by predicting the *next* token ($\mathcal{M}(s_i|s_{1:i-1})$). Figure 1 provides a conceptual overview of the Telescope Perplexity. Please note that the Telescope Perplexity deviates from standard Perplexity due to the fact that the negative log likelihood is not exponentiated; however, this is more efficient to compute and does not affect any detection results with a fixed threshold since exponentiation is a monotonic function.

## 1.3 Core Contributions

We propose and investigate the "Vestigial Heuristics" hypothesis, which presents a novel way to view how early training dynamics can affect the final model's behaviors. We introduce Telescope Perplexity (Eq. 1) as a simple, conceptually grounded probe specifically targeting one of these hypothesized **Vestigial Heuristics** that lead to a particular and extreme aversion to repetition. Additionally, we provide extensive testing to verify our hypothesis of the "Vestigial Heuristics" and the signature's early emergence/ locality, which allows us to better understand how Telescope Perplexity works, where it can fail, and how it can be improved in the future. Finally, we provide extensive empirical validation demonstrating Telescope Perplexity's state-of-the-art or competitive zero-shot detection performance and robustness across diverse scenarios, including on novel evaluation sets using contemporary LLMs (GPT4o Mini, Deepseek-V3).

## 2 Related Work

Detecting whether or not a piece of text is generated by some "target model" is extremely difficult because it requires high accuracy in the high dimensional and complex space of human text. For this reason, many opt not to analyze this space directly with neural networks or classical statistical models and instead choose to inference the text with a "reference model" and then analyze the output tokens of that reference model. Additional clarification about the relationship between reference models and target models can be found in the Appendix Section 9.1.

Early attempts at creating a zero-shot detector for LLM generated text consisted of using the log Perplexity or loss of a reference language model to detect text generated by a target language model. As previous work by Xu et al. (2024) has shown, a higher Perplexity score from a reference model means that the piece of text is less likely to be outputted by the reference model. Higher Perplexity can also help indicate that it has never seen any piece of text in its training similar to the piece of text in inference time, since large language models tend to write in distributions similar to the distributions found during training. Additionally, since large language models are generally trained on very similar, wide ranging datasets, if a reference LLM is perplexed by a piece of text and has never seen anything like it before, that means that other language models likely haven't seen anything like it in their training data. If other language models are less likely to have seen similar text, they would also be less likely to write the text (Choshen et al. (2021)), and this creates a separation between human written and LLM generated text.

Alongside Perplexity based techniques, there also exist rank based techniques, such as DetectLLM (Su et al. (2023)), which utilize the log rank of a logits distribution. The rank of a token in a distribution is the index of a token in a logits distribution sorted by their probabilities. The token with the most probability in a distribution has rank 1, the token with the second most probability in a distribution has rank 2, etc. The log rank is simply the natural logarithm of the computed rank. Using token rank, Su et al. (2023) devised a novel detector called LRR (Log Rank Ratio), which is highly accurate at distinguishing LLM generated and human written text.

The state of the art in zero shot black box detectors was achieved using a key refinement to Perplexity. This technique is called the Binoculars Score (Hans et al. (2024)), and it attempts to normalize the Perplexity score with a Cross Perplexity score from two models, so that it works better across domains. Computing the Binoculars Score requires two language models instead of one, which doubles the compute requirements relative to simply using Perplexity; however, Hans et al. (2024) show that it is worth the extra cost since the performance gains are very significant. To the best of our knowledge, the Binoculars Score currently stands as the state-of-the-art for zero shot detection of LLM generated text.

## 3 Understanding Vestigial Heuristics

Having introduced the "Vestigial Heuristics" hypothesis and the Telescope Perplexity probe (Eq. 1) designed to detect these persistent local biases, we now undertake an investigation to characterize the nature of the signal captured by Telescope Perplexity. We aim to understand its origins, structure, specific biases, and generality, thereby empirically grounding our hypothesis.

### 3.1 Training Dynamics

If the statistical biases Telescope Perplexity detects are indeed "Vestigial Heuristics" established early in training, we would expect the signature to emerge relatively quickly and remain stable throughout the later stages of pre-training. To investigate this, we analyzed the Telescope Perplexity of generated text using checkpoints from different stages of LLM pre-training for Amber-7B (Liu et al. (2023)) and Pythia (Biderman et al. (2023)). Figure 2 shows the evolution of Telescope Perplexity over text generated using the training checkpoints of Pythia-2.8B.

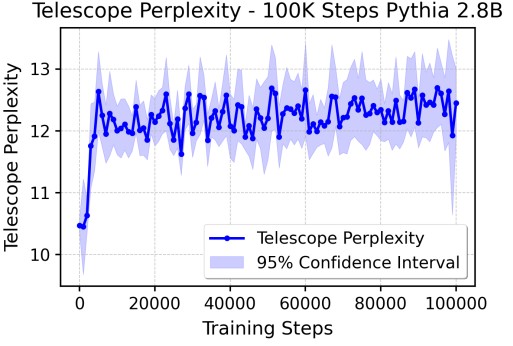

(a) Final Pythia checkpoint as reference model

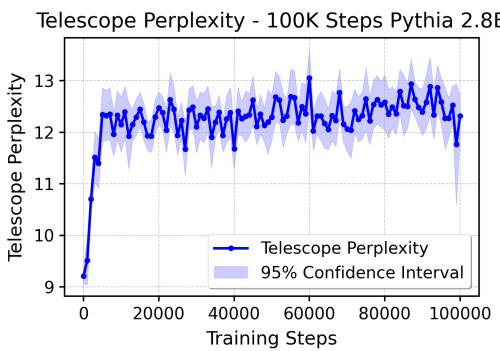

(b) SmolLM-360M as reference model

Figure 2: Telescope Perplexity evaluated on text generated by Pythia-2.8B model checkpoints throughout training. Note the early stabilization of the Telescope Perplexity.

The results in Figure 2 reveal a key characteristic: the Telescope Perplexity rises sharply early in training and then largely plateaus, indicating stability through later training stages. Similar stabilization was observed across various model sizes and architectures in Appendix 9.14. This early emergence and subsequent persistence strongly support the hypothesis that Telescope Perplexity captures a "Vestigial" characteristic established during the foundational learning phase, rather than a property that evolves continuously with model capability.

### 3.2 Signature Locality

The "Vestigial Heuristics" hypothesis posits that the aversion to repetition that we are probing relates to local, statistical patterns learned early in training. If Telescope Perplexity primarily captures behviors from an early stage bigram model, its effectiveness should largely persist even when the reference model is provided with very limited context. We tested this by computing Telescope Perplexity and standard Perplexity using only the preceding one (bigram context) or two (trigram context) tokens, compared to using the full preceding context available. Table 1 presents the detection performance (AUC) on the HC3 dataset under these conditions.

Table 1 reveals that Telescope Perplexity maintains a remarkably high AUC (0.897) even with purely bigram context, significantly outperforming standard Perplexity under the same constraint (0.761 AUC). This strong performance with extremely limited context confirms that the signature Telescope Perplexity measures is indeed fundamentally rooted in *local* token relationships, aligning with the hypothesis that it reflects simple pattern-formation biases learned early in training.

Table 1: Detection performance (AUC) on the HC3 dataset using Telescope Perplexity and standard Perplexity with full context versus limited (Bigram, Trigram) context provided to the reference model. Confidence Intervals (95%) are shown in parentheses. Results demonstrate Telescope Perplexity's strong reliance on local information.

| Method | Context | AUC (95% Confidence Interval) |
|---|---|---|
| Telescope | Full | 0.995 (0.993-0.996) |
| Telescope | Bigram | 0.897 (0.893-0.905) |
| Telescope | Trigram | 0.921 (0.915-0.926) |
| Perplexity | Full | 0.991 (0.990-0.993) |
| Perplexity | Bigram | 0.761 (0.753-0.770) |
| Perplexity | Trigram | 0.925 (0.915-0.926) |

## 3.3 Generality of the Phenomenon

Is the "Vestigial Heuristic" detected by Telescope Perplexity an idiosyncrasy of specific models or a more general artifact? Our experiments provide two lines of evidence for generality. First, as detailed in Section 5 (Table 2), Telescope Perplexity demonstrates strong detection performance when using a wide variety of reference models, spanning different architectures (Gemma, Llama, Falcon, SmolLM) and parameter counts (135M to 9B). This suggests the underlying signature is not confined to one model family.

Second, using the SmolLM 360M model and its distinct training corpus components FineWeb (human written data) and Cosmopedia V2 (synthetic LLM data) (Ben Allal et al. (2024)), we found that Telescope Perplexity could distinguish text originating from these different internal training sources with very high accuracy (AUC 0.996, F1 0.987). See Appendix 9.3 for additional details. This confirms models **do** internalize fine-grained statistical properties related to their training experience, lending credence to the idea that persistent, detectable artifacts like "Vestigial Heuristics" can indeed be generally learned. Importantly, this demonstrates that we are not measuring the text's similarity to the training data, but a property learned in training!

Taken together, these results suggest that the persistent local biases probed by Telescope Perplexity are likely a relatively general characteristic associated with current deep learning approaches to language modeling.

## 4 Experimental Setup

To empirically validate the effectiveness of Telescope Perplexity in detecting the hypothesized "Vestigial Heuristics" and compare it against existing methods, we designed a comprehensive experimental setup covering diverse datasets, contemporary language models, and evaluation procedures.

### 4.1 Datasets

We gathered an array of reputable datasets that distinguish between AI-generated text and human-generated text. These datasets include HC3 (Guo et al. (2023)), HC3 Plus (Su et al. (2024)), a popular Kaggle competition dataset named LLM - Detect AI Generated Text (King et al. (2023)), another, more difficult, Kaggle dataset named AI Vs Human Text (Gerami (2024)), and the set of Ghostbusters paper datasets (Verma et al. (2024)), whose names will be preceded by "GB" from now on to identify them easily.

#### 4.1.1 Novel Datasets with Stronger Target Models

One of the key limitations of the datasets in the literature is that they are generated using rather old language models, which are not commonly in use today; some have even been deprecated. This makes many of the publicly available datasets somewhat unrepresentative for modern LLM detection applications. Therefore, in addition to our existing suite of

datasets, we also generated several novel datasets using the GPT4o Mini (OpenAI et al. (2024)) and Deepseek V3 models (DeepSeek-AI et al. (2025)). These models are regarded to be highly capable by the research community, placing near or at the top in a variety of benchmarks, and importantly, the free tiers for both GPT4o Mini and Deepseek are quite generous for users and allow anyone access to them easily. Many of the datasets that we used did not provide the prompts given to the language models to generate the data; however, the essays and creative writing portions of the Ghostbusters dataset did contain the prompts used in their creation. Using these prompts, we generated the GB Essays GPT4o Mini, GB Creative Writing Deepseek, GB Creative Writing GPT4o Mini, and GB Creative Writing Deepseek datasets. Full details on prompts and generation parameters for these novel evaluation sets are provided in Appendix Section 9.7.

### 4.1.2 English as a Second Language Text

To evaluate our model's robustness to ESL (English Second Language) text, which is a known failure mode for AI-detection schema (Liang et al. (2023)), we introduce the ESL GPT4o Mini dataset. The human responses for this dataset are adapted from an existing corpus of ESL student essays (Franklin et al. (2022)). We then tasked GPT4o Mini to rewrite the essays to improve clarity and structure. Rewriting is a difficult environment for AI detection models since entire sentences and ideas are often re-used from the human written text, possibly obfuscating the statistical fingerprints.

### 4.1.3 Adversarial Perturbations

A student asking an LLM to write their essay may submit a rephrased version of an LLM's text. We wish to evaluate each detector's robustness to such adversarial attacks, using the "perturbations" datasets by Verma et al. (2024), which contains attacks that can range from rephrasing a word or sentence to changing the ordering of paragraphs. This dataset also contains varying levels of each perturbation, which allows us to test how each detector performs with varying levels of perturbations. We additionally test each method with a adversarial "humanizer" system described in 9.16.

## 4.2 Detection Methods and Reference Models

We evaluate our proposed Telescope Perplexity method (Eq. 1) against several established zero-shot baselines representing diverse approaches. These include standard Perplexity based on next-token prediction ($P(s_i|s_{1:i-1})$), the rank-based DetectLLM LRR, which utilizes log rank (Su et al. (2023)), and Binoculars, which utilizes Cross-Perplexity between two models (Hans et al. (2024)). To assess the generality and robustness of detection methods with respect to the underlying inference engine, we employed a diverse set of 10 reference models varying in size, architecture, and origin: Gemma 2 2B, Gemma 2 9B, Llama 3.1 8B, Falcon 7B, SmolLM 135M, SmolLM 360M, SmolLM 1.7B, SmolLM2 135M, SmolLM2 360M, SmolLM2 1.7B. When a single reference model is needed, we use the instruct variant, and in Binoculars, which requires two reference models, we use the instruct and pretrained variant.

## 4.3 Evaluation Metrics and Procedure

We primarily evaluate detection performance using the Area Under the Receiver Operating Curve (AUROC) score. We also report the maximum F1-Score achievable by finding the optimal threshold on the test set itself. To assess practical threshold robustness across domains, we employ a Transferability Test: for each target dataset, an optimal threshold is determined via logistic regression trained on scores from all *other* datasets, and the resulting F1-Score on the held out target dataset is reported. Unless otherwise stated, reported AUROC and Transferability F1-Scores are averaged across all 10 reference models.

## 5 EXPERIMENTAL RESULTS

We now present the empirical results evaluating Telescope Perplexity's effectiveness as a zero-shot detector, its robustness under various conditions, and its practical limitations.

### 5.1 DETECTION PERFORMANCE

First, we assess the overall detection performance of Telescope Perplexity compared to baselines across our suite of evaluation sets. Table 2 summarizes the average AUROC scores across all reference models for each dataset. Full results per reference model are available in Appendix Section 9.19.

Table 2: Detection performance (Average AUROC across 10 reference models) of Telescope Perplexity, Binoculars, Perplexity, and DetectLLM LRR across diverse datasets. Bold indicates the best performance per dataset. Since we test on so many different variations of SmolLM, our averaging will be inherently biased and overvalue performance on the SmolLM architecture. Readers are encouraged to view our full results with the performance of each reference model on each dataset with each detection technique in Appendix Figure 9.19

| Dataset | AUROC | | | |
| | Telescope (ours) | Binoculars | Perplexity | DetectLLM |
|---|---|---|---|---|
| Detect LLM Text | **0.99320** | 0.79382 | 0.89891 | 0.93347 |
| AI vs Human | **0.95918** | 0.87573 | 0.91193 | 0.91103 |
| HC3 | **0.99510** | 0.99412 | 0.99395 | 0.98212 |
| HC3 Plus | **0.98748** | 0.88344 | 0.90612 | 0.87604 |
| ESL GPT4o Mini | **0.99985** | 0.81157 | 0.83375 | 0.71806 |
| GB Essay ChatGPT | **0.99982** | 0.88163 | 0.99938 | 0.99800 |
| GB News ChatGPT | 0.97151 | 0.98886 | 0.98950 | **0.99068** |
| GB Creative ChatGPT | **0.99793** | 0.92174 | 0.95956 | 0.93796 |
| GB Essay GPT4o Mini | **0.99937** | 0.83141 | 0.99540 | 0.99471 |
| GB Creative GPT4o Mini | **0.99437** | 0.94769 | 0.97028 | 0.94906 |
| GB News Claude | 0.88038 | **0.91866** | 0.87195 | 0.86226 |
| GB Creative Claude | **0.97885** | 0.86070 | 0.89336 | 0.88309 |
| GB Essay Claude | **0.97499** | 0.79078 | 0.94051 | 0.95557 |
| GB Essay Deepseek V3 | **0.99974** | 0.99060 | 0.99966 | 0.99759 |
| GB Creative Deepseek V3 | **0.99603** | 0.99573 | 0.99189 | 0.97049 |

The results demonstrate that Telescope Perplexity consistently achieves high AUROC scores, often outperforming the current state-of-the-art method, Binoculars, as well as other strong baselines like Perplexity and DetectLLM LRR, when averaged across reference models and on each reference model. Notably, Telescope Perplexity shows exceptional performance on our novel evaluation sets featuring contemporary models like GPT4o Mini and Deepseek-V3 (e.g., GB Essay GPT4o Mini, ESL GPT4o Mini). This validates that probing the hypothesized "Vestigial Heuristic" via Telescope Perplexity provides a powerful and broadly effective signal for zero-shot LLM detection, even against modern target models.

### 5.2 ROBUSTNESS

Figure 3 demonstrates the impact of minimum text length on detector performance. While longer texts generally provide more evidence, Telescope Perplexity maintains strong performance even on relatively shorter texts (e.g., $< 100$ or $< 200$ words), suggesting the underlying local bias aggregates reliably. Its performance degrades less sharply with decreasing length compared to some baselines in certain datasets. For more examples, see Appendix Section 9.17.

To assess resilience to simple obfuscation, we evaluated performance on texts where words were increasingly replaced by synonyms and where sentences were increasingly paraphrased. Figure 4 illustrates that while performance degrades for all detectors as perturbations increase, Telescope Perplexity maintains relatively high AUROC compared to baselines, indicating the local bias it measures is not solely dependent on exact lexical choice. Further evaluation

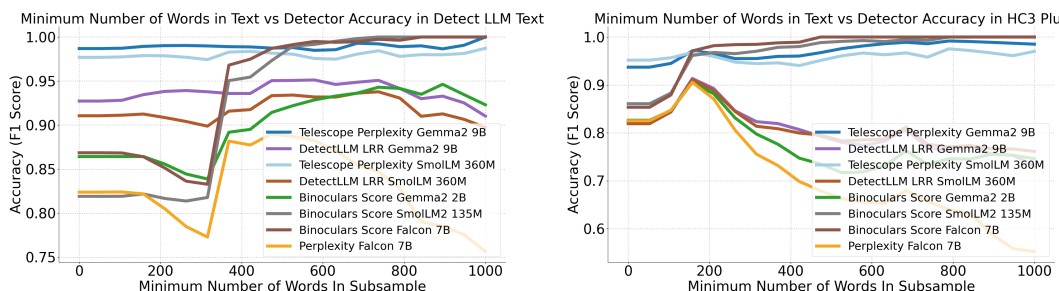

Figure 3: Impact of minimum text length on the AUROC performance of several detectors on the Detect LLM Text dataset and the HC3 Plus dataset.

on dedicated perturbation datasets by Verma et al. (2024) confirms Telescope Perplexity's robustness against various character, word, and paragraph-level modifications, which can be found in Appendix Section 9.17. Our tests of sophisticated "AI Humanizer" attacks show only small degradations in performance (see Appendix Section 9.16).

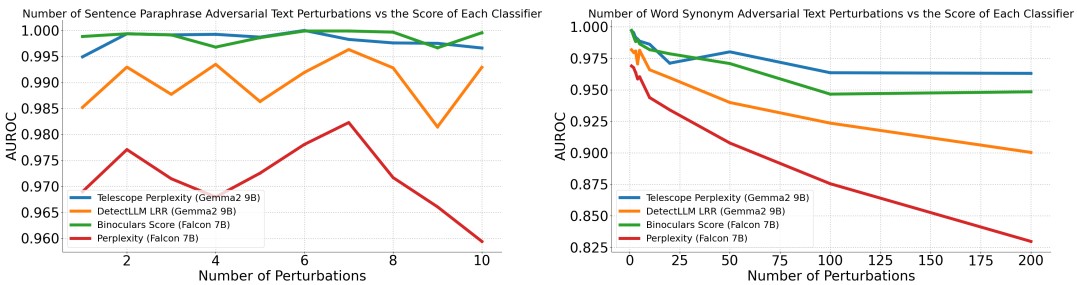

Figure 4: Impact of swapping words with synonyms and paraphrasing sentences on detector AUROC performance on the Ghostbusters dataset.

These results suggest the "Vestigial Heuristic" signature captured by Telescope Perplexity is reasonably robust to variations in text length and common perturbation strategies.

## 5.3 DOMAIN SENSITIVITY

While generally effective, Telescope Perplexity's performance varies across domains. As seen in Table 2, Telescope Perplexity achieves near-perfect scores when detecting ESL rewriting with an average 0.99985 AUROC across every reference model tested. We hypothesize this is a result of the ESL writing triggering the "Vestigial Heuristic" even more weakly than standard human text. Conversely, Telescope Perplexity's performance, while still often strong, is comparatively lower on some news writing datasets (e.g., Ghostbusters News Claude with an average AUROC of 0.88038 across every reference model tested). Additionally, Telescope Perplexity consistently misclassified AI-generated poetry as human-written, as seen in Appendix Section 9.8. This suggests that genres with highly stylized structures or perhaps very formulaic writing (potentially some news styles) might obscure the typical "Vestigial Heuristic" signature, which defines limitations for the current probe. Even though Telescope Perplexity seems to struggle with extremely formulaic text such as news writing, Telescope Perplexity also seems to perform better on essay writing than creative writing. Since essay writing is generally more formulaic than creative writing, this suggests that there isn't a clear direct correlation between detection performance and formulaicity.

## 5.4 PRACTICAL CONSIDERATIONS

Finally, we examine practical aspects relevant to deployment. Table 3 presents the results of our Transferability Test, evaluating how well a threshold tuned on N-1 datasets generalizes to the held-out Nth dataset.

Table 3: Transferability of Telescope Perplexity, Binoculars, Perplexity, and DetectLLM LRR when tuned on every other dataset and tested on a specific dataset. We report the F1-Score of an algorithm's performance on the test dataset averaged across all of the reference models tested. Similarly to Table 2, we also recommend that readers view our full results in Appendix Section 9.19

| Test Dataset | F1 Score | | | |
| --- | --- | --- | --- | --- |
| | Telescope (ours) | Binoculars | Perplexity | DetectLLM |
| GB Essay ChatGPT | **0.96703** | 0.83946 | 0.93737 | 0.94364 |
| GB News ChatGPT | 0.92323 | **0.95153** | 0.84217 | 0.89723 |
| GB Creative ChatGPT | **0.97105** | 0.86168 | 0.90985 | 0.87532 |
| GB Essay GPT4o Mini | **0.96682** | 0.81123 | 0.93982 | 0.94028 |
| GB Creative GPT4o Mini | **0.96215** | 0.88642 | 0.87746 | 0.87031 |
| GB News Claude | 0.71359 | **0.81504** | 0.76863 | 0.74661 |
| GB Creative Claude | **0.92302** | 0.79374 | 0.67196 | 0.74789 |
| GB Essay Claude | **0.91576** | 0.76458 | 0.85450 | 0.88074 |
| GB Essay Deepseek V3 | **0.96675** | 0.95048 | 0.93842 | 0.94473 |
| GB Creative Deepseek V3 | 0.96256 | 0.90692 | **0.96357** | 0.91913 |

While Telescope Perplexity often performs well even in this challenging scenario, its F1 score sometimes drops compared to its potential maximum F1 (achieved when tuning on the test set itself). This indicates that while the "Vestigial Heuristic" signature is generally present, its baseline level (and thus the optimal threshold) can vary slightly depending on the domain, style, or target model.

Like most other zero-shot detectors we tested, Telescope Perplexity's raw output scores are poorly calibrated and should not be interpreted as true probabilities. They serve as effective discriminative scores for classification but exhibit overconfidence as shown in Appendix Section 9.5.

## 6 LIMITATIONS

While we have attempted to rigorously test Telescope Perplexity against other LLM text detectors, there are still a couple of limitations to our experimentation methods due to compute constraints and practical limitations.

**1) Perturbation Based Detectors:** We do not benchmark on DetectLLM NPR, Detect-GPT or other perturbation based detection algorithms because of their massive computational cost. Preliminary testing showed worse performance on NPR compared to LRR, and there have been studies that show that DetectGPT heavily underperforms compared to other modern techniques. (Li et al. (2024)).

**2) Datasets Representative of Deployment:** We attempt to address data limitations in AI detection by introducing a suite of new datasets. While these datasets use modern LLMs as target models, they do not accurately reflect the data distribution in deployment environments where students may employ combinations of AI and human text, as well as adversarial "AI humanizers" to avoid detection.

## 7 CONCLUSION

In this work, we present the Telescope Perplexity metric, which we hypothesize probes an underlying behavior that forms early in training which discourages token repetition with a "Vestigial Heuristic". We demonstrate the local nature of this "Vestigial Heuristic" and track its development in LLM training. We empirically, through rigorous experimentation, compare our method with baselines and show that Telescope Perplexity is effective at discriminating between LLM generated and human written text and is particularly effective when tuned to a particular domain, while being robust to the choice of reference model.

## 8  Reproducibility Statement

Our implementation of Telescope Perplexity, and the code to reproduce experiments is open sourced on Github at _ _ _ _ _ _. This repository also includes detailed instructions for reproducing the main results of the paper. All standardized datasets used in this work are available at _ _ _ _ _ _ via Hugging Face. We notably exclude our "AI Humanizer" from our open source repository.

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

# 9 APPENDICES

## 9.1 CLARIFICATIONS ON VOCABULARY

Neural networks can become arbitrarily good at a task with enough data (Cybenko (1989)), and we can apply them to classify whether or not a piece of text has been generated by LLMs; however they do have some very major drawbacks. Simply using a neural networks to classify text generated by one language model may not perform well on language models it hasn't been trained on. In a world where new language models are released constantly, it is uniquely valuable to create an LLM generated text detector that will work on any language model trained in the future. In addition, data for these tasks on a wide variety of current language models in a wide variety of tasks is often difficult to find which can limit these detectors' reliability. This is why in the literature, there is a distinction between zero shot methods and non-zero shot methods; zero-shot methods promise to correctly classify LLM generated text without training on the specific language model that may have generated the text, while non-zero-shot methods must be trained on specific language models and may not be able to generalize to newly released language models. Since consumers will switch to whichever language model performs the best at a given point in time, zero shot methods are generally thought to be more robust but are more difficult to design and make accurate.

Because of their desirable properties, many researchers have focused on developing and improving the performance of these zero-shot detectors. Zero shot detection is a difficult problem since it requires directly analyzing the high dimensional space of text, so in an attempt to compress this space many of the zero shot detectors in the literature opt to use a reference model to help detect the text generated by a given target model. A target model is simply the model that may have generated the text under question, while the reference model is a model that a specific AI generated text detector may use to produce the probabilities for the next token for every word in the sentence. For example, in the sentence "The quick brown fox jumped over the lazy dog", the reference language model will produce the probability for any word to come after "The", "The quick", "The quick brown", etc. These probabilities can then be analyzed using metrics such as the average Perplexity of each word that the model predicted. It turns out that depending on the reference model used, the logits distributions can often convey very interesting information for whether the text under question was generated by some target language model. The target model is generally unknown, since we generally don't have a good idea which language models may have been used to generate text out in the wild; however, the reference model is chosen by the algorithm designer, and different reference models may have different properties that make them better or worse at detecting AI generated content from a variety of target models. The relationship between reference models and target models is illustrated in Figure 5. In this paper, we mainly focus on zero-shot methods and attempt to push the boundaries of the accuracy that can be achieved with them while still being able to easily generalize to any language model.

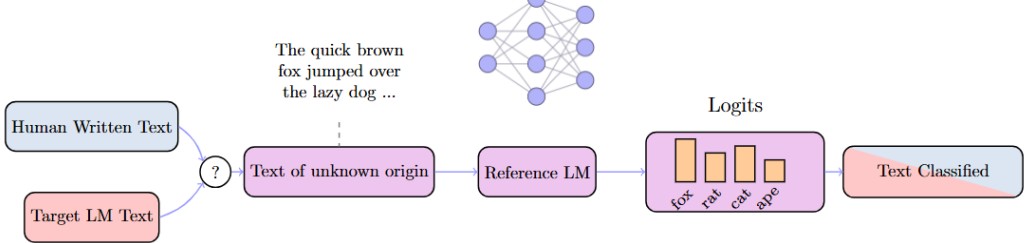

Figure 5: A graphical illustration of the relationship between the reference language model and the target language model.

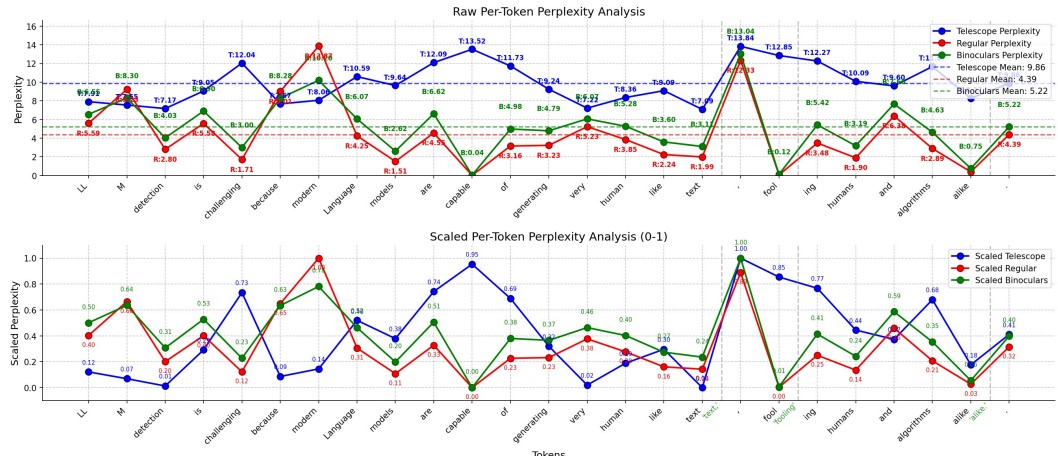

Figure 6: Per-token Telescope Perplexity, Perplexity, and Binoculars Score over the sequence

## 9.2 Analysis of Frequency Components of the Telescope Score

We now consider the single token Telescope Perplexity:

$$\text{Single Token Log Telescope PPL}_{\mathcal{M}}(s_i) = -log\mathcal{M}(s_i \mid s_{1:i}) \tag{2}$$

We can use this to plot our per-token Telescope Perplexity as in Figure 6. Readers familiar with signal processing may notice that the Telescope Perplexity over the token position resembles a signal. Interrogating this similarity, we found that the only truly relevant component of this signal is its mean. Future work applying sequence models in this domain is needed to fully evaluate if only the mean of this signal is truly important; our analysis applying signal processing yielded no results.

## 9.3 Training Data Separability

Here we present additional results on using SmolLM-360M to perform detection on a 10k subsample of its synthetic and human written training data. We find that, interestingly, standard Perplexity is able to somewhat effectively separate its own training data into synthetic and human data, implying Perplexity's effectiveness in detecting LLM generated text needs further study. The Binoculars Score does not effectively delineate between AI and human training data.

| Method | AUC | F1 Score |
|---|---|---|
| Binoculars | 0.7692 | 0.7327 |
| Perplexity | 0.9523 | 0.9056 |
| Telescope Perplexity | 0.9956 | 0.9874 |

Table 4: F1 and AUC of SmolLM-360M on its own training data.

## 9.4 Error Independence Analysis

We analyzed error independence across all detection methods to identify potential ensemble opportunities. For each classifier pair, we computed Cohen's Kappa, Yule's Q-statistic, and Normalized Mutual Information (NMI) to measure how independently they make errors. The Q-statistic, calculated as $Q = \frac{n_{11} \cdot n_{00} - n_{10} \cdot n_{01}}{n_{11} \cdot n_{00} + n_{10} \cdot n_{01}}$, ranges from -1 to 1, with lower values indicating greater error independence between classifiers.

Table 5 shows aggregated statistics across all datasets, sorted by Q-statistic. DetectLLM using Falcon 7B exhibited the highest independence (Q = 0.017), making errors on almost

completely different examples than other methods. Telescope Perplexity and standard Perplexity from the same model showed moderate dependence ($Q = 0.7$-$0.8$), while Binoculars showed lower dependence due to its Cross-Perplexity normalization.

| Detector | Reference Model | Avg Kappa | Avg Q-statistic | Avg Mutual Info |
|---|---|---|---|---|
| DetectLLM | Falcon 7B | 0.009 | 0.017 | 0.001 |
| Binoculars | SmolLM 1.7B | 0.400 | 0.572 | 0.040 |
| Binoculars | SmolLM 360M | 0.503 | 0.693 | 0.050 |
| Binoculars | Gemma2 9B | 0.605 | 0.731 | 0.091 |
| Perplexity | Gemma2 9B | 0.606 | 0.732 | 0.149 |
| Telescope Perplexity | Gemma2 9B | 0.613 | 0.735 | 0.067 |
| DetectLLM | Gemma2 9B | 0.606 | 0.736 | 0.115 |
| Binoculars | SmolLM 135M | 0.566 | 0.743 | 0.061 |
| DetectLLM | SmolLM2 135M | 0.611 | 0.775 | 0.122 |
| DetectLLM | SmolLM2 360M | 0.638 | 0.796 | 0.132 |
| DetectLLM | Gemma2 2B | 0.673 | 0.800 | 0.118 |
| DetectLLM | SmolLM 135M | 0.646 | 0.801 | 0.120 |
| DetectLLM | Llama3 8B | 0.678 | 0.804 | 0.126 |
| DetectLLM | SmolLM 360M | 0.678 | 0.814 | 0.137 |
| DetectLLM | SmolLM2 1.7B | 0.674 | 0.815 | 0.149 |
| DetectLLM | SmolLM 1.7B | 0.690 | 0.821 | 0.131 |
| Telescope Perplexity | Falcon 7B | 0.663 | 0.826 | 0.054 |
| Binoculars | Llama3 8B | 0.674 | 0.847 | 0.058 |
| Telescope Perplexity | SmolLM2 135M | 0.714 | 0.855 | 0.078 |
| Binoculars | SmolLM2 1.7B | 0.675 | 0.857 | 0.056 |
| Telescope Perplexity | SmolLM2 360M | 0.725 | 0.864 | 0.093 |
| Binoculars | SmolLM2 360M | 0.692 | 0.865 | 0.067 |
| Telescope Perplexity | SmolLM 135M | 0.724 | 0.867 | 0.086 |
| Telescope Perplexity | Gemma2 2B | 0.720 | 0.868 | 0.054 |
| Perplexity | SmolLM 135M | 0.728 | 0.869 | 0.179 |
| Perplexity | SmolLM2 135M | 0.727 | 0.869 | 0.191 |
| Binoculars | SmolLM2 135M | 0.715 | 0.871 | 0.086 |
| Telescope Perplexity | SmolLM2 1.7B | 0.744 | 0.873 | 0.086 |
| Binoculars | Gemma2 2B | 0.741 | 0.877 | 0.084 |
| Perplexity | SmolLM 360M | 0.747 | 0.878 | 0.197 |
| Telescope Perplexity | SmolLM 360M | 0.746 | 0.878 | 0.095 |
| Perplexity | SmolLM2 360M | 0.741 | 0.879 | 0.191 |
| Binoculars | Falcon 7B | 0.747 | 0.880 | 0.074 |
| Telescope Perplexity | SmolLM 1.7B | 0.750 | 0.881 | 0.092 |
| Telescope Perplexity | Llama3 8B | 0.749 | 0.881 | 0.087 |
| Perplexity | SmolLM 1.7B | 0.752 | 0.881 | 0.200 |
| Perplexity | Falcon 7B | 0.742 | 0.882 | 0.185 |
| Perplexity | SmolLM2 1.7B | 0.761 | 0.886 | 0.201 |
| Perplexity | Gemma2 2B | 0.766 | 0.887 | 0.192 |
| Perplexity | Llama3 8B | 0.771 | 0.890 | 0.182 |

Table 5: Error independence statistics aggregated across all datasets. Lower Q-statistic values indicate greater error independence.

Figure 7 visualizes pairwise error independence, with cooler colors indicating more independent errors.

These results suggest promising ensemble combinations: DetectLLM (especially with Falcon 7B) paired with any Perplexity-based method, different model sizes within the same architecture, and Telescope Perplexity combined with Binoculars. The low mutual information values (mostly $< 0.2$) indicate substantial opportunities for ensemble improvements through complementary error patterns.

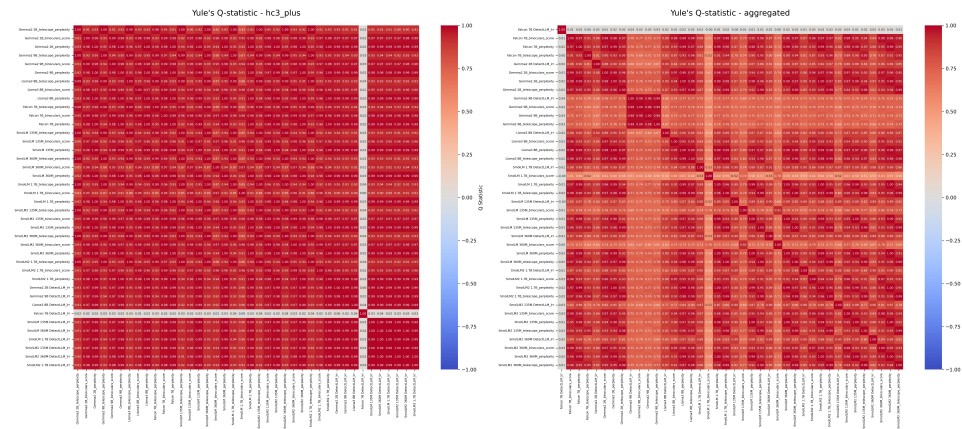

Figure 7: Q-statistic heatmaps for (left) HC3 Plus dataset and (right) aggregated across all datasets.

## 9.5 The Calibration of Zero-shot Detection Methods

Both Binoculars and Telescope Perplexity are natively very poorly calibrated 8. Meaning the scores produced by these models do not accuracy reflect the real probabilities of a given text being LLM generated. The calibration behavior of both detectors is similar to 8 with one particularly poorly calibrated example shown in Figure 9.

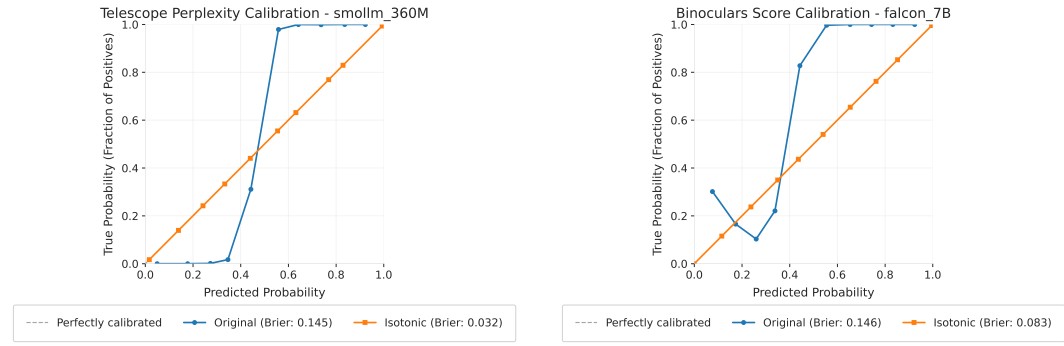

Figure 8: Calibration comparison between SmolLM 360M Telescope Perplexity (left) and Falcon Binoculars method (right)

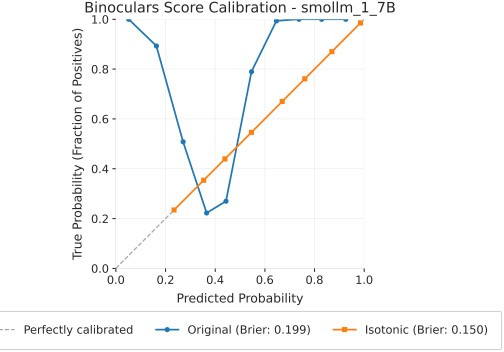

Figure 9: An extremely poorly calibrated Binoculars example using SmolLm-1.7B as a reference model.

## 9.6 Datasets

For each dataset, we filter out samples that are less than 100 words long in each dataset since when text is shorter, it may be statistically impossible to determine whether the text is written by a large language model. Shorter pieces of text have fewer different ways to write them and because of this, the way that a human is likely to write the text may be very similar to the way that an LLM would have written the text. Intuitively, this is similar to trying to make a decision based off of too little evidence. Including these short samples introduces unwanted variance to our measurements since the best the models can do some of the time is simply guess. To the best of our knowledge, there isn't any research specifically targeting how long the input to these detectors should be to minimize the data-points that are statistically impossible or difficult to classify for a theoretically perfect classifier, so we choose to filter out all text samples with fewer than 100 words from our datasets. A popular commercial detector called GPTZero also recommends that text be longer than 100 words for accurate classification. Other than following the lead of GPTZero, this decision is fairly arbitrary, and we decided on it at the beginning of testing and have not changed since. In addition, to save on computation cost, we also filter out any samples that are larger than 5000 words, since some of the LLM generated samples were incredibly long and would often induce out of memory errors. To reduce out of memory errors on our limited compute resources, we decided to simply take out the samples which were long enough to cause errors. For extremely long datasets with hundreds of thousands of samples, we only use the first 10,000 samples to save on compute.

## 9.7 Custom Datasets

To generate the ESL dataset, we used the following parameters for generation:

Table 6: Parameters for Generating the ESL Dataset

| Parameter | Value |
|---|---|
| System Prompt | "You are tasked with rewriting text. Provide ONLY the rewritten version, without additional comments, explanations, or formatting. Maintain the core meaning while improving clarity and structure." |
| Prompt | "Rewrite this text. Provide only the rewritten version" |
| Maximum length | None |
| Temperature | 0.7 |
| Top-p sampling | 1.0 |

To generate the GPT4o Mini and Deepseek versions of the Ghostbusters datasets, we used the following parameters for generation:

Table 7: Parameters for Generating the GPT4o Mini and Deepseek Versions of the Ghostbusters Datasets

| Parameter | Value |
|---|---|
| System Prompt | "You are a helpful assistant.\n\nUser: " |
| Prompt | "\n\nAssistant:" |
| Maximum length | 1024 |
| Temperature | 1.0 |
| Top-p sampling | 0.9 |

Below is an example of a sample from the ESL dataset:

**Human Text:** I think that students would benefit from learning at home,because they wont have to change and get up early in the morning to shower and do there hair. taking only classes helps them because at there house they'll be pay more attention. they will be comfortable at home. The hardest part of school is getting ready. you wake up go brush your teeth and go to your closet and look at your cloths. after you think you picked a outfit u go look in the mirror and youll either not like it or you look and see a stain. Then you'll have to change. with the online classes you can wear anything and stay home and you wont need to stress about what to wear. most students usually take showers before school. they either take it before they sleep or when they wake up. some students do both to smell good. that causes them do miss the bus and effects on there lesson time cause they come late to school. when u have online classes u wont need to miss lessons cause you can get everything set up and go take a shower and when u get out your ready to go. when your home your comfortable and you pay attention. it gives then an advantage to be smarter and even pass there classmates on class work. public schools are difficult even if you try. some teacher dont know how to teach it in then way that students understand it. that causes students to fail and they may repeat the class.

**LLM Rewritten Text:** I believe students would benefit from learning at home because they wouldn't need to wake up early to shower and style their hair. Attending classes from home allows them to focus better in a comfortable environment.The most challenging aspect of school is getting ready in the morning. You wake up, brush your teeth, and sift through your closet for clothes. After selecting an outfit, you might look in the mirror and either dislike it or notice a stain, forcing you to change again. With online classes, you can wear anything, stay at home, and avoid the stress of choosing an outfit. Most students typically shower before school, either at night or in the morning, and some do both to smell fresh. This routine can lead to them missing the bus, which impacts their lesson time as they arrive late. With online classes, they can set everything up and take a shower, so when they finish, they are ready to participate. Being at home provides comfort and enhances focus, giving students an advantage to excel and even surpass their classmates in assignments. Public schools can be challenging, even for dedicated students, as some teachers may struggle to present material in ways that students understand, which can lead to failure and the possibility of repeating a class.

Figure 10: Human Written Text from our custom ESL Dataset. This human text is then rewritten by an LLM to produce corresponding LLM generated text. The goal of the classifier is to distinguish which version was rewritten by an LLM, and which version was the original human written text. The human written text contains many typos and errors, which are then corrected by the LLM tasked with rewriting the text.

## 9.8 Example of Misclassification on Poetry/ Stylized Text

In this section, we show an example of an extremely misclassified human written text sample in the GB Creative Writing ChatGPT dataset that Telescope Perplexity very confidently classified as human written. This text sample is stylized to be pirate-like, which may indicate that this is a way to avoid detection (although this may be a rather impractical way to avoid detection in many circumstances).

---

**Text:**

Ahoy, me hearties, gather round, I'll spin for ye a tale profound, Of an island in the distant deep, Where dragons fly and secrets keep. 'Twas a land surrounded by azure seas, Where the dragon isles danced with the breeze, Guarded fiercely by three women bold, Their stories, legends yet untold. The first was Rosie, with flaming hair, A tempest rage, none could compare, With fiery spirit, she stood her ground, To protect the island, her dragons, renowned. Then there was Lily, with eyes so keen, A huntress born, her aim, unseen, With bow in hand and arrow true, No enemy could escape her view. Lastly was Bella, the wise and serene, Her words like gold, her heart a queen, She bore the knowledge of ancient roam, And in her presence, wisdom will bloom. Together they sailed amidst the tide, With dragon wings, they took their stride, The island shrouded in mystery rare, Its secrets guarded with utmost care. Dragons gleamed in the sun's warm glow, With scales that shimmered, a mighty show, In harmony, they danced and soared, Their fiery breath, their mighty hoard. But one day came a pirate crew, With greed and darkness to pursue, To snatch the dragons, to rule with might, And claim the island, lost from sight. But Rosie, Lily, and Bella stood, United as one, they knew they should, With swords unsheathed and hearts aflame, They fought for the dragons they held no blame. As the battle raged with thunderous sound, The pirates fled, their treasure unfound, The island stood proud, the dragons remained, A testament of the women unchained. So hear me shanty, a tale of courage bold, Of an island guarded against pirates untold, Through legends sung, let their bravery live, Three fierce women, their souls will thrive. Ahoy, me hearties, let our voices ring, Of the island where dragons took wing, And when the sea sings this tale divine, May their bravery echo throughout all time.

---

**Telescope Perplexity:** 12.028450012207031

---

**Label:** LLM Generated

---

**Decision Boundary for This Dataset:** 13.36

## 9.9 Misclassifications Analysis

Understanding misclassifications is important in any high impact application. We find that for Telescope Perplexity the expected distance from the threshold for a classification made in error is $\approx 1$ standard deviation, with human written texts misclassified as AI being less than that on average and AI text misclassified as human typically being more than a single standard deviation from the threshold. Figure 11 shows a couple of examples of the distribution of misclassifications with Deepseek-v3 being the only model in the entire suite of reference models to have a lower $\sigma$ for the AI generated misclassifications then human ones.

## 9.10 Statistical Significance and Confidence Intervals

To compute our confidence intervals, we use a python package by Gildenblat (2023), which contains functions to compute the 2 standard deviation confidence intervals of a AUROC score using the Delong method (DeLong et al. (1988), Sun and Xu (2014)) and the 2 standard deviation confidence intervals of the F1-Score using the Takahashi method (Takahashi et al. (2022)).

## 9.11 Poor Transfer Examples

If only shown a specific target model on a specific type of text, Telescope Perplexity can have poor threshold transfer capabilities. An illustrative example of poor threshold transfer

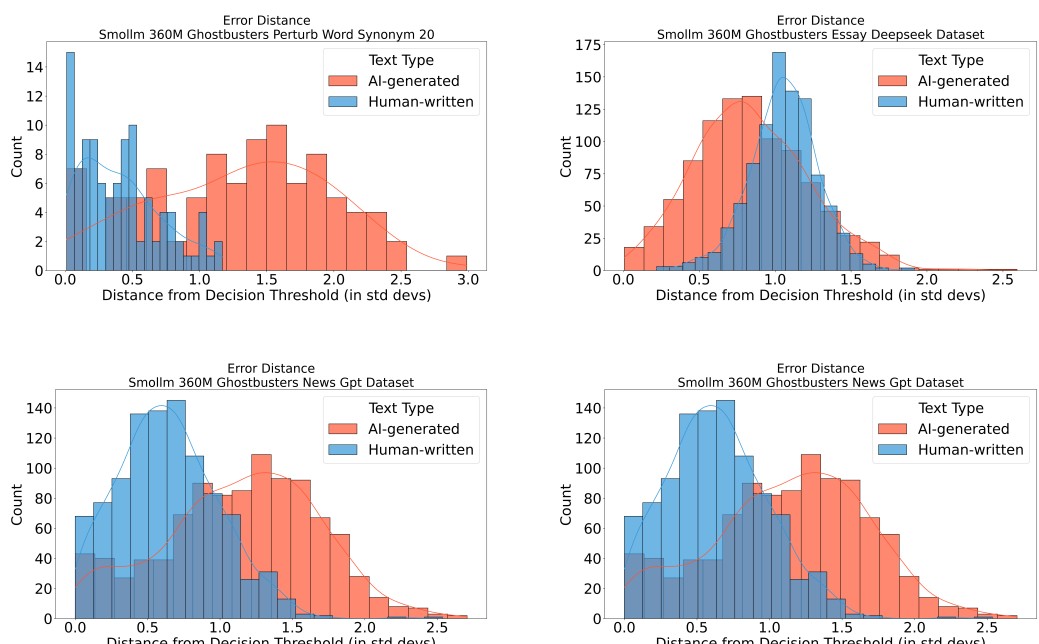

Figure 11: Error distance metrics for a 360M parameter model across different datasets and perturbation types. Top-left: Word Synonym Perturbation. Top-right: DeepSeek Essay Dataset. Bottom-left: ChatGPT News dataset. Bottom-right: Claude Essay Dataset.

for Telescope Perplexity is shown in Figure 12. This highlights the importance of using a diverse set of data to tune Telescope Perplexity's threshold.

Even with this poor transfer, it is important to note that Telescope Perplexity's transfer performance is still generally better than other approaches as seen in Appendix Section 9.19.

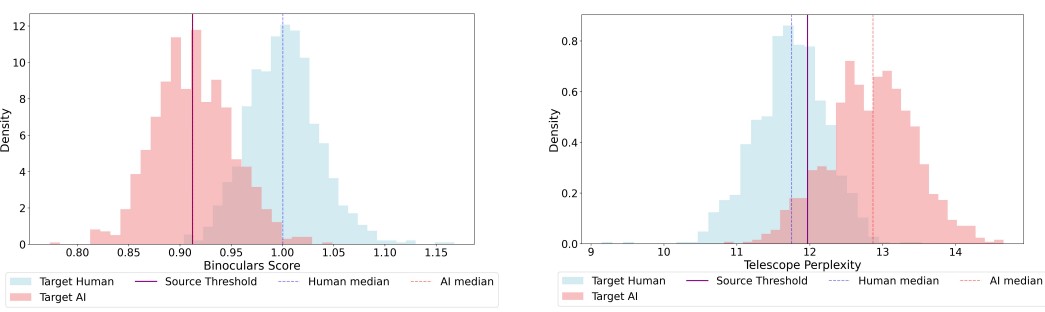

Figure 12: Example of poor threshold transfer. Comparing transfer from AI-Human to GB Creative Claude

## 9.12 COMPUTATIONAL RESOURCES AND EXECUTION TIME

The majority of these experiments were performed on Nvidia L40s GPUs with 48 GB of VRAM. Every model was run using 8 bit quantization using the BitsAndBytes[1] library to ensure that there was minimal loss of precision when benchmarking each detector, while also allowing us to run two models at once on single GPU. A subset of the ghostbusters datasets with large models such as Gemma2 9B were prone to out of memory exceptions, so for this reason, we utilized A100 GPUs with 80 GB of VRAM instead when necessary.

---

[1]https://huggingface.co/docs/bitsandbytes/en/index

Compute resource constraints did not allow us to test on reference models larger than 9 billion parameters. We have shown that in this range there is no direct correlation between model size and detector accuracy, but this relationship may not hold at larger model sizes.

The execution time of the experiments were highly dependent on the reference model used, the number of reference models a detection algorithm used, and the dataset. Different datasets contain different numbers of samples while different reference models may have more or less active parameters. In order to save time and effort while running these experiments, many of these experiments were run while batching multiple reference models and detection techniques in a single run, which further complicates matters, and makes it difficult to ascertain the run-time of a single experiment. Therefore in this section, we will report the total runtime of experiments on an L40s GPU on the Detect LLM Text dataset (10,000 samples). Knowing these execution times should provide a rough sense of how long all of the other experiments in this paper would take to run individually.

Table 8: Execution times of each technique used in this study on the Detect LLM Text dataset (10,000 samples) with L40s GPUs.

| Reference Model | Execution Time (minutes) | | | |
| --- | --- | --- | --- | --- |
| | Telescope | Binoculars | Perplexity | DetectLLM LRR |
| Gemma2 2B | 73.05 | 95.20 | 73.19 | 43.71 |
| Gemma2 9B | 90.53 | 123.63 | 90.47 | 61.00 |
| Llama3 8B | 57.47 | 80.53 | 57.78 | 47.13 |
| Falcon 7B | 42.00 | 62.97 | 42.86 | 45.02 |
| SmolLM 135M | 37.57 | 53.02 | 38.01 | 40.14 |
| SmolLM 360M | 38.08 | 36.83 | 29.06 | 33.27 |
| SmolLM 1.7B | 35.27 | 49.32 | 36.47 | 38.49 |
| SmolLM2 135M | 38.54 | 52.39 | 37.65 | 39.76 |
| SmolLM2 360M | 38.12 | 58.45 | 41.14 | 43.37 |
| SmolLM2 1.7B | 33.98 | 49.32 | 35.85 | 38.14 |

### 9.13 Unused Performance Metrics

Unlike Binoculars, we decide to completely disregard using the TPR at ultra low FPR score since this we simply do not have enough data in our datasets to make accurate and reliable measurements of behavior that only happens once in every 10,000 samples. We initially attempted to use these metrics, but during testing, we noticed substantial jumps of around 40% near the end of testing a dataset, which helps corroborate that using these FPR at low TPR scores as performance metrics to rate classifiers is unreliable and high variance for the number of samples that we have from each dataset. Other works by Tufts et al. (2025) show that the confidence intervals for this metric are massive and inconclusive.

### 9.14 Additional Training Dynamics

Liu et al. (2023) provide checkpoints throughout the training of there 7 billion parameter model Amber, these are taken regularly but a number of steps per checkpoint is not given. Compute constraints make evaluating all the Amber-7B model checkpoints infeasible so we instead focus on the early training steps to validate our hypothesis. We also present a chart showing the emergence of the "Vestigial Heuristic" in a much smaller model (Pythia-160M). As shown in Figure 13 we observe Telescope Perplexity rising early in training before stabilizing across both model architectures and across a few model sizes.

### 9.15 Adversarial Temperature and Prompt Attacks

In order to gauge how the temperature of the target model affects Telescope Perplexity's performance on detecting text from that target model, we have decided to provide a small

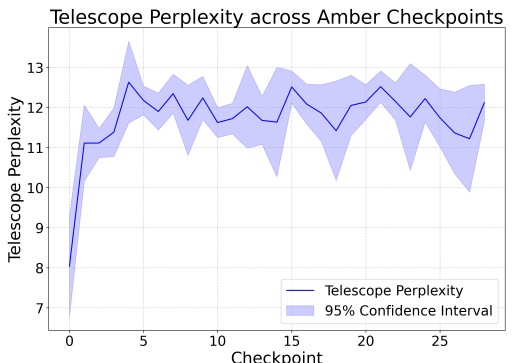

(a) Early training of Liu et al. (2023)'s Amber-7B

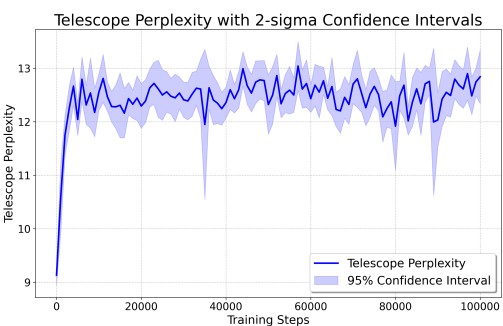

(b) First 100k training steps of Biderman et al. (2023)'s Pythia-160M model

Figure 13: Training curves for language models

ablation study where we generate new datasets using different target model sampling temperatures. To conduct this ablation, we regenerate the Ghostbusters Essay dataset with the GPT4o target model with sampling temperatures of 0, 1 (the default for the OpenAI API), and 1.2 (the highest temperature that was also stable). Using a reference model of SmolLM 360M, we find that with a temperature of 0 corresponds to an AUROC of 0.999992, a temperature of 1 corresponds to an AUROC of 0.99993, and a temperature of 1.2 corresponds to an AUROC of 0.9988. Overall, we see a trend that a lower sampling temperature leads to better detection performance, which makes sense, since as the temperature of the target language model decreases, the chance for the language model to accidentally mimic human text via random sampling also decreases.

In addition, we also wish to study how prompt attacks affect the performance of the Telescope Perplexity. One simple prompting attack that people may employ to dodge detection is asking the target model to attempt to repeat words, which may interfere with the token repetition signal. To accomplish this, we regenerated the Ghostbusters Essay dataset with the GPT4o target model, but instead of using the system prompt "You are a helpful assistant.", we use the system prompt "You are a helpful assistant. Try to repeat key words one after another while still following the prompt". In doing so, we found that the AUROC on the dataset with the reference model of SmolLM 360M degraded from 0.99993 to 0.996. This suggests a slight but statistically significant degradation; however, this is not as bad as one may expect from these types of adversarial prompting attacks.

## 9.16 Adversarial AI Humanizers

To test our techniques efficacy under more difficult conditions we create a so-called 'AI Humanizer' that uses a bert-based model to rephrase the sample text subject to an optimization problem to minimize the distance of the AI texts score from the human average.

Table 9: Detection performance of Telescope Perplexity, Binoculars, and Perplexity across humanizer-perturbed datasets using SmolLM 360M as reference model. We report the AUROC of each detection technique. The best performance on each dataset is bolded.

| Dataset | Reference Model | AUROC Telescope | Binoculars | Perplexity |
|---|---|---|---|---|
| GB Creative Claude | SmolLM 360M | **0.99015** | 0.88342 | 0.95402 |
| GB Creative DeepSeek | SmolLM 360M | 0.90043 | **0.99089** | 0.97240 |
| GB Creative GPT-4o Adversarial Prompt | SmolLM 360M | **0.98044** | 0.94541 | 0.97186 |
| GB Creative GPT-4o | SmolLM 360M | **0.98578** | 0.95538 | 0.93773 |
| GB Creative ChatGPT | SmolLM 360M | **0.99996** | 0.86604 | 0.98511 |
| GB Essay Claude | SmolLM 360M | **0.97786** | 0.79498 | 0.93211 |
| GB Essay DeepSeek | SmolLM 360M | 0.92084 | 0.91773 | **0.99876** |
| GB Essay GPT-4o Adversarial Prompt | SmolLM 360M | 0.84411 | 0.75711 | **0.97582** |
| GB Essay GPT-4o | SmolLM 360M | 0.94491 | 0.79534 | **0.98208** |
| GB Essay ChatGPT | SmolLM 360M | 0.99186 | 0.75106 | **0.99351** |
| GB News Claude | SmolLM 360M | 0.88834 | 0.90718 | **0.95337** |
| GB News ChatGPT | SmolLM 360M | 0.98715 | 0.99098 | **0.99831** |

Table 10: AUROC confidence intervals (95%) for detection methods across humanizer-perturbed datasets using SmolLM 360M. The method with the narrowest confidence interval for the highest AUROC is bolded.

| Dataset | Reference Model | AUROC Confidence Intervals Telescope | Binoculars | Perplexity |
|---|---|---|---|---|
| GB Creative Claude | SmolLM 360M | (0.97187, 1.00842) | (0.76823, 0.99861) | (0.90407, 1.00398) |
| GB Creative DeepSeek | SmolLM 360M | (0.86354, 0.93731) | (0.98246, 0.99931) | (0.95634, 0.98845) |
| GB Creative GPT-4o Adversarial Prompt | SmolLM 360M | 0.96345, 0.99743) | (0.91899, 0.97184) | (0.95185, 0.99187) |
| GB Creative GPT-4o | SmolLM 360M | (0.97541, 0.99615) | (0.93203, 0.97872) | (0.91150, 0.96397) |
| GB Creative ChatGPT | SmolLM 360M | (0.99983, 1.00008) | (0.82319, 0.90889) | (0.97252, 0.99770) |
| GB Essay Claude | SmolLM 360M | (0.96445, 0.99126) | (0.74451, 0.84547) | (0.90322, 0.96099) |
| GB Essay DeepSeek | SmolLM 360M | (0.89007, 0.95162) | (0.88480, 0.95067) | (0.99692, 1.00059) |
| GB Essay GPT-4o Adversarial Prompt | SmolLM 360M | (0.79349, 0.89472) | (0.70054, 0.81369) | (0.96055, 0.99109) |
| GB Essay GPT-4o | SmolLM 360M | (0.91988, 0.96994) | (0.74314, 0.84754) | (0.96786, 0.99631) |
| GB Essay ChatGPT | SmolLM 360M | (0.97905, 1.00468) | (0.69412, 0.80799) | (0.98685, 1.00017) |
| GB News Claude | SmolLM 360M | (0.85199, 0.92469) | (0.87398, 0.94039) | (0.93007, 0.97667) |
| GB News ChatGPT | SmolLM 360M | (0.97890, 0.99541) | (0.98257, 0.99939) | (0.99611, 1.00051) |

Table 11: F1 scores for detection methods across humanizer-perturbed datasets using SmolLM 360M as reference model. We report the F1 score of each detection technique. The best performance on each dataset is bolded.

| Dataset | Reference Model | F1 Score | | |
| --- | --- | --- | --- | --- |
| | | Telescope | Binoculars | Perplexity |
| GB Creative Claude | SmolLM 360M | **0.94000** | 0.88000 | 0.86000 |
| GB Creative DeepSeek | SmolLM 360M | 0.84333 | **0.96667** | 0.93000 |
| GB Creative GPT-4o Adversarial Prompt | SmolLM 360M | **0.96333** | 0.89000 | 0.94000 |
| GB Creative GPT-4o | SmolLM 360M | **0.94333** | 0.90000 | 0.87333 |
| GB Creative ChatGPT | SmolLM 360M | **0.99333** | 0.79667 | 0.95333 |
| GB Essay Claude | SmolLM 360M | **0.91000** | 0.70000 | 0.85667 |
| GB Essay DeepSeek | SmolLM 360M | 0.86000 | 0.84333 | **0.98667** |
| GB Essay GPT-4o Adversaria Prompt | SmolLM 360M | 0.81333 | 0.69667 | **0.91000** |
| GB Essay GPT-4o | SmolLM 360M | 0.87667 | 0.75000 | **0.95667** |
| GB Essay ChatGPT | SmolLM 360M | **0.98000** | 0.69333 | 0.97333 |
| GB News Claude | SmolLM 360M | 0.82667 | 0.81000 | **0.88000** |
| GB News ChatGPT | SmolLM 360M | 0.93000 | 0.97667 | **0.98667** |

## 9.17 ADDITIONAL ABLATION RESULTS

Figures 14 and 15 show model performance over text length. Figures 16, 17, and 18 show model performance over perturbations for a number of different perturbation regimes described in Verma et al. (2024).

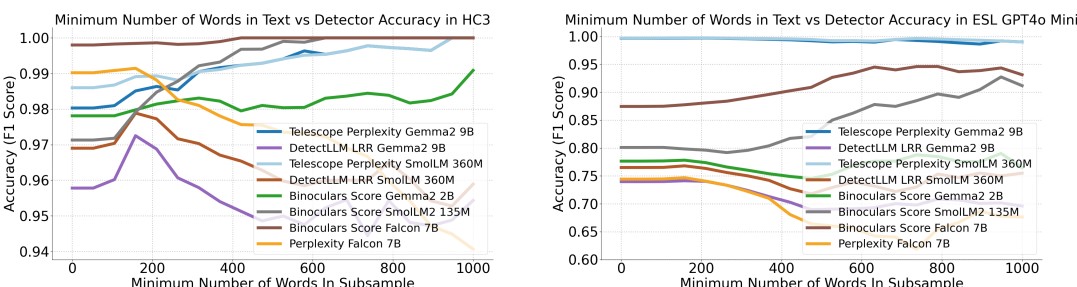

Figure 14: Impact of text length on the performance of several detectors. We filter out any words smaller than the minimum number of words from the dataset and then report the accuracy on that subsample of the data.

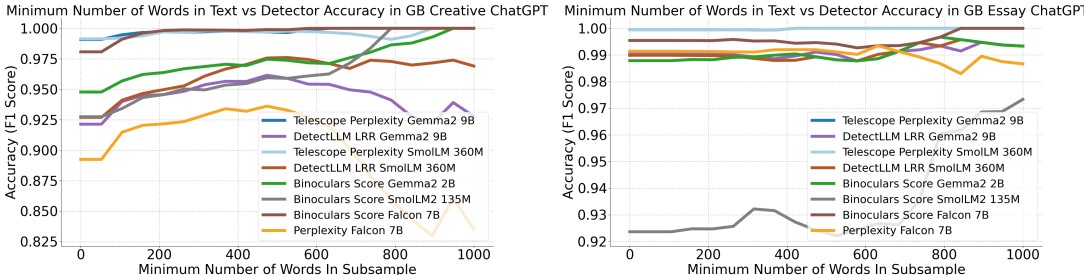

Figure 15: Impact of text length on the performance of several detectors. We filter out any words smaller than the minimum number of words from the dataset and then report the accuracy on that subsample of the data.

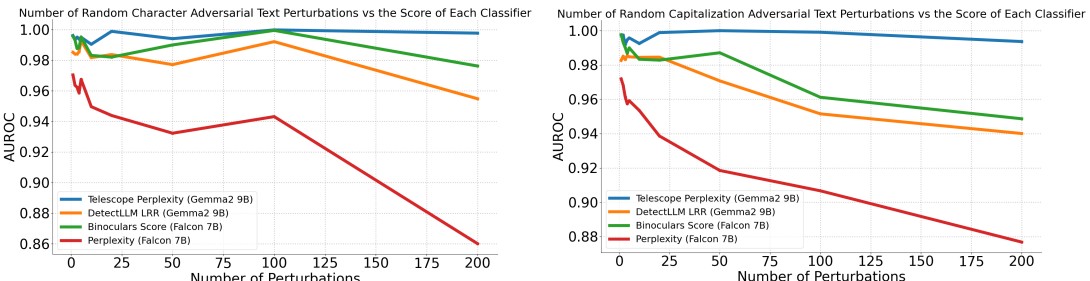

Figure 16: Impact of random character and random capitalization perturbations on the AUROC of each detector.

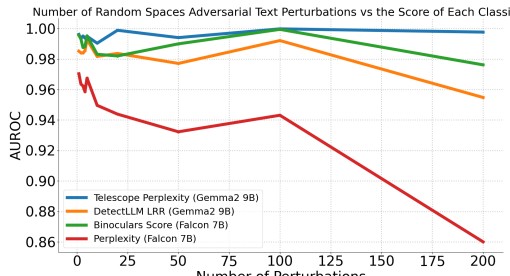

Figure 17: Impact of random space perturbations on the AUROC of each detector.

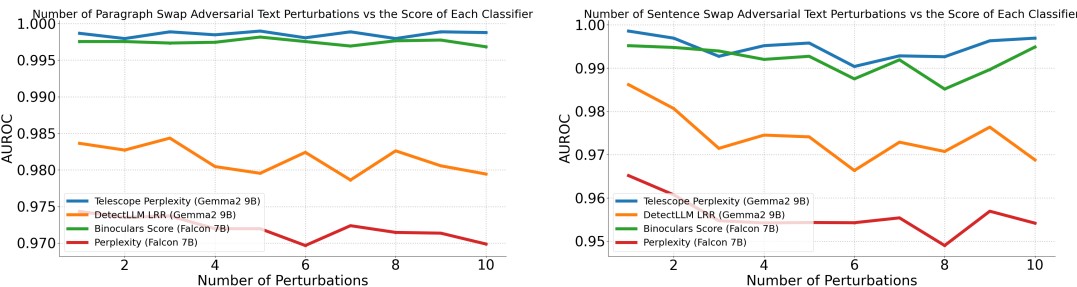

Figure 18: Impact of changing the ordering of paragraphs and sentences on the AUROC of each detector.

## 9.18 MULTILINGUAL ABLATION

The Telescope Perplexity probe is designed to apply to English-language models. For completeness, we include limited multilingual testing on Wang et al. (2024)'s M4 dataset using Team et al. (2024)'s Gemma2-2b. Variance in the original data sources across languages is large enough to make comparison between languages difficult. Additionally, Team et al. (2024)'s models are "not trained specifically for state-of-the-art multilingual capabilities," making the performance of each technique quite remarkable.

Table 12: Performance Metrics by Language and Method with Gemma2-2B as reference model

| Language | Metric | AUC | Confidence Interval |
|---|---|---|---|
| Chinese | Telescope Perplexity | 0.93092 | (0.89295, 0.96889) |
| | Binoculars Score | 0.98028 | (0.96183, 0.99874) |
| | Perplexity | 0.91321 | (0.86888, 0.95755) |
| | LRR | 0.90932 | (0.86777, 0.95088) |
| Russian | Telescope Perplexity | 0.69622 | (0.66416, 0.72828) |
| | Binoculars Score | 0.91334 | (0.89657, 0.93011) |
| | Perplexity | 0.72116 | (0.69027, 0.75204) |
| | LRR | 0.65098 | (0.61692, 0.68504) |
| Urdu | Telescope Perplexity | 0.9439 | (0.92869, 0.95920) |
| | Binoculars Score | 0.9999 | (0.99968, 1.00002) |
| | Perplexity | 0.9835 | (0.97634, 0.99071) |
| | LRR | 0.9796 | (0.97243, 0.98681) |
| Arabic | Telescope Perplexity | 0.94736 | (0.93425, 0.96046) |
| | Binoculars Score | 0.97469 | (0.96586, 0.98352) |
| | Perplexity | 0.92101 | (0.90331, 0.93872) |
| | LRR | 0.90559 | (0.88753, 0.92365) |

## 9.19 RAW EXPERIMENTAL RESULTS

Table 13: Detection performance of Telescope Perplexity, Binoculars, Perplexity, and DetectLLM LRR across datasets and reference models. We report the AUROC of each reference model on each dataset and detection technique. The best performance on a dataset is bolded.

| Dataset | Reference Model | AUROC | | | |
| --- | --- | --- | --- | --- | --- |
| | | Telescope | Binoculars | Perplexity | DetectLLM LRR |
| Detect LLM Text | Gemma2 2B | **0.99627** | 0.94029 | 0.90826 | 0.94490 |
| | Gemma2 9B | **0.99853** | 0.95644 | 0.85440 | 0.97712 |
| | Llama3 8B | **0.99271** | 0.86165 | 0.93482 | 0.88611 |
| | Falcon 7B | **0.98178** | 0.85026 | 0.88983 | 0.91463 |
| | SmolLM 135M | **0.99522** | 0.68742 | 0.90336 | 0.96209 |
| | SmolLM 360M | **0.99611** | 0.70934 | 0.92178 | 0.96744 |
| | SmolLM 1.7B | **0.99725** | 0.65607 | 0.89349 | 0.97851 |
| | SmolLM2 135M | **0.98979** | 0.70871 | 0.87103 | 0.90272 |
| | SmolLM2 360M | **0.99304** | 0.73988 | 0.88711 | 0.91351 |
| | SmolLM2 1.7B | **0.99132** | 0.82810 | 0.92501 | 0.88770 |
| AI vs Human | Gemma2 2B | **0.96146** | 0.90519 | 0.92369 | 0.91939 |
| | Gemma2 9B | **0.99533** | 0.96861 | 0.92884 | 0.97717 |
| | Llama3 8B | **0.96341** | 0.88881 | 0.93674 | 0.91168 |
| | Falcon 7B | 0.92601 | **0.94411** | 0.92006 | 0.90392 |
| | SmolLM 135M | **0.94966** | 0.81341 | 0.89354 | 0.89933 |
| | SmolLM 360M | **0.95920** | 0.79542 | 0.90200 | 0.91356 |
| | SmolLM 1.7B | **0.97198** | 0.75474 | 0.90678 | 0.92784 |
| | SmolLM2 135M | **0.94186** | 0.88560 | 0.88453 | 0.86757 |
| | SmolLM2 360M | **0.95319** | 0.89617 | 0.90367 | 0.88912 |
| | SmolLM2 1.7B | **0.96974** | 0.90528 | 0.91944 | 0.90076 |
| HC3 | Gemma2 2B | 0.99001 | 0.99647 | **0.99865** | 0.99505 |
| | Gemma2 9B | 0.99744 | 0.99473 | **0.99762** | 0.99154 |
| | Llama3 8B | 0.99374 | 0.99011 | **0.99636** | 0.98459 |
| | Falcon 7B | 0.99817 | **0.99898** | 0.99852 | 0.99332 |
| | SmolLM 135M | 0.99678 | 0.99176 | **0.99843** | 0.99475 |
| | SmolLM 360M | **0.99865** | 0.99134 | 0.99847 | 0.99421 |
| | SmolLM 1.7B | 0.99636 | 0.98957 | **0.99867** | 0.99106 |
| | SmolLM2 135M | 0.98877 | **0.99270** | 0.95722 | 0.89274 |
| | SmolLM2 360M | 0.99719 | **0.99883** | 0.99803 | 0.99425 |
| | SmolLM2 1.7B | 0.99393 | 0.99670 | **0.99753** | 0.98965 |
| HC3 Plus | Gemma2 2B | **0.98586** | 0.88591 | 0.92229 | 0.91333 |
| | Gemma2 9B | **0.98481** | 0.88504 | 0.91200 | 0.88826 |
| | Llama3 8B | **0.98696** | 0.90868 | 0.91041 | 0.88212 |
| | Falcon 7B | **0.97651** | 0.87907 | 0.89431 | 0.87889 |
| | SmolLM 135M | **0.99082** | 0.87527 | 0.89593 | 0.86697 |
| | SmolLM 360M | **0.99097** | 0.85483 | 0.89674 | 0.87039 |
| | SmolLM 1.7B | **0.98846** | 0.85201 | 0.89856 | 0.86909 |
| | SmolLM2 135M | **0.99166** | 0.91025 | 0.90703 | 0.85955 |
| | SmolLM2 360M | **0.99145** | 0.89248 | 0.90853 | 0.86163 |
| | SmolLM2 1.7B | **0.98733** | 0.89091 | 0.91535 | 0.87017 |
| ESL GPT4o Mini | Gemma2 2B | **0.99994** | 0.85862 | 0.84968 | 0.63665 |
| | Gemma2 9B | **0.99990** | 0.86940 | 0.69852 | 0.81899 |
| | Llama3 8B | **0.99992** | 0.78777 | 0.82383 | 0.59675 |
| | Falcon 7B | **0.99943** | 0.94200 | 0.80106 | 0.59125 |
| | SmolLM 135M | **0.99986** | 0.69568 | 0.87086 | 0.84039 |
| | SmolLM 360M | **0.99997** | 0.66281 | 0.84698 | 0.84095 |
| | SmolLM 1.7B | **0.99985** | 0.65235 | 0.84094 | 0.83732 |
| | SmolLM2 135M | **0.99979** | 0.88172 | 0.87168 | 0.69616 |
| | SmolLM2 360M | **0.99985** | 0.88361 | 0.87089 | 0.68868 |
| | SmolLM2 1.7B | **0.99995** | 0.88177 | 0.86309 | 0.63351 |

Table 14: Detection performance of Telescope Perplexity, Binoculars, Perplexity, and DetectLLM LRR across datasets and reference models. We report the AUROC of each reference model on each dataset and detection technique. The best performance on a dataset is bolded.

| Dataset | Reference Model | AUROC | | | |
|---|---|---|---|---|---|
| | | Telescope | Binoculars | Perplexity | DetectLLM LRR |
| GB Essay ChatGPT | Gemma2 2B | **0.99986** | 0.99900 | 0.99977 | 0.99862 |
| | Gemma2 9B | **1.00000** | 0.99854 | 0.99927 | 0.99948 |
| | Llama3 8B | **0.99999** | 0.94275 | 0.99954 | 0.99873 |
| | Falcon 7B | 0.99845 | **0.99994** | 0.99936 | 0.99689 |
| | SmolLM 135M | **0.99995** | 0.84014 | 0.99919 | 0.99733 |
| | SmolLM 360M | **1.00000** | 0.66860 | 0.99934 | 0.99895 |
| | SmolLM 1.7B | **0.99999** | 0.53700 | 0.99956 | 0.99912 |
| | SmolLM2 135M | **0.99999** | 0.97878 | 0.99867 | 0.99453 |
| | SmolLM2 360M | **1.00000** | 0.96815 | 0.99946 | 0.99764 |
| | SmolLM2 1.7B | **0.99999** | 0.88342 | 0.99967 | 0.99871 |
| GB News ChatGPT | Gemma2 2B | 0.97926 | **0.99819** | 0.99083 | 0.98720 |
| | Gemma2 9B | 0.99475 | **0.99617** | 0.97741 | 0.98308 |
| | Llama3 8B | 0.97193 | 0.98977 | 0.98966 | **0.99015** |
| | Falcon 7B | 0.85106 | **0.99979** | 0.99101 | 0.99229 |
| | SmolLM 135M | **0.99465** | 0.98102 | 0.99272 | 0.99243 |
| | SmolLM 360M | **0.99497** | 0.97836 | 0.99237 | 0.99339 |
| | SmolLM 1.7B | **0.99552** | 0.94788 | 0.99186 | 0.99389 |
| | SmolLM2 135M | 0.97685 | **0.99931** | 0.99159 | 0.99307 |
| | SmolLM2 360M | 0.97393 | **0.99976** | 0.99050 | 0.99286 |
| | SmolLM2 1.7B | 0.98224 | **0.99835** | 0.98703 | 0.98843 |
| GB Creative ChatGPT | Gemma2 2B | **0.99459** | 0.97578 | 0.96926 | 0.93619 |
| | Gemma2 9B | **0.99892** | 0.97495 | 0.96134 | 0.96911 |
| | Llama3 8B | **0.99839** | 0.94702 | 0.97879 | 0.93971 |
| | Falcon 7B | **0.99614** | 0.99560 | 0.94276 | 0.88114 |
| | SmolLM 135M | **0.99753** | 0.93538 | 0.96179 | 0.95977 |
| | SmolLM 360M | **0.99877** | 0.85480 | 0.96264 | 0.97139 |
| | SmolLM 1.7B | **0.99829** | 0.68608 | 0.96304 | 0.98008 |
| | SmolLM2 135M | **0.99814** | 0.96998 | 0.94289 | 0.89945 |
| | SmolLM2 360M | **0.99892** | 0.94987 | 0.95357 | 0.91260 |
| | SmolLM2 1.7B | **0.99965** | 0.92787 | 0.95954 | 0.93014 |
| GB Essay GPT4o Mini | Gemma2 2B | **0.99993** | 0.99788 | 0.99884 | 0.99801 |
| | Gemma2 9B | **1.00000** | 0.99669 | 0.99701 | 0.99955 |
| | Llama3 8B | **1.00000** | 0.99330 | 0.99846 | 0.99780 |
| | Falcon 7B | 0.99471 | **0.99982** | 0.99581 | 0.98953 |
| | SmolLM 135M | **0.99952** | 0.71978 | 0.98947 | 0.99176 |
| | SmolLM 360M | **0.99993** | 0.52293 | 0.99478 | 0.99819 |
| | SmolLM 1.7B | **0.99990** | 0.27946 | 0.99363 | 0.99833 |
| | SmolLM2 135M | **0.99980** | 0.97986 | 0.99077 | 0.98268 |
| | SmolLM2 360M | **0.99998** | 0.95253 | 0.99648 | 0.99361 |
| | SmolLM2 1.7B | **0.99999** | 0.87184 | 0.99878 | 0.99764 |
| GB Creative GPT4o Mini | Gemma2 2B | 0.99016 | **0.99692** | 0.98990 | 0.96815 |
| | Gemma2 9B | **0.99847** | 0.99417 | 0.98249 | 0.98570 |
| | Llama3 8B | **0.99851** | 0.98707 | 0.99008 | 0.97072 |
| | Falcon 7B | 0.98278 | **0.99900** | 0.93752 | 0.88254 |
| | SmolLM 135M | **0.98973** | 0.93788 | 0.96516 | 0.94462 |
| | SmolLM 360M | **0.99604** | 0.88977 | 0.97242 | 0.96671 |
| | SmolLM 1.7B | **0.99810** | 0.69969 | 0.97047 | 0.98419 |
| | SmolLM2 135M | 0.99273 | **0.99444** | 0.95470 | 0.89840 |
| | SmolLM2 360M | **0.99767** | 0.99325 | 0.96640 | 0.93059 |
| | SmolLM2 1.7B | **0.99955** | 0.98469 | 0.97370 | 0.95899 |
| GB News Claude | Gemma2 2B | 0.92323 | **0.95335** | 0.92909 | 0.92141 |
| | Gemma2 9B | **0.98606** | 0.98490 | 0.90440 | 0.97738 |
| | Llama3 8B | 0.94636 | 0.90848 | **0.96910** | 0.92222 |
| | Falcon 7B | 0.68234 | **0.95968** | 0.93036 | 0.90284 |
| | SmolLM 135M | **0.85742** | 0.85452 | 0.78960 | 0.78297 |
| | SmolLM 360M | **0.92549** | 0.85647 | 0.82940 | 0.85358 |
| | SmolLM 1.7B | **0.95882** | 0.87374 | 0.85406 | 0.88406 |
| | SmolLM2 135M | 0.78133 | **0.89907** | 0.78764 | 0.75799 |
| | SmolLM2 360M | 0.85359 | **0.93165** | 0.84057 | 0.80215 |
| | SmolLM2 1.7B | 0.88922 | **0.96472** | 0.88525 | 0.81799 |

Table 15: Detection performance of Telescope Perplexity, Binoculars, Perplexity, and DetectLLM LRR across datasets and reference models. We report the AUROC of each reference model on each dataset and detection technique. The best performance on a dataset is bolded.

| Dataset | Reference Model | AUROC | | | |
| --- | --- | --- | --- | --- | --- |
| | | Telescope | Binoculars | Perplexity | DetectLLM LRR |
| GB Creative Claude | Gemma2 2B | **0.96746** | 0.96120 | 0.93449 | 0.92119 |
| | Gemma2 9B | **0.99630** | 0.98096 | 0.94156 | 0.97729 |
| | Llama3 8B | **0.99248** | 0.85395 | 0.97208 | 0.94490 |
| | Falcon 7B | **0.97249** | 0.92225 | 0.89814 | 0.86264 |
| | SmolLM 135M | **0.95998** | 0.82650 | 0.83028 | 0.83977 |
| | SmolLM 360M | **0.98570** | 0.71090 | 0.86238 | 0.90021 |
| | SmolLM 1.7B | **0.98879** | 0.65321 | 0.88351 | 0.93776 |
| | SmolLM2 135M | **0.95535** | 0.87506 | 0.81886 | 0.74448 |
| | SmolLM2 360M | **0.97731** | 0.87312 | 0.87218 | 0.81175 |
| | SmolLM2 1.7B | **0.99267** | 0.94983 | 0.92011 | 0.89093 |
| GB Essay Claude | Gemma2 2B | 0.96250 | 0.94713 | **0.97973** | 0.97075 |
| | Gemma2 9B | 0.99448 | 0.99049 | 0.96795 | **0.99559** |
| | Llama3 8B | **0.99021** | 0.86505 | 0.98662 | 0.98410 |
| | Falcon 7B | 0.93540 | 0.96346 | **0.96349** | 0.95940 |
| | SmolLM 135M | **0.94992** | 0.49071 | 0.87188 | 0.92822 |
| | SmolLM 360M | **0.98389** | 0.37742 | 0.90774 | 0.97015 |
| | SmolLM 1.7B | 0.98279 | 0.50181 | 0.94059 | **0.98445** |
| | SmolLM2 135M | **0.96392** | 0.90800 | 0.89104 | 0.89514 |
| | SmolLM2 360M | **0.98676** | 0.90540 | 0.93776 | 0.95119 |
| | SmolLM2 1.7B | **1.00000** | 0.95833 | 0.95833 | 0.91667 |
| GB Essay Deepseek V3 | Gemma2 2B | **0.99995** | 0.99987 | 0.99976 | 0.99962 |
| | Gemma2 9B | **1.00000** | 0.99999 | 0.99954 | 0.99974 |
| | Llama3 8B | **0.99999** | 0.99921 | 0.99988 | 0.99690 |
| | Falcon 7B | 0.99795 | **0.99998** | 0.99916 | 0.99640 |
| | SmolLM 135M | **0.99979** | 0.98998 | 0.99959 | 0.99561 |
| | SmolLM 360M | **0.99993** | 0.97493 | 0.99981 | 0.99882 |
| | SmolLM 1.7B | **0.99998** | 0.94539 | 0.99988 | 0.99859 |
| | SmolLM2 135M | **0.99987** | 0.99953 | 0.99932 | 0.99347 |
| | SmolLM2 360M | **0.99997** | 0.99951 | 0.99979 | 0.99781 |
| | SmolLM2 1.7B | **1.00000** | 0.99756 | 0.99991 | 0.99893 |
| GB Creative Deepseek V3 | Gemma2 2B | 0.99592 | **0.99987** | 0.99811 | 0.98965 |
| | Gemma2 9B | 0.99973 | **0.99993** | 0.99879 | 0.99022 |
| | Llama3 8B | 0.99979 | **0.99983** | 0.99887 | 0.99144 |
| | Falcon 7B | 0.98625 | **0.99988** | 0.98682 | 0.96662 |
| | SmolLM 135M | 0.98708 | **0.99219** | 0.98588 | 0.94288 |
| | SmolLM 360M | **0.99700** | 0.98975 | 0.99117 | 0.96628 |
| | SmolLM 1.7B | **0.99877** | 0.97779 | 0.99328 | 0.97729 |
| | SmolLM2 135M | 0.99646 | **0.99875** | 0.98203 | 0.93775 |
| | SmolLM2 360M | 0.99941 | **0.99956** | 0.98894 | 0.96067 |
| | SmolLM2 1.7B | **0.99993** | 0.99978 | 0.99499 | 0.98213 |

Table 16: Detection performance of Telescope Perplexity, Binoculars, Perplexity, and DetectLLM LRR across a variety of perturbation schemes and reference models. We report the AUROC of each reference model on each perturbation scheme and detection technique. The best performance on a dataset is bolded.

| Dataset | Reference Model | AUROC | | | |
| --- | --- | --- | --- | --- | --- |
| | | Telescope | Binoculars | Perplexity | DetectLLM LRR |
| GB Spelling Error Perturbation | Gemma2 2B | 0.93238 | **0.95579** | 0.91095 | 0.91524 |
| | Gemma2 9B | **0.98474** | 0.96697 | 0.91064 | 0.95704 |
| | Llama3 8B | **0.95328** | 0.86936 | 0.93342 | 0.95318 |
| | Falcon 7B | 0.82275 | **0.96929** | 0.89840 | 0.88909 |
| | SmolLM 135M | **0.94586** | 0.53062 | 0.85337 | 0.87385 |
| | SmolLM 360M | **0.95360** | 0.45882 | 0.87281 | 0.91137 |
| | SmolLM 1.7B | **0.96206** | 0.42130 | 0.88660 | 0.92893 |
| | SmolLM2 135M | **0.90771** | 0.87301 | 0.83225 | 0.85013 |
| | SmolLM2 360M | **0.94147** | 0.85849 | 0.86204 | 0.88075 |
| | SmolLM2 1.7B | **0.96812** | 0.83163 | 0.89381 | 0.91712 |
| GB Character Capitalization Perturbation | Gemma2 2B | **0.99673** | 0.97514 | 0.93054 | 0.94180 |
| | Gemma2 9B | **1.00000** | 0.98404 | 0.93658 | 0.97064 |
| | Llama3 8B | **0.99785** | 0.84984 | 0.95591 | 0.94977 |
| | Falcon 7B | 0.91612 | **0.98773** | 0.91840 | 0.90437 |
| | SmolLM 135M | **0.95765** | 0.63922 | 0.87572 | 0.88881 |
| | SmolLM 360M | **0.96911** | 0.52046 | 0.89218 | 0.92021 |
| | SmolLM 1.7B | **0.97473** | 0.45888 | 0.90743 | 0.92860 |
| | SmolLM2 135M | 0.91510 | **0.92502** | 0.85403 | 0.85587 |
| | SmolLM2 360M | **0.96297** | 0.88104 | 0.88993 | 0.90170 |
| | SmolLM2 1.7B | **0.98732** | 0.83664 | 0.91735 | 0.92932 |
| GB Space Insertion Perturbation | Gemma2 2B | 0.96614 | **0.98014** | 0.94429 | 0.95485 |
| | Gemma2 9B | **0.99404** | 0.98997 | 0.94461 | 0.97722 |
| | Llama3 8B | **0.97502** | 0.90615 | 0.96112 | 0.96906 |
| | Falcon 7B | 0.87045 | **0.98983** | 0.92961 | 0.92811 |
| | SmolLM 135M | **0.94795** | 0.64402 | 0.90050 | 0.91043 |
| | SmolLM 360M | **0.95757** | 0.63200 | 0.91325 | 0.93687 |
| | SmolLM 1.7B | **0.96426** | 0.52801 | 0.92005 | 0.95192 |
| | SmolLM2 135M | 0.91210 | **0.93959** | 0.88043 | 0.89350 |
| | SmolLM2 360M | **0.93907** | 0.91796 | 0.90677 | 0.91900 |
| | SmolLM2 1.7B | **0.95736** | 0.90531 | 0.93499 | 0.94147 |
| GB Swap Adjacent Paragraphs Perturbation | Gemma2 2B | 0.99243 | **0.99509** | 0.97893 | 0.95765 |
| | Gemma2 9B | **0.99898** | 0.98660 | 0.96461 | 0.97985 |
| | Llama3 8B | **0.99376** | 0.92257 | 0.98128 | 0.96553 |
| | Falcon 7B | 0.93994 | **0.99800** | 0.97105 | 0.94887 |
| | SmolLM 135M | **0.99560** | 0.85188 | 0.96747 | 0.97606 |
| | SmolLM 360M | **0.99642** | 0.77833 | 0.96778 | 0.98517 |
| | SmolLM 1.7B | **0.99427** | 0.62909 | 0.96604 | 0.98517 |
| | SmolLM2 135M | **0.99274** | 0.97975 | 0.96543 | 0.95724 |
| | SmolLM2 360M | **0.99212** | 0.96471 | 0.96911 | 0.97330 |
| | SmolLM2 1.7B | **0.99417** | 0.90487 | 0.97330 | 0.97095 |
| GB Paraphrase Paragraphs Perturbation | Gemma2 2B | 0.97583 | **0.98288** | 0.96237 | 0.95127 |
| | Gemma2 9B | **0.99634** | 0.98850 | 0.93324 | 0.97008 |
| | Llama3 8B | **0.97230** | 0.88398 | 0.96237 | 0.96655 |
| | Falcon 7B | 0.90170 | **0.99918** | 0.95229 | 0.94428 |
| | SmolLM 135M | **0.99660** | 0.79788 | 0.96681 | 0.98119 |
| | SmolLM 360M | 0.98484 | 0.72877 | 0.95793 | **0.98994** |
| | SmolLM 1.7B | 0.99046 | 0.60988 | 0.95558 | **0.99321** |
| | SmolLM2 135M | **0.97596** | 0.96904 | 0.96094 | 0.95297 |
| | SmolLM2 360M | **0.98419** | 0.93415 | 0.96028 | 0.96381 |
| | SmolLM2 1.7B | **0.97870** | 0.88085 | 0.95754 | 0.97518 |

Table 17: Detection performance of Telescope Perplexity, Binoculars, Perplexity, and DetectLLM LRR across a variety of perturbation schemes and reference models. We report the AUROC of each reference model on each perturbation scheme and detection technique. The best performance on a dataset is bolded.

| Dataset | Reference Model | AUROC | | | |
| --- | --- | --- | --- | --- | --- |
| | | Telescope | Binoculars | Perplexity | DetectLLM LRR |
| GB Swap Adjacent Sentences Perturbation | Gemma2 2B | **0.98742** | 0.98732 | 0.97412 | 0.96696 |
| | Gemma2 9B | **0.99581** | 0.99243 | 0.96123 | 0.97453 |
| | Llama3 8B | **0.98169** | 0.93597 | 0.97085 | 0.95734 |
| | Falcon 7B | 0.93015 | **0.99291** | 0.95374 | 0.94025 |
| | SmolLM 135M | **0.99315** | 0.87019 | 0.96062 | 0.97770 |
| | SmolLM 360M | **0.99110** | 0.86283 | 0.96185 | 0.98292 |
| | SmolLM 1.7B | **0.98353** | 0.74529 | 0.96502 | 0.97903 |
| | SmolLM2 135M | 0.97473 | **0.99038** | 0.96154 | 0.94834 |
| | SmolLM2 360M | **0.98609** | 0.97831 | 0.96451 | 0.96062 |
| | SmolLM2 1.7B | **0.98711** | 0.92891 | 0.96972 | 0.96972 |
| GB Paraphrase Sentences Perturbation | Gemma2 2B | 0.98771 | **0.99466** | 0.98697 | 0.97308 |
| | Gemma2 9B | **0.99872** | 0.99551 | 0.97575 | 0.98632 |
| | Llama3 8B | **0.98814** | 0.91731 | 0.98622 | 0.97959 |
| | Falcon 7B | 0.93999 | **0.99835** | 0.97222 | 0.95160 |
| | SmolLM 135M | **0.98771** | 0.86752 | 0.97340 | 0.98173 |
| | SmolLM 360M | **0.98793** | 0.81122 | 0.97564 | 0.98579 |
| | SmolLM 1.7B | 0.98216 | 0.72372 | 0.97714 | **0.98622** |
| | SmolLM2 135M | 0.96635 | **0.99177** | 0.97489 | 0.95897 |
| | SmolLM2 360M | **0.97799** | 0.96806 | 0.97585 | 0.97169 |
| | SmolLM2 1.7B | **0.98942** | 0.89818 | 0.98088 | 0.98408 |
| GB Swap Adjacent Words Perturbation | Gemma2 2B | 0.73159 | **0.95162** | 0.89587 | 0.90190 |
| | Gemma2 9B | 0.88369 | **0.95724** | 0.90119 | 0.94415 |
| | Llama3 8B | 0.73108 | 0.90671 | 0.91203 | **0.92083** |
| | Falcon 7B | 0.44458 | **0.96672** | 0.87308 | 0.87556 |
| | SmolLM 135M | 0.85055 | 0.65702 | 0.84216 | **0.87797** |
| | SmolLM 360M | 0.86999 | 0.51954 | 0.84820 | **0.90446** |
| | SmolLM 1.7B | 0.88022 | 0.34707 | 0.85065 | **0.91581** |
| | SmolLM2 135M | 0.72944 | **0.89198** | 0.81567 | 0.84114 |
| | SmolLM2 360M | 0.70080 | **0.85843** | 0.83592 | 0.85812 |
| | SmolLM2 1.7B | 0.84390 | 0.76688 | 0.86221 | **0.90129** |
| GB Swap Word with Synonyms Perturbation | Gemma2 2B | 0.91111 | **0.94118** | 0.92134 | 0.91592 |
| | Gemma2 9B | **0.98016** | 0.92993 | 0.91070 | 0.93996 |
| | Llama3 8B | 0.88451 | 0.86794 | **0.93290** | 0.92993 |
| | Falcon 7B | 0.61860 | **0.93920** | 0.84003 | 0.82458 |
| | SmolLM 135M | **0.94957** | 0.77803 | 0.89791 | 0.91847 |
| | SmolLM 360M | **0.94896** | 0.63011 | 0.89914 | 0.93177 |
| | SmolLM 1.7B | **0.96522** | 0.43249 | 0.90640 | 0.93597 |
| | SmolLM2 135M | 0.88175 | **0.93914** | 0.88574 | 0.90088 |
| | SmolLM2 360M | 0.88124 | **0.91182** | 0.89495 | 0.90691 |
| | SmolLM2 1.7B | **0.94149** | 0.83101 | 0.91203 | 0.92185 |

Table 18: Two standard deviation confidence interval of Telescope Perplexity, Binoculars, Perplexity, and DetectLLM LRR across datasets and reference models. We report the confidence interval of the AUROC of each reference model on each perturbation scheme and detection technique.

| Dataset | Reference Model | AUROC | | | |
|---|---|---|---|---|---|
| | | Telescope | Binoculars | Perplexity | DetectLLM LRR |
| Detect LLM Text | Gemma2 2B | (0.99536, 0.99719) | (0.93594, 0.94464) | (0.90256, 0.91397) | (0.94060, 0.94920) |
| | Gemma2 9B | (0.99794, 0.99912) | (0.95287, 0.96001) | (0.84690, 0.86190) | (0.97456, 0.97968) |
| | Llama3 8B | (0.99152, 0.99390) | (0.85407, 0.86922) | (0.93035, 0.93929) | (0.87954, 0.89268) |
| | Falcon 7B | (0.97958, 0.98398) | (0.84110, 0.85943) | (0.88317, 0.89649) | (0.90890, 0.92037) |
| | SmolLM 135M | (0.99411, 0.99632) | (0.67503, 0.69981) | (0.89739, 0.90933) | (0.95864, 0.96554) |
| | SmolLM 360M | (0.99511, 0.99712) | (0.69776, 0.72093) | (0.91662, 0.92695) | (0.96433, 0.97054) |
| | SmolLM 1.7B | (0.99642, 0.99809) | (0.64391, 0.66824) | (0.88730, 0.89969) | (0.97609, 0.98092) |
| | SmolLM2 135M | (0.98824, 0.99135) | (0.69515, 0.72227) | (0.86376, 0.87830) | (0.89673, 0.90872) |
| | SmolLM2 360M | (0.99182, 0.99426) | (0.72723, 0.75254) | (0.88050, 0.89372) | (0.90797, 0.91904) |
| | SmolLM2 1.7B | (0.98993, 0.99272) | (0.81902, 0.83718) | (0.92008, 0.92994) | (0.88125, 0.89415) |
| AI vs Human | Gemma2 2B | (0.95757, 0.96535) | (0.89844, 0.91195) | (0.91756, 0.92981) | (0.91324, 0.92553) |
| | Gemma2 9B | (0.99280, 0.99786) | (0.96200, 0.97522) | (0.91740, 0.94028) | (0.97085, 0.98348) |
| | Llama3 8B | (0.95978, 0.96704) | (0.88103, 0.89658) | (0.93134, 0.94214) | (0.90530, 0.91806) |
| | Falcon 7B | (0.91939, 0.93262) | (0.93715, 0.95106) | (0.91262, 0.92750) | (0.89575, 0.91208) |
| | SmolLM 135M | (0.94504, 0.95427) | (0.80303, 0.82379) | (0.88603, 0.90106) | (0.89222, 0.90645) |
| | SmolLM 360M | (0.95516, 0.96323) | (0.78459, 0.80625) | (0.89486, 0.90914) | (0.90704, 0.92009) |
| | SmolLM 1.7B | (0.96877, 0.97518) | (0.74310, 0.76637) | (0.89989, 0.91368) | (0.92195, 0.93372) |
| | SmolLM2 135M | (0.93688, 0.94683) | (0.87691, 0.89429) | (0.87677, 0.89229) | (0.85940, 0.87574) |
| | SmolLM2 360M | (0.94879, 0.95759) | (0.88814, 0.90421) | (0.89664, 0.91070) | (0.88163, 0.89661) |
| | SmolLM2 1.7B | (0.96633, 0.97315) | (0.89810, 0.91246) | (0.91308, 0.92581) | (0.89386, 0.90767) |
| HC3 | Gemma2 2B | (0.98830, 0.99172) | (0.99538, 0.99756) | (0.99800, 0.99929) | (0.99390, 0.99621) |
| | Gemma2 9B | (0.99672, 0.99817) | (0.99344, 0.99601) | (0.99676, 0.99847) | (0.99006, 0.99302) |
| | Llama3 8B | (0.99215, 0.99533) | (0.98842, 0.99179) | (0.99508, 0.99765) | (0.98241, 0.98678) |
| | Falcon 7B | (0.99765, 0.99868) | (0.99816, 0.99980) | (0.99781, 0.99923) | (0.99198, 0.99465) |
| | SmolLM 135M | (0.99601, 0.99754) | (0.99022, 0.99329) | (0.99775, 0.99911) | (0.99353, 0.99597) |
| | SmolLM 360M | (0.99822, 0.99908) | (0.98983, 0.99285) | (0.99782, 0.99913) | (0.99295, 0.99547) |
| | SmolLM 1.7B | (0.99535, 0.99736) | (0.98796, 0.99117) | (0.99807, 0.99927) | (0.98945, 0.99266) |
| | SmolLM2 135M | (0.98703, 0.99050) | (0.99124, 0.99416) | (0.95266, 0.96178) | (0.88570, 0.89977) |
| | SmolLM2 360M | (0.99648, 0.99790) | (0.99803, 0.99962) | (0.99724, 0.99883) | (0.99298, 0.99552) |
| | SmolLM2 1.7B | (0.99232, 0.99553) | (0.99574, 0.99765) | (0.99663, 0.99844) | (0.98778, 0.99152) |
| HC3 Plus | Gemma2 2B | (0.98392, 0.98781) | (0.87911, 0.89270) | (0.91712, 0.92747) | (0.90775, 0.91890) |
| | Gemma2 9B | (0.98296, 0.98667) | (0.87819, 0.89188) | (0.90637, 0.91763) | (0.88177, 0.89474) |
| | Llama3 8B | (0.98473, 0.98918) | (0.90269, 0.91466) | (0.90473, 0.91609) | (0.87537, 0.88886) |
| | Falcon 7B | (0.97408, 0.97894) | (0.87165, 0.88649) | (0.88785, 0.90078) | (0.87181, 0.88598) |
| | SmolLM 135M | (0.98951, 0.99212) | (0.86809, 0.88244) | (0.88959, 0.90227) | (0.85958, 0.87437) |
| | SmolLM 360M | (0.98970, 0.99224) | (0.84722, 0.86244) | (0.89047, 0.90301) | (0.86312, 0.87766) |
| | SmolLM 1.7B | (0.98686, 0.99007) | (0.84444, 0.85958) | (0.89237, 0.90474) | (0.86181, 0.87637) |
| | SmolLM2 135M | (0.99044, 0.99288) | (0.90424, 0.91627) | (0.90121, 0.91285) | (0.85195, 0.86715) |
| | SmolLM2 360M | (0.99021, 0.99268) | (0.88579, 0.89918) | (0.90279, 0.91428) | (0.85411, 0.86915) |
| | SmolLM2 1.7B | (0.98538, 0.98929) | (0.88428, 0.89755) | (0.90989, 0.92082) | (0.86296, 0.87737) |
| ESL GPT4o Mini | Gemma2 2B | (0.99990, 0.99999) | (0.84932, 0.86792) | (0.83972, 0.85963) | (0.62260, 0.65070) |
| | Gemma2 9B | (0.99983, 0.99996) | (0.86054, 0.87825) | (0.68511, 0.71194) | (0.80845, 0.82952) |
| | Llama3 8B | (0.99985, 0.99999) | (0.77638, 0.79915) | (0.81309, 0.83456) | (0.58232, 0.61117) |
| | Falcon 7B | (0.99904, 0.99983) | (0.93618, 0.94783) | (0.78931, 0.81280) | (0.57633, 0.60617) |
| | SmolLM 135M | (0.99979, 0.99994) | (0.68241, 0.70895) | (0.86144, 0.88029) | (0.83037, 0.85040) |
| | SmolLM 360M | (0.99994, 0.99999) | (0.64906, 0.67656) | (0.83678, 0.85718) | (0.83099, 0.85091) |
| | SmolLM 1.7B | (0.99973, 0.99996) | (0.63851, 0.66620) | (0.83059, 0.85129) | (0.82730, 0.84734) |
| | SmolLM2 135M | (0.99961, 0.99996) | (0.87330, 0.89013) | (0.86228, 0.88108) | (0.68278, 0.70955) |
| | SmolLM2 360M | (0.99964, 1.00007) | (0.87521, 0.89201) | (0.86149, 0.88029) | (0.67524, 0.70211) |
| | SmolLM2 1.7B | (0.99991, 0.99998) | (0.87335, 0.89019) | (0.85354, 0.87264) | (0.61941, 0.64761) |

Table 19: Two standard deviation confidence interval of Telescope Perplexity, Binoculars, Perplexity, and DetectLLM LRR across datasets and reference models. We report the confidence interval of the AUROC of each reference model on each perturbation scheme and detection technique.

| Dataset | Reference Model | AUROC | | | |
| | | Telescope | Binoculars | Perplexity | DetectLLM LRR |
|---|---|---|---|---|---|
| GB Essay ChatGPT | Gemma2 2B | (0.99973, 1.00000) | (0.99844, 0.99957) | (0.99935, 1.00020) | (0.99760, 0.99964) |
| | Gemma2 9B | (0.99999, 1.00000) | (0.99773, 0.99935) | (0.99847, 1.00007) | (0.99911, 0.99984) |
| | Llama3 8B | (0.99998, 1.00001) | (0.93296, 0.95254) | (0.99894, 1.00014) | (0.99763, 0.99983) |
| | Falcon 7B | (0.99746, 0.99945) | (0.99988, 1.00000) | (0.99856, 1.00016) | (0.99526, 0.99853) |
| | SmolLM 135M | (0.99990, 1.00000) | (0.82283, 0.85745) | (0.99856, 0.99981) | (0.99601, 0.99866) |
| | SmolLM 360M | (1.00000, 1.00000) | (0.64461, 0.69260) | (0.99876, 0.99991) | (0.99806, 0.99983) |
| | SmolLM 1.7B | (0.99997, 1.00001) | (0.51108, 0.56291) | (0.99910, 1.00003) | (0.99827, 0.99996) |
| | SmolLM2 135M | (0.99998, 1.00001) | (0.97368, 0.98389) | (0.99772, 0.99963) | (0.99212, 0.99694) |
| | SmolLM2 360M | (0.99999, 1.00000) | (0.96137, 0.97492) | (0.99888, 1.00003) | (0.99626, 0.99901) |
| | SmolLM2 1.7B | (0.99998, 1.00001) | (0.86865, 0.89820) | (0.99927, 1.00006) | (0.99759, 0.99984) |
| GB News ChatGPT | Gemma2 2B | (0.97429, 0.98422) | (0.99704, 0.99933) | (0.98703, 0.99463) | (0.98272, 0.99168) |
| | Gemma2 9B | (0.99266, 0.99684) | (0.99449, 0.99786) | (0.97091, 0.98392) | (0.97757, 0.98860) |
| | Llama3 8B | (0.96479, 0.97906) | (0.98659, 0.99295) | (0.98552, 0.99381) | (0.98655, 0.99375) |
| | Falcon 7B | (0.83455, 0.86757) | (0.99950, 1.00008) | (0.98729, 0.99472) | (0.98907, 0.99551) |
| | SmolLM 135M | (0.99259, 0.99670) | (0.97543, 0.98661) | (0.98937, 0.99607) | (0.98905, 0.99581) |
| | SmolLM 360M | (0.99296, 0.99698) | (0.97225, 0.98446) | (0.98803, 0.99570) | (0.99023, 0.99654) |
| | SmolLM 1.7B | (0.99391, 0.99713) | (0.93796, 0.95780) | (0.98835, 0.99538) | (0.99095, 0.99683) |
| | SmolLM2 135M | (0.97143, 0.98227) | (0.99841, 1.00021) | (0.98791, 0.99526) | (0.98972, 0.99642) |
| | SmolLM2 360M | (0.96788, 0.97998) | (0.99955, 0.99998) | (0.98660, 0.99440) | (0.98958, 0.99613) |
| | SmolLM2 1.7B | (0.97732, 0.98715) | (0.99710, 0.99961) | (0.98241, 0.99164) | (0.98438, 0.99248) |
| GB Creative ChatGPT | Gemma2 2B | (0.99022, 0.99895) | (0.96793, 0.98363) | (0.96060, 0.97792) | (0.92377, 0.94862) |
| | Gemma2 9B | (0.99735, 1.00049) | (0.96839, 0.98152) | (0.95241, 0.97026) | (0.96164, 0.97657) |
| | Llama3 8B | (0.99754, 0.99923) | (0.93716, 0.95689) | (0.97265, 0.98493) | (0.92830, 0.95113) |
| | Falcon 7B | (0.99395, 0.99834) | (0.99281, 0.99840) | (0.93190, 0.95363) | (0.86561, 0.89667) |
| | SmolLM 135M | (0.99539, 0.99968) | (0.92413, 0.94663) | (0.95278, 0.97080) | (0.95129, 0.96825) |
| | SmolLM 360M | (0.99772, 0.99982) | (0.83778, 0.87182) | (0.95378, 0.97151) | (0.96408, 0.97870) |
| | SmolLM 1.7B | (0.99689, 0.99969) | (0.66240, 0.70977) | (0.95427, 0.97181) | (0.97402, 0.98615) |
| | SmolLM2 135M | (0.99686, 0.99942) | (0.96271, 0.97726) | (0.93180, 0.95397) | (0.88519, 0.91371) |
| | SmolLM2 360M | (0.99799, 0.99984) | (0.94028, 0.95946) | (0.94359, 0.96355) | (0.89929, 0.92591) |
| | SmolLM2 1.7B | (0.99936, 0.99993) | (0.91661, 0.93913) | (0.95037, 0.96872) | (0.91824, 0.94205) |
| GB Essay GPT4o Mini | Gemma2 2B | (0.99986, 1.00000) | (0.99677, 0.99900) | (0.99760, 1.00007) | (0.99675, 0.99927) |
| | Gemma2 9B | (0.99999, 1.00000) | (0.99528, 0.99810) | (0.99483, 0.99918) | (0.99919, 0.99992) |
| | Llama3 8B | (0.99999, 1.00000) | (0.99060, 0.99599) | (0.99711, 0.99980) | (0.99613, 0.99947) |
| | Falcon 7B | (0.99232, 0.99710) | (0.99962, 1.00002) | (0.99364, 0.99797) | (0.98611, 0.99294) |
| | SmolLM 135M | (0.99914, 0.99999) | (0.69436, 0.74521) | (0.98542, 0.99351) | (0.98837, 0.99514) |
| | SmolLM 360M | (0.99981, 1.00005) | (0.49166, 0.55421) | (0.99197, 0.99759) | (0.99671, 0.99967) |
| | SmolLM 1.7B | (0.99978, 1.00002) | (0.25383, 0.30509) | (0.99050, 0.99677) | (0.99679, 0.99987) |
| | SmolLM2 135M | (0.99960, 0.99999) | (0.97435, 0.98538) | (0.98730, 0.99425) | (0.97769, 0.98768) |
| | SmolLM2 360M | (0.99995, 1.00001) | (0.94353, 0.96152) | (0.99453, 0.99844) | (0.99095, 0.99628) |
| | SmolLM2 1.7B | (0.99998, 1.00001) | (0.85574, 0.88793) | (0.99792, 0.99964) | (0.99648, 0.99880) |
| GB Creative GPT4o Mini | Gemma2 2B | (0.98634, 0.99397) | (0.99527, 0.99857) | (0.98651, 0.99329) | (0.96138, 0.97491) |
| | Gemma2 9B | (0.99679, 1.00015) | (0.99159, 0.99675) | (0.97757, 0.98742) | (0.98167, 0.98972) |
| | Llama3 8B | (0.99774, 0.99928) | (0.98281, 0.99134) | (0.98667, 0.99349) | (0.96433, 0.97710) |
| | Falcon 7B | (0.97798, 0.98757) | (0.99827, 0.99973) | (0.92705, 0.94800) | (0.86756, 0.89752) |
| | SmolLM 135M | (0.98577, 0.99369) | (0.92695, 0.94881) | (0.95732, 0.97300) | (0.93477, 0.95448) |
| | SmolLM 360M | (0.99384, 0.99825) | (0.87477, 0.90477) | (0.96551, 0.97933) | (0.95948, 0.97394) |
| | SmolLM 1.7B | (0.99680, 0.99940) | (0.67549, 0.72389) | (0.96334, 0.97760) | (0.97947, 0.98890) |
| | SmolLM2 135M | (0.98968, 0.99577) | (0.99204, 0.99685) | (0.94575, 0.96365) | (0.88447, 0.91234) |
| | SmolLM2 360M | (0.99639, 0.99894) | (0.99054, 0.99596) | (0.95884, 0.97395) | (0.91942, 0.94176) |
| | SmolLM2 1.7B | (0.99911, 0.99999) | (0.98013, 0.98925) | (0.96732, 0.98008) | (0.95109, 0.96689) |
| GB News Claude | Gemma2 2B | (0.91199, 0.93448) | (0.94472, 0.96197) | (0.91812, 0.94006) | (0.90993, 0.93289) |
| | Gemma2 9B | (0.98210, 0.99002) | (0.98037, 0.98944) | (0.89134, 0.91747) | (0.97217, 0.98259) |
| | Llama3 8B | (0.93713, 0.95558) | (0.89441, 0.92256) | (0.96245, 0.97574) | (0.91095, 0.93349) |
| | Falcon 7B | (0.65829, 0.70638) | (0.95177, 0.96759) | (0.91929, 0.94142) | (0.88960, 0.91609) |
| | SmolLM 135M | (0.84124, 0.87359) | (0.83804, 0.87099) | (0.76988, 0.80932) | (0.76306, 0.80287) |
| | SmolLM 360M | (0.91429, 0.93669) | (0.83972, 0.87323) | (0.81165, 0.84715) | (0.83732, 0.86985) |
| | SmolLM 1.7B | (0.95112, 0.96651) | (0.85796, 0.88953) | (0.83773, 0.87040) | (0.86953, 0.89859) |
| | SmolLM2 135M | (0.76155, 0.80112) | (0.88573, 0.91240) | (0.76792, 0.80735) | (0.73724, 0.77873) |
| | SmolLM2 360M | (0.83754, 0.86963) | (0.92108, 0.94222) | (0.82341, 0.85773) | (0.78321, 0.82109) |
| | SmolLM2 1.7B | (0.87556, 0.90288) | (0.95756, 0.97188) | (0.87090, 0.89960) | (0.79979, 0.83619) |

Table 20: Two standard deviation confidence interval of Telescope Perplexity, Binoculars, Perplexity, and DetectLLM LRR across datasets and reference models. We report the confidence interval of the AUROC of each reference model on each perturbation scheme and detection technique.

| Dataset | Reference Model | AUROC | | | |
| --- | --- | --- | --- | --- | --- |
| | | Telescope | Binoculars | Perplexity | DetectLLM LRR |
| GB Creative Claude | Gemma2 2B | (0.96042, 0.97450) | (0.95328, 0.96911) | (0.92396, 0.94502) | (0.90971, 0.93267) |
| | Gemma2 9B | (0.99389, 0.99871) | (0.97575, 0.98618) | (0.93161, 0.95151) | (0.97175, 0.98284) |
| | Llama3 8B | (0.98960, 0.99536) | (0.83676, 0.87114) | (0.96539, 0.97878) | (0.93566, 0.95415) |
| | Falcon 7B | (0.96639, 0.97860) | (0.91029, 0.93421) | (0.88438, 0.91190) | (0.84678, 0.87850) |
| | SmolLM 135M | (0.95214, 0.96783) | (0.80834, 0.84466) | (0.81235, 0.84821) | (0.82253, 0.85701) |
| | SmolLM 360M | (0.98154, 0.98986) | (0.68791, 0.73389) | (0.84620, 0.87856) | (0.88684, 0.91358) |
| | SmolLM 1.7B | (0.98471, 0.99288) | (0.62846, 0.67795) | (0.86868, 0.89833) | (0.92758, 0.94794) |
| | SmolLM2 135M | (0.94717, 0.96353) | (0.85952, 0.89061) | (0.80038, 0.83733) | (0.72310, 0.76586) |
| | SmolLM2 360M | (0.97199, 0.98264) | (0.85758, 0.88866) | (0.85664, 0.88772) | (0.79309, 0.83041) |
| | SmolLM2 1.7B | (0.99011, 0.99523) | (0.94054, 0.95912) | (0.90808, 0.93215) | (0.87700, 0.90486) |
| GB Essay Claude | Gemma2 2B | (0.95471, 0.97029) | (0.93825, 0.95601) | (0.97475, 0.98471) | (0.96453, 0.97698) |
| | Gemma2 9B | (0.99163, 0.99733) | (0.98740, 0.99359) | (0.96115, 0.97476) | (0.99380, 0.99739) |
| | Llama3 8B | (0.98688, 0.99354) | (0.84856, 0.88154) | (0.98233, 0.99090) | (0.97978, 0.98843) |
| | Falcon 7B | (0.92479, 0.94600) | (0.95575, 0.97116) | (0.95607, 0.97090) | (0.95176, 0.96704) |
| | SmolLM 135M | (0.94091, 0.95893) | (0.46461, 0.51680) | (0.85661, 0.88715) | (0.91721, 0.93923) |
| | SmolLM 360M | (0.97916, 0.98862) | (0.35167, 0.40318) | (0.89509, 0.92039) | (0.96357, 0.97672) |
| | SmolLM 1.7B | (0.97799, 0.98758) | (0.47562, 0.52801) | (0.93072, 0.95047) | (0.98025, 0.98866) |
| | SmolLM2 135M | (0.95660, 0.97125) | (0.89523, 0.92078) | (0.87707, 0.90501) | (0.88139, 0.90888) |
| | SmolLM2 360M | (0.98287, 0.99066) | (0.89130, 0.91950) | (0.92772, 0.94780) | (0.94260, 0.95978) |
| | SmolLM2 1.7B | (1.00000, 1.00000) | (0.84284, 1.07383) | (0.84284, 1.07383) | (0.72145, 1.11188) |
| GB Essay Deepseek V3 | Gemma2 2B | (0.99987, 1.00004) | (0.99961, 1.00013) | (0.99943, 1.00009) | (0.99924, 0.99999) |
| | Gemma2 9B | (1.00000, 1.00000) | (0.99998, 1.00000) | (0.99888, 1.00020) | (0.99942, 1.00006) |
| | Llama3 8B | (0.99998, 1.00000) | (0.99786, 1.00056) | (0.99975, 1.00001) | (0.99453, 0.99928) |
| | Falcon 7B | (0.99686, 0.99904) | (0.99995, 1.00001) | (0.99825, 1.00007) | (0.99399, 0.99881) |
| | SmolLM 135M | (0.99955, 1.00003) | (0.98618, 0.99378) | (0.99914, 1.00004) | (0.99377, 0.99745) |
| | SmolLM 360M | (0.99984, 1.00002) | (0.96866, 0.98120) | (0.99959, 1.00004) | (0.99782, 0.99983) |
| | SmolLM 1.7B | (0.99995, 1.00001) | (0.93550, 0.95528) | (0.99974, 1.00001) | (0.99739, 0.99979) |
| | SmolLM2 135M | (0.99969, 1.00005) | (0.99914, 0.99992) | (0.99864, 1.00000) | (0.99080, 0.99614) |
| | SmolLM2 360M | (0.99992, 1.00002) | (0.99914, 0.99988) | (0.99953, 1.00006) | (0.99638, 0.99924) |
| | SmolLM2 1.7B | (0.99999, 1.00000) | (0.99522, 0.99991) | (0.99983, 1.00000) | (0.99791, 0.99995) |
| GB Creative Deepseek V3 | Gemma2 2B | (0.99401, 0.99784) | (0.99973, 1.00002) | (0.99685, 0.99938) | (0.98536, 0.99393) |
| | Gemma2 9B | (0.99926, 1.00019) | (0.99984, 1.00002) | (0.99789, 0.99968) | (0.98584, 0.99460) |
| | Llama3 8B | (0.99959, 0.99999) | (0.99963, 1.00002) | (0.99797, 0.99977) | (0.98732, 0.99555) |
| | Falcon 7B | (0.98199, 0.99050) | (0.99975, 1.00001) | (0.98234, 0.99130) | (0.95837, 0.97488) |
| | SmolLM 135M | (0.98261, 0.99155) | (0.98876, 0.99562) | (0.98082, 0.99093) | (0.93127, 0.95449) |
| | SmolLM 360M | (0.99559, 0.99840) | (0.98559, 0.99391) | (0.98734, 0.99500) | (0.95730, 0.97526) |
| | SmolLM 1.7B | (0.99806, 0.99947) | (0.97078, 0.98480) | (0.99028, 0.99629) | (0.97040, 0.98419) |
| | SmolLM2 135M | (0.99470, 0.99823) | (0.99776, 0.99974) | (0.97595, 0.98811) | (0.92541, 0.95008) |
| | SmolLM2 360M | (0.99896, 0.99987) | (0.99918, 0.99994) | (0.98446, 0.99342) | (0.95102, 0.97031) |
| | SmolLM2 1.7B | (0.99983, 1.00003) | (0.99951, 1.00006) | (0.99263, 0.99735) | (0.97618, 0.98808) |

Table 21: Two standard deviation confidence interval of Telescope Perplexity, Binoculars, Perplexity, and DetectLLM LRR across a variety of perturbation schemes and reference models. We report the confidence interval of the AUROC of each reference model on each perturbation scheme and detection technique.

| Dataset | Reference Model | AUROC | | | |
| --- | --- | --- | --- | --- | --- |
| | | Telescope | Binoculars | Perplexity | DetectLLM LRR |
| GB Spelling Error Perturbation | Gemma2 2B | (0.89706, 0.96770) | (0.92201, 0.98957) | (0.86514, 0.95677) | (0.87097, 0.95951) |
| | Gemma2 9B | (0.96960, 0.99988) | (0.94488, 0.98906) | (0.86559, 0.95569) | (0.92085, 0.99324) |
| | Llama3 8B | (0.92175, 0.98481) | (0.81703, 0.92168) | (0.89235, 0.97450) | (0.91602, 0.99034) |
| | Falcon 7B | (0.76016, 0.88534) | (0.93508, 1.00350) | (0.84511, 0.95170) | (0.83493, 0.94324) |
| | SmolLM 135M | (0.91329, 0.97843) | (0.44627, 0.61498) | (0.79617, 0.91056) | (0.82102, 0.92668) |
| | SmolLM 360M | (0.92204, 0.98515) | (0.37137, 0.54627) | (0.81892, 0.92669) | (0.86581, 0.95693) |
| | SmolLM 1.7B | (0.93630, 0.98782) | (0.33335, 0.50925) | (0.83489, 0.93831) | (0.88754, 0.97032) |
| | SmolLM2 135M | (0.86443, 0.95100) | (0.81885, 0.92718) | (0.77220, 0.89231) | (0.79362, 0.90663) |
| | SmolLM2 360M | (0.90884, 0.97410) | (0.80062, 0.91635) | (0.80655, 0.91753) | (0.82875, 0.93274) |
| | SmolLM2 1.7B | (0.94442, 0.99182) | (0.77290, 0.89035) | (0.84458, 0.94304) | (0.87279, 0.96145) |
| GB Character Capitalization Perturbation | Gemma2 2B | (0.99266, 1.00079) | (0.95477, 0.99551) | (0.88753, 0.97355) | (0.90654, 0.97705) |
| | Gemma2 9B | (1.00000, 1.00000) | (0.96987, 0.99822) | (0.90071, 0.97245) | (0.94812, 0.99317) |
| | Llama3 8B | (0.99519, 1.00052) | (0.79451, 0.90516) | (0.92390, 0.98792) | (0.91387, 0.98568) |
| | Falcon 7B | (0.87793, 0.95431) | (0.97356, 1.00189) | (0.87029, 0.96651) | (0.85482, 0.95392) |
| | SmolLM 135M | (0.92424, 0.99106) | (0.55782, 0.72061) | (0.81997, 0.93146) | (0.83644, 0.94118) |
| | SmolLM 360M | (0.94397, 0.99424) | (0.43327, 0.60765) | (0.83883, 0.94554) | (0.87139, 0.96904) |
| | SmolLM 1.7B | (0.95565, 0.99382) | (0.37143, 0.54633) | (0.85671, 0.95814) | (0.88300, 0.97420) |
| | SmolLM2 135M | (0.87054, 0.95966) | (0.88446, 0.96558) | (0.79533, 0.91273) | (0.79800, 0.91374) |
| | SmolLM2 360M | (0.93434, 0.99161) | (0.82902, 0.93305) | (0.83628, 0.94359) | (0.85019, 0.95320) |
| | SmolLM2 1.7B | (0.97159, 1.00305) | (0.77918, 0.89410) | (0.86905, 0.96565) | (0.88572, 0.97291) |
| GB Space Insertion Perturbation | Gemma2 2B | (0.94373, 0.98854) | (0.96222, 0.99806) | (0.90591, 0.98267) | (0.92311, 0.98659) |
| | Gemma2 9B | (0.98647, 1.00162) | (0.97830, 1.00163) | (0.90931, 0.97990) | (0.95532, 0.99911) |
| | Llama3 8B | (0.95347, 0.99658) | (0.86363, 0.94866) | (0.93016, 0.99209) | (0.94176, 0.99636) |
| | Falcon 7B | (0.81865, 0.92226) | (0.97081, 1.00885) | (0.88583, 0.97338) | (0.88486, 0.97136) |
| | SmolLM 135M | (0.91288, 0.98303) | (0.56294, 0.72510) | (0.85214, 0.94887) | (0.86342, 0.95744) |
| | SmolLM 360M | (0.92748, 0.98765) | (0.54763, 0.71638) | (0.86789, 0.95861) | (0.89417, 0.97958) |
| | SmolLM 1.7B | (0.93828, 0.99023) | (0.43957, 0.61645) | (0.87574, 0.96435) | (0.91438, 0.98946) |
| | SmolLM2 135M | (0.86772, 0.95648) | (0.90301, 0.97617) | (0.82832, 0.93255) | (0.84130, 0.94570) |
| | SmolLM2 360M | (0.90327, 0.97486) | (0.87928, 0.95663) | (0.85955, 0.95399) | (0.87261, 0.96539) |
| | SmolLM2 1.7B | (0.92624, 0.98848) | (0.86014, 0.95048) | (0.89458, 0.97540) | (0.90052, 0.98242) |
| GB Swap Adjacent Paragraphs Perturbation | Gemma2 2B | (0.98497, 0.99989) | (0.99012, 1.00006) | (0.95966, 0.99819) | (0.92879, 0.98651) |
| | Gemma2 9B | (0.99717, 1.00078) | (0.97609, 0.99711) | (0.94165, 0.98757) | (0.96480, 0.99490) |
| | Llama3 8B | (0.98698, 1.00054) | (0.88247, 0.96266) | (0.96449, 0.99807) | (0.93724, 0.99381) |
| | Falcon 7B | (0.90597, 0.97391) | (0.99480, 1.00121) | (0.94750, 0.99460) | (0.91564, 0.98210) |
| | SmolLM 135M | (0.99073, 1.00047) | (0.79712, 0.90664) | (0.94274, 0.99221) | (0.95689, 0.99524) |
| | SmolLM 360M | (0.99233, 1.00051) | (0.71092, 0.84575) | (0.94300, 0.99256) | (0.97118, 0.99915) |
| | SmolLM 1.7B | (0.98846, 1.00008) | (0.54732, 0.71086) | (0.94090, 0.99118) | (0.96700, 1.00333) |
| | SmolLM2 135M | (0.98543, 1.00004) | (0.96480, 0.99469) | (0.93978, 0.99107) | (0.92859, 0.98589) |
| | SmolLM2 360M | (0.98388, 1.00037) | (0.94124, 0.98818) | (0.94316, 0.99505) | (0.94893, 0.99767) |
| | SmolLM2 1.7B | (0.98594, 1.00240) | (0.86191, 0.94783) | (0.94988, 0.99673) | (0.94664, 0.99526) |
| GB Paraphrase Paragraphs Perturbation | Gemma2 2B | (0.95522, 0.99643) | (0.96764, 0.99813) | (0.93241, 0.99234) | (0.92306, 0.97947) |
| | Gemma2 9B | (0.99146, 1.00123) | (0.97740, 0.99960) | (0.89578, 0.97069) | (0.94971, 0.99045) |
| | Llama3 8B | (0.95097, 0.99364) | (0.83319, 0.93477) | (0.93175, 0.99300) | (0.94043, 0.99267) |
| | Falcon 7B | (0.85532, 0.94808) | (0.99762, 1.00073) | (0.91876, 0.98581) | (0.91218, 0.97637) |
| | SmolLM 135M | (0.99177, 1.00144) | (0.73079, 0.86498) | (0.93856, 0.99507) | (0.96630, 0.99607) |
| | SmolLM 360M | (0.96443, 1.00526) | (0.65265, 0.80489) | (0.92474, 0.99112) | (0.98019, 0.99969) |
| | SmolLM 1.7B | (0.97920, 1.00173) | (0.52400, 0.69575) | (0.92268, 0.98847) | (0.98586, 1.00055) |
| | SmolLM2 135M | (0.95569, 0.99623) | (0.94691, 0.99116) | (0.92959, 0.99228) | (0.92389, 0.98204) |
| | SmolLM2 360M | (0.96968, 0.99870) | (0.89913, 0.96917) | (0.92929, 0.99127) | (0.93859, 0.98903) |
| | SmolLM2 1.7B | (0.95949, 0.99792) | (0.83069, 0.93100) | (0.92603, 0.98905) | (0.95519, 0.99516) |

Table 22: Two standard deviation confidence interval of Telescope Perplexity, Binoculars, Perplexity, and DetectLLM LRR across a variety of perturbation schemes and reference models. We report the confidence interval of the AUROC of each reference model on each perturbation scheme and detection technique.

| Dataset | Reference Model | AUROC | | | |
|---|---|---|---|---|---|
| | | Telescope | Binoculars | Perplexity | DetectLLM LRR |
| GB Swap Adjacent Sentences Perturbation | Gemma2 2B | (0.97641, 0.99842) | (0.97397, 1.00066) | (0.94944, 0.99880) | (0.94015, 0.99377) |
| | Gemma2 9B | (0.99004, 1.00157) | (0.98525, 0.99962) | (0.93456, 0.98790) | (0.95349, 0.99557) |
| | Llama3 8B | (0.96529, 0.99809) | (0.90465, 0.96728) | (0.94615, 0.99555) | (0.92569, 0.98900) |
| | Falcon 7B | (0.89307, 0.96724) | (0.98327, 1.00254) | (0.92329, 0.98419) | (0.90703, 0.97348) |
| | SmolLM 135M | (0.98666, 0.99964) | (0.81919, 0.92119) | (0.93136, 0.98988) | (0.95634, 0.99906) |
| | SmolLM 360M | (0.98232, 0.99988) | (0.80867, 0.91698) | (0.93198, 0.99172) | (0.96607, 0.99977) |
| | SmolLM 1.7B | (0.97057, 0.99649) | (0.67273, 0.81786) | (0.93596, 0.99407) | (0.95653, 1.00153) |
| | SmolLM2 135M | (0.95341, 0.99606) | (0.98198, 0.99879) | (0.93257, 0.99050) | (0.91504, 0.98165) |
| | SmolLM2 360M | (0.97177, 1.00041) | (0.95934, 0.99729) | (0.93538, 0.99363) | (0.93066, 0.99058) |
| | SmolLM2 1.7B | (0.97335, 1.00087) | (0.89167, 0.96614) | (0.94312, 0.99632) | (0.94138, 0.99807) |
| GB Paraphrase Sentences Perturbation | Gemma2 2B | (0.97462, 1.00081) | (0.98885, 1.00046) | (0.96921, 1.00472) | (0.95114, 0.99501) |
| | Gemma2 9B | (0.99682, 1.00061) | (0.99067, 1.00036) | (0.95573, 0.99577) | (0.96893, 1.00372) |
| | Llama3 8B | (0.97723, 0.99905) | (0.87918, 0.95543) | (0.97032, 1.00212) | (0.95756, 1.00163) |
| | Falcon 7B | (0.90602, 0.97396) | (0.99502, 1.00168) | (0.94784, 0.99660) | (0.92045, 0.98274) |
| | SmolLM 135M | (0.97462, 1.00081) | (0.81528, 0.91976) | (0.94974, 0.99705) | (0.96305, 1.00041) |
| | SmolLM 360M | (0.97527, 1.00058) | (0.74674, 0.87570) | (0.95201, 0.99927) | (0.96924, 1.00234) |
| | SmolLM 1.7B | (0.96516, 0.99916) | (0.64774, 0.79970) | (0.95370, 1.00058) | (0.96519, 1.00724) |
| | SmolLM2 135M | (0.94108, 0.99161) | (0.98417, 0.99937) | (0.95065, 0.99914) | (0.93258, 0.98537) |
| | SmolLM2 360M | (0.95640, 0.99958) | (0.94407, 0.99204) | (0.95169, 1.00002) | (0.94602, 0.99735) |
| | SmolLM2 1.7B | (0.97536, 1.00349) | (0.85060, 0.94577) | (0.95837, 1.00338) | (0.96229, 1.00587) |
| GB Swap Adjacent Words Perturbation | Gemma2 2B | (0.66144, 0.80173) | (0.91533, 0.98790) | (0.84487, 0.94686) | (0.85062, 0.95319) |
| | Gemma2 9B | (0.83579, 0.93160) | (0.92388, 0.99060) | (0.85190, 0.95048) | (0.90688, 0.98141) |
| | Llama3 8B | (0.66148, 0.80067) | (0.86353, 0.94989) | (0.86393, 0.96013) | (0.87371, 0.96794) |
| | Falcon 7B | (0.35994, 0.52922) | (0.93366, 0.99979) | (0.81474, 0.93142) | (0.81799, 0.93312) |
| | SmolLM 135M | (0.79515, 0.90595) | (0.57796, 0.73607) | (0.78210, 0.90223) | (0.82376, 0.93218) |
| | SmolLM 360M | (0.81805, 0.92193) | (0.43443, 0.60464) | (0.78924, 0.90716) | (0.85364, 0.95528) |
| | SmolLM 1.7B | (0.83187, 0.92857) | (0.26621, 0.42793) | (0.79182, 0.90949) | (0.86730, 0.96433) |
| | SmolLM2 135M | (0.65869, 0.80019) | (0.84341, 0.94055) | (0.75251, 0.87883) | (0.78131, 0.90098) |
| | SmolLM2 360M | (0.62794, 0.77365) | (0.80572, 0.91113) | (0.77549, 0.89636) | (0.80048, 0.91576) |
| | SmolLM2 1.7B | (0.78859, 0.89921) | (0.69973, 0.83403) | (0.80576, 0.91866) | (0.84993, 0.95265) |
| GB Swap Word with Synonyms Perturbation | Gemma2 2B | (0.86923, 0.95298) | (0.90085, 0.98152) | (0.87530, 0.96738) | (0.86819, 0.96364) |
| | Gemma2 9B | (0.95891, 1.00140) | (0.88777, 0.97209) | (0.86402, 0.95738) | (0.89657, 0.98334) |
| | Llama3 8B | (0.83784, 0.93118) | (0.81696, 0.91893) | (0.88876, 0.97704) | (0.88450, 0.97536) |
| | Falcon 7B | (0.50681, 0.73040) | (0.87293, 1.00547) | (0.74407, 0.93600) | (0.72793, 0.92124) |
| | SmolLM 135M | (0.91723, 0.98191) | (0.70964, 0.84642) | (0.84719, 0.94864) | (0.87161, 0.96533) |
| | SmolLM 360M | (0.91745, 0.98047) | (0.54775, 0.71248) | (0.84825, 0.95003) | (0.88635, 0.97719) |
| | SmolLM 1.7B | (0.94216, 0.98828) | (0.34810, 0.51687) | (0.85712, 0.95568) | (0.89075, 0.98119) |
| | SmolLM2 135M | (0.83181, 0.93169) | (0.90108, 0.97720) | (0.83330, 0.93818) | (0.85126, 0.95050) |
| | SmolLM2 360M | (0.83101, 0.93147) | (0.86717, 0.95648) | (0.84369, 0.94621) | (0.85792, 0.95591) |
| | SmolLM2 1.7B | (0.90914, 0.97384) | (0.77168, 0.89035) | (0.86393, 0.96013) | (0.87505, 0.96865) |

Table 23: Detection performance of Telescope Perplexity, Binoculars, Perplexity, and DetectLLM LRR when transfering a classification threshold from all of the other Ghostbusters datasets to this dataset. We report the F1-Score of each reference model on each test dataset.

| Test Dataset | Reference Model | F1-Score | | | |
| --- | --- | --- | --- | --- | --- |
| | | Telescope | Binoculars | Perplexity | DetectLLM LRR |
| GB Essay ChatGPT | Gemma2 2B | **0.98280** | 0.98179 | **0.98280** | 0.97926 |
| | Gemma2 9B | 0.97825 | 0.97825 | 0.98483 | **0.98837** |
| | Llama3 8B | **0.99090** | 0.86646 | 0.97218 | 0.97926 |
| | Falcon 7B | 0.92828 | **0.97988** | 0.97007 | 0.97059 |
| | SmolLM 135M | **0.93222** | 0.76227 | 0.85635 | 0.87203 |
| | SmolLM 360M | **0.96055** | 0.62822 | 0.88872 | 0.91502 |
| | SmolLM 1.7B | **0.93728** | 0.58371 | 0.92666 | 0.92615 |
| | SmolLM2 135M | **0.97977** | 0.92564 | 0.89479 | 0.89732 |
| | SmolLM2 360M | **0.98938** | 0.90541 | 0.92919 | 0.93981 |
| | SmolLM2 1.7B | **0.99090** | 0.78300 | 0.96813 | 0.96864 |
| GB News ChatGPT | Gemma2 2B | 0.93256 | **0.94263** | 0.81933 | 0.81580 |
| | Gemma2 9B | **0.96779** | 0.95269 | 0.73729 | 0.87066 |
| | Llama3 8B | 0.91394 | **0.94967** | 0.85606 | 0.85959 |
| | Falcon 7B | 0.76181 | **0.99289** | 0.83647 | 0.84104 |
| | SmolLM 135M | 0.95068 | 0.93407 | 0.89532 | **0.95672** |
| | SmolLM 360M | **0.95873** | 0.92854 | 0.86563 | 0.95219 |
| | SmolLM 1.7B | 0.95672 | 0.86814 | 0.83644 | **0.95873** |
| | SmolLM2 135M | 0.92552 | **0.98390** | 0.88274 | 0.92803 |
| | SmolLM2 360M | 0.92246 | **0.98288** | 0.85801 | 0.90836 |
| | SmolLM2 1.7B | 0.94212 | **0.97987** | 0.83442 | 0.88123 |
| GB Creative ChatGPT | Gemma2 2B | **0.96361** | 0.94052 | 0.92862 | 0.83625 |
| | Gemma2 9B | **0.99037** | 0.92596 | 0.91329 | 0.90162 |
| | Llama3 8B | **0.98182** | 0.88503 | 0.92139 | 0.87807 |
| | Falcon 7B | **0.97094** | 0.96471 | 0.88791 | 0.81266 |
| | SmolLM 135M | **0.96146** | 0.84077 | 0.91329 | 0.89959 |
| | SmolLM 360M | **0.97769** | 0.76876 | 0.90923 | 0.92292 |
| | SmolLM 1.7B | **0.98884** | 0.63540 | 0.90923 | 0.94168 |
| | SmolLM2 135M | **0.93306** | 0.91024 | 0.88945 | 0.83824 |
| | SmolLM2 360M | **0.96197** | 0.88742 | 0.90872 | 0.85041 |
| | SmolLM2 1.7B | **0.98073** | 0.85801 | 0.91734 | 0.87170 |
| GB Essay GPT4o Mini | Gemma2 2B | **0.98278** | 0.97822 | 0.98075 | 0.97872 |
| | Gemma2 9B | 0.97822 | 0.96555 | 0.98075 | **0.99037** |
| | Llama3 8B | **0.99088** | 0.95289 | 0.97163 | 0.97720 |
| | Falcon 7B | 0.92827 | **0.97906** | 0.96649 | 0.94817 |
| | SmolLM 135M | **0.92890** | 0.65372 | 0.86833 | 0.86899 |
| | SmolLM 360M | **0.96439** | 0.55415 | 0.89466 | 0.91098 |
| | SmolLM 1.7B | **0.93459** | 0.47280 | 0.92876 | 0.92163 |
| | SmolLM2 135M | **0.97902** | 0.93084 | 0.90079 | 0.89966 |
| | SmolLM2 360M | **0.98902** | 0.87870 | 0.93524 | 0.93853 |
| | SmolLM2 1.7B | **0.99213** | 0.74634 | 0.97075 | 0.96850 |

Table 24: Detection performance of Telescope Perplexity, Binoculars, Perplexity, and DetectLLM LRR when transfering a classification threshold from all of the other Ghostbusters datasets to this dataset. We report the F1-Score of each reference model on each test dataset.

| Test Dataset | Reference Model | F1-Score | | | |
| --- | --- | --- | --- | --- | --- |
| | | Telescope | Binoculars | Perplexity | DetectLLM LRR |
| GB Creative GPT4o Mini | Gemma2 2B | 0.94489 | **0.96576** | 0.94382 | 0.88711 |
| | Gemma2 9B | **0.98876** | 0.96843 | 0.91493 | 0.91867 |
| | Llama3 8B | **0.98074** | 0.94007 | 0.90904 | 0.89674 |
| | Falcon 7B | 0.93664 | **0.96804** | 0.82700 | 0.77906 |
| | SmolLM 135M | **0.94971** | 0.83842 | 0.84430 | 0.85340 |
| | SmolLM 360M | **0.97325** | 0.78384 | 0.86196 | 0.89246 |
| | SmolLM 1.7B | **0.98395** | 0.64205 | 0.85286 | 0.92616 |
| | SmolLM2 135M | 0.92670 | **0.92884** | 0.85126 | 0.81113 |
| | SmolLM2 360M | **0.95827** | 0.91814 | 0.87640 | 0.85393 |
| | SmolLM2 1.7B | **0.97860** | 0.91065 | 0.89299 | 0.88443 |
| GB News Claude | Gemma2 2B | 0.78285 | **0.89017** | 0.81695 | 0.79940 |
| | Gemma2 9B | 0.89719 | **0.93831** | 0.73621 | 0.87513 |
| | Llama3 8B | **0.87061** | 0.85908 | 0.86409 | 0.82698 |
| | Falcon 7B | 0.60933 | 0.81389 | **0.81715** | 0.79707 |
| | SmolLM 135M | 0.56971 | **0.73069** | 0.69157 | 0.63240 |
| | SmolLM 360M | 0.70461 | **0.76680** | 0.74524 | 0.71414 |
| | SmolLM 1.7B | 0.74122 | **0.80090** | 0.76881 | 0.74122 |
| | SmolLM2 135M | 0.59127 | **0.71615** | 0.68806 | 0.64594 |
| | SmolLM2 360M | 0.67252 | **0.76179** | 0.75928 | 0.70612 |
| | SmolLM2 1.7B | 0.69659 | **0.87262** | 0.79890 | 0.72768 |
| GB Creative Claude | Gemma2 2B | **0.91107** | 0.88984 | 0.72865 | 0.76503 |
| | Gemma2 9B | **0.96614** | 0.93380 | 0.75392 | 0.89186 |
| | Llama3 8B | **0.94846** | 0.78626 | 0.77665 | 0.82011 |
| | Falcon 7B | **0.90026** | 0.81410 | 0.75091 | 0.74674 |
| | SmolLM 135M | **0.87974** | 0.75493 | 0.57100 | 0.67155 |
| | SmolLM 360M | **0.94189** | 0.67004 | 0.58717 | 0.73876 |
| | SmolLM 1.7B | **0.91410** | 0.61799 | 0.61294 | 0.77312 |
| | SmolLM2 135M | **0.88580** | 0.79282 | 0.60081 | 0.62405 |
| | SmolLM2 360M | **0.92420** | 0.79788 | 0.63315 | 0.67256 |
| | SmolLM2 1.7B | **0.95856** | 0.87974 | 0.70440 | 0.77514 |
| GB Essay Claude | Gemma2 2B | 0.86950 | 0.80223 | **0.89985** | 0.88720 |
| | Gemma2 9B | **0.97167** | 0.93677 | 0.84623 | 0.97016 |
| | Llama3 8B | 0.93475 | 0.79464 | **0.94638** | 0.92514 |
| | Falcon 7B | 0.86472 | **0.89906** | 0.87201 | 0.86733 |
| | SmolLM 135M | **0.88771** | 0.55943 | 0.78806 | 0.83612 |
| | SmolLM 360M | **0.94234** | 0.52757 | 0.82954 | 0.89479 |
| | SmolLM 1.7B | **0.92564** | 0.56601 | 0.86899 | 0.91705 |
| | SmolLM2 135M | **0.84016** | 0.80425 | 0.81386 | 0.81335 |
| | SmolLM2 360M | **0.92109** | 0.84674 | 0.86191 | 0.87810 |
| | SmolLM2 1.7B | **1.00000** | 0.90909 | 0.81818 | 0.81818 |
| GB Essay Deepseek V3 | Gemma2 2B | 0.98038 | 0.98197 | **0.98250** | 0.98197 |
| | Gemma2 9B | 0.97826 | **0.99311** | 0.98409 | 0.99205 |
| | Llama3 8B | **0.99046** | 0.96182 | 0.97296 | 0.97773 |
| | Falcon 7B | 0.92942 | **0.98046** | 0.96960 | 0.96797 |
| | SmolLM 135M | **0.93372** | 0.92418 | 0.85949 | 0.87487 |
| | SmolLM 360M | **0.95970** | 0.89555 | 0.89130 | 0.91622 |
| | SmolLM 1.7B | **0.93690** | 0.88070 | 0.92789 | 0.92630 |
| | SmolLM2 135M | **0.97932** | 0.96448 | 0.89714 | 0.90138 |
| | SmolLM2 360M | **0.98887** | 0.96607 | 0.93001 | 0.94115 |
| | SmolLM2 1.7B | **0.99046** | 0.95652 | 0.96925 | 0.96766 |
| GB Creative Deepseek V3 | Gemma2 2B | 0.94497 | 0.96980 | **0.98054** | 0.95168 |
| | Gemma2 9B | **0.99396** | 0.97987 | 0.98255 | 0.96174 |
| | Llama3 8B | **0.98926** | 0.94899 | 0.98725 | 0.96174 |
| | Falcon 7B | 0.93973 | **0.97123** | 0.94110 | 0.90342 |
| | SmolLM 135M | 0.94094 | 0.85369 | **0.94631** | 0.87584 |
| | SmolLM 360M | **0.96913** | 0.82282 | 0.96107 | 0.90738 |
| | SmolLM 1.7B | **0.98121** | 0.75369 | 0.96779 | 0.92617 |
| | SmolLM2 135M | 0.92819 | 0.92886 | **0.94362** | 0.87047 |
| | SmolLM2 360M | **0.95772** | 0.92148 | 0.95638 | 0.89799 |
| | SmolLM2 1.7B | **0.98054** | 0.91879 | 0.96913 | 0.93490 |

Table 25: Two standard deviation confidence interval of Telescope Perplexity, Binoculars, Perplexity, and DetectLLM LRR when transfering a classification threshold from all of the other Ghostbusters datasets to this dataset. We report the Confidence Interval of the F1-Score of each reference model on each test dataset.

| Test Dataset | Reference Model | F1-Score | | | |
| --- | --- | --- | --- | --- | --- |
| | | Telescope | Binoculars | Perplexity | DetectLLM LRR |
| GB Essay ChatGPT | Gemma2 2B | (0.97707, 0.98853) | (0.97590, 0.98768) | (0.97707, 0.98853) | (0.97298, 0.98554) |
| | Gemma2 9B | (0.97182, 0.98468) | (0.97182, 0.98468) | (0.97944, 0.99021) | (0.98364, 0.99309) |
| | Llama3 8B | (0.98671, 0.99508) | (0.85147, 0.88146) | (0.96493, 0.97943) | (0.97298, 0.98554) |
| | Falcon 7B | (0.91679, 0.93976) | (0.97362, 0.98613) | (0.96249, 0.97766) | (0.96307, 0.97811) |
| | SmolLM 135M | (0.92114, 0.94330) | (0.74350, 0.78103) | (0.84089, 0.87181) | (0.85730, 0.88675) |
| | SmolLM 360M | (0.95197, 0.96913) | (0.60692, 0.64953) | (0.87486, 0.90258) | (0.90273, 0.92731) |
| | SmolLM 1.7B | (0.92659, 0.94797) | (0.56198, 0.60544) | (0.91516, 0.93815) | (0.91462, 0.93768) |
| | SmolLM2 135M | (0.97356, 0.98597) | (0.91408, 0.93721) | (0.88127, 0.90831) | (0.88394, 0.91070) |
| | SmolLM2 360M | (0.98486, 0.99390) | (0.89251, 0.91831) | (0.91788, 0.94049) | (0.92932, 0.95029) |
| | SmolLM2 1.7B | (0.98671, 0.99508) | (0.76483, 0.80117) | (0.96039, 0.97588) | (0.96096, 0.97632) |
| GB News ChatGPT | Gemma2 2B | (0.92154, 0.94359) | (0.93240, 0.95285) | (0.80241, 0.83624) | (0.79876, 0.83285) |
| | Gemma2 9B | (0.96003, 0.97555) | (0.94336, 0.96203) | (0.71794, 0.75664) | (0.85590, 0.88541) |
| | Llama3 8B | (0.90161, 0.92627) | (0.94006, 0.95929) | (0.84063, 0.87150) | (0.84431, 0.87486) |
| | Falcon 7B | (0.74299, 0.78062) | (0.98918, 0.99660) | (0.82013, 0.85280) | (0.82489, 0.85719) |
| | SmolLM 135M | (0.94116, 0.96020) | (0.92316, 0.94498) | (0.88186, 0.90878) | (0.94777, 0.96567) |
| | SmolLM 360M | (0.94999, 0.96748) | (0.91721, 0.93986) | (0.85063, 0.88062) | (0.94281, 0.96157) |
| | SmolLM 1.7B | (0.94777, 0.96567) | (0.85327, 0.88302) | (0.82017, 0.85270) | (0.94999, 0.96748) |
| | SmolLM2 135M | (0.91397, 0.93706) | (0.97836, 0.98943) | (0.86859, 0.89688) | (0.91667, 0.93940) |
| | SmolLM2 360M | (0.91069, 0.93422) | (0.97718, 0.98859) | (0.84265, 0.87336) | (0.89567, 0.92105) |
| | SmolLM2 1.7B | (0.93186, 0.95239) | (0.97369, 0.98604) | (0.81808, 0.85077) | (0.86700, 0.89545) |
| GB Creative ChatGPT | Gemma2 2B | (0.95390, 0.97332) | (0.92825, 0.95278) | (0.91527, 0.94197) | (0.81706, 0.85544) |
| | Gemma2 9B | (0.98605, 0.99468) | (0.91441, 0.93752) | (0.90087, 0.92571) | (0.88848, 0.91477) |
| | Llama3 8B | (0.97576, 0.98787) | (0.87057, 0.89948) | (0.90919, 0.93359) | (0.86324, 0.89290) |
| | Falcon 7B | (0.96344, 0.97844) | (0.95647, 0.97295) | (0.87382, 0.90199) | (0.79524, 0.83008) |
| | SmolLM 135M | (0.95296, 0.96996) | (0.82462, 0.85692) | (0.90087, 0.92571) | (0.88633, 0.91286) |
| | SmolLM 360M | (0.97117, 0.98421) | (0.75015, 0.78737) | (0.89655, 0.92191) | (0.91115, 0.93469) |
| | SmolLM 1.7B | (0.98421, 0.99348) | (0.61415, 0.65664) | (0.89655, 0.92191) | (0.93134, 0.95203) |
| | SmolLM2 135M | (0.92203, 0.94409) | (0.89763, 0.92286) | (0.87561, 0.90329) | (0.82198, 0.85449) |
| | SmolLM2 360M | (0.95353, 0.97041) | (0.87347, 0.90137) | (0.89601, 0.92143) | (0.83466, 0.86615) |
| | SmolLM2 1.7B | (0.97466, 0.98680) | (0.84261, 0.87342) | (0.90519, 0.92950) | (0.85694, 0.88646) |
| GB Essay GPT4o Mini | Gemma2 2B | (0.97704, 0.98852) | (0.97178, 0.98466) | (0.97469, 0.98681) | (0.97236, 0.98509) |
| | Gemma2 9B | (0.97178, 0.98466) | (0.95751, 0.97360) | (0.97469, 0.98681) | (0.98607, 0.99468) |
| | Llama3 8B | (0.98669, 0.99507) | (0.94354, 0.96223) | (0.96431, 0.97896) | (0.97062, 0.98379) |
| | Falcon 7B | (0.91670, 0.93984) | (0.97264, 0.98548) | (0.95842, 0.97456) | (0.93823, 0.95811) |
| | SmolLM 135M | (0.91598, 0.94182) | (0.62979, 0.67765) | (0.85133, 0.88534) | (0.85202, 0.88596) |
| | SmolLM 360M | (0.95450, 0.97428) | (0.52762, 0.58069) | (0.87827, 0.91105) | (0.89578, 0.92618) |
| | SmolLM 1.7B | (0.92225, 0.94692) | (0.44789, 0.49770) | (0.91593, 0.94159) | (0.90823, 0.93504) |
| | SmolLM2 135M | (0.97234, 0.98571) | (0.91900, 0.94268) | (0.88684, 0.91474) | (0.88564, 0.91368) |
| | SmolLM2 360M | (0.98424, 0.99381) | (0.86371, 0.89370) | (0.92394, 0.94654) | (0.92750, 0.94956) |
| | SmolLM2 1.7B | (0.98802, 0.99623) | (0.72612, 0.76657) | (0.96292, 0.97859) | (0.96039, 0.97662) |

Table 26: Two standard deviation confidence interval of Telescope Perplexity, Binoculars, Perplexity, and DetectLLM LRR when transfering a classification threshold from all of the other Ghostbusters datasets to this dataset. We report the Confidence Interval of the F1-Score of each reference model on each test dataset.

| Test Dataset | Reference Model | F1-Score | | | |
| --- | --- | --- | --- | --- | --- |
| | | Telescope | Binoculars | Perplexity | DetectLLM LRR |
| GB Creative GPT4o Mini | Gemma2 2B | (0.93454, 0.95524) | (0.95751, 0.97400) | (0.93338, 0.95426) | (0.87276, 0.90145) |
| | Gemma2 9B | (0.98399, 0.99354) | (0.96051, 0.97636) | (0.90228, 0.92758) | (0.90628, 0.93107) |
| | Llama3 8B | (0.97451, 0.98697) | (0.92931, 0.95084) | (0.89601, 0.92208) | (0.88294, 0.91053) |
| | Falcon 7B | (0.92543, 0.94785) | (0.95995, 0.97614) | (0.80960, 0.84440) | (0.75998, 0.79815) |
| | SmolLM 135M | (0.93980, 0.95961) | (0.82173, 0.85510) | (0.82786, 0.86074) | (0.83736, 0.86943) |
| | SmolLM 360M | (0.96593, 0.98056) | (0.76518, 0.80250) | (0.84632, 0.87760) | (0.87841, 0.90650) |
| | SmolLM 1.7B | (0.97825, 0.98965) | (0.62032, 0.66379) | (0.83680, 0.86892) | (0.91431, 0.93802) |
| | SmolLM2 135M | (0.91488, 0.93851) | (0.91718, 0.94049) | (0.83513, 0.86739) | (0.79338, 0.82887) |
| | SmolLM2 360M | (0.94920, 0.96733) | (0.90571, 0.93057) | (0.86148, 0.89133) | (0.83792, 0.86994) |
| | SmolLM2 1.7B | (0.97204, 0.98516) | (0.89772, 0.92358) | (0.87898, 0.90701) | (0.86994, 0.89892) |
| GB News Claude | Gemma2 2B | (0.76475, 0.80095) | (0.87645, 0.90389) | (0.79998, 0.83392) | (0.78182, 0.81697) |
| | Gemma2 9B | (0.88386, 0.91052) | (0.92776, 0.94887) | (0.71687, 0.75555) | (0.86062, 0.88964) |
| | Llama3 8B | (0.85588, 0.88534) | (0.84381, 0.87435) | (0.84905, 0.87913) | (0.81038, 0.84358) |
| | Falcon 7B | (0.58706, 0.63161) | (0.79612, 0.83166) | (0.79950, 0.83479) | (0.77871, 0.81543) |
| | SmolLM 135M | (0.54798, 0.59144) | (0.71122, 0.75016) | (0.67130, 0.71185) | (0.61123, 0.65356) |
| | SmolLM 360M | (0.68459, 0.72464) | (0.74824, 0.78536) | (0.72611, 0.76436) | (0.69431, 0.73397) |
| | SmolLM 1.7B | (0.72200, 0.76045) | (0.78338, 0.81843) | (0.75030, 0.78731) | (0.72200, 0.76045) |
| | SmolLM2 135M | (0.56970, 0.61285) | (0.69636, 0.73594) | (0.66773, 0.70840) | (0.62495, 0.66693) |
| | SmolLM2 360M | (0.65192, 0.69312) | (0.74309, 0.78048) | (0.74051, 0.77804) | (0.68612, 0.72611) |
| | SmolLM2 1.7B | (0.67641, 0.71677) | (0.85798, 0.88725) | (0.78130, 0.81649) | (0.70814, 0.74722) |
| GB Creative Claude | Gemma2 2B | (0.89853, 0.92361) | (0.87605, 0.90364) | (0.70906, 0.74824) | (0.74635, 0.78371) |
| | Gemma2 9B | (0.95818, 0.97411) | (0.92285, 0.94476) | (0.73494, 0.77289) | (0.87818, 0.90555) |
| | Llama3 8B | (0.93872, 0.95820) | (0.76819, 0.80432) | (0.75831, 0.79500) | (0.80319, 0.83703) |
| | Falcon 7B | (0.88684, 0.91368) | (0.79668, 0.83152) | (0.73154, 0.77028) | (0.72726, 0.76621) |
| | SmolLM 135M | (0.86541, 0.89407) | (0.73598, 0.77388) | (0.54919, 0.59280) | (0.65086, 0.69224) |
| | SmolLM 360M | (0.93158, 0.95220) | (0.64932, 0.69075) | (0.56547, 0.60886) | (0.71940, 0.75811) |
| | SmolLM 1.7B | (0.90175, 0.92644) | (0.59658, 0.63940) | (0.59148, 0.63440) | (0.75467, 0.79157) |
| | SmolLM2 135M | (0.87179, 0.89981) | (0.77497, 0.81068) | (0.57923, 0.62239) | (0.60271, 0.64539) |
| | SmolLM2 360M | (0.91254, 0.93587) | (0.78018, 0.81557) | (0.61191, 0.65438) | (0.65189, 0.69324) |
| | SmolLM2 1.7B | (0.94978, 0.96735) | (0.86541, 0.89407) | (0.68429, 0.72450) | (0.75675, 0.79353) |
| GB Essay Claude | Gemma2 2B | (0.85465, 0.88435) | (0.78467, 0.81978) | (0.88662, 0.91308) | (0.87326, 0.90115) |
| | Gemma2 9B | (0.96436, 0.97899) | (0.92605, 0.94750) | (0.83033, 0.86213) | (0.96266, 0.97766) |
| | Llama3 8B | (0.92386, 0.94564) | (0.77683, 0.81245) | (0.93645, 0.95631) | (0.91354, 0.93674) |
| | Falcon 7B | (0.84943, 0.88001) | (0.88560, 0.91253) | (0.85707, 0.88694) | (0.85216, 0.88249) |
| | SmolLM 135M | (0.87379, 0.90163) | (0.53755, 0.58132) | (0.77005, 0.80608) | (0.81980, 0.85243) |
| | SmolLM 360M | (0.93206, 0.95261) | (0.50556, 0.54957) | (0.81296, 0.84612) | (0.88127, 0.90831) |
| | SmolLM 1.7B | (0.91408, 0.93721) | (0.54416, 0.58786) | (0.85412, 0.88387) | (0.90489, 0.92920) |
| | SmolLM2 135M | (0.82401, 0.85632) | (0.78676, 0.82174) | (0.79670, 0.83102) | (0.79618, 0.83053) |
| | SmolLM2 360M | (0.90921, 0.93298) | (0.83086, 0.86262) | (0.84670, 0.87712) | (0.86368, 0.89252) |
| | SmolLM2 1.7B | (1.00000, 1.00000) | (0.73920, 1.07898) | (0.59026, 1.04611) | (0.59026, 1.04611) |
| GB Essay Deepseek V3 | Gemma2 2B | (0.97412, 0.98664) | (0.97597, 0.98798) | (0.97659, 0.98842) | (0.97597, 0.98798) |
| | Gemma2 9B | (0.97168, 0.98484) | (0.98937, 0.99684) | (0.97845, 0.98974) | (0.98804, 0.99606) |
| | Llama3 8B | (0.98607, 0.99484) | (0.95318, 0.97047) | (0.96564, 0.98028) | (0.97107, 0.98439) |
| | Falcon 7B | (0.91773, 0.94112) | (0.97413, 0.98678) | (0.96176, 0.97744) | (0.95993, 0.97601) |
| | SmolLM 135M | (0.92249, 0.94495) | (0.91223, 0.93612) | (0.84381, 0.87517) | (0.85993, 0.88980) |
| | SmolLM 360M | (0.95083, 0.96858) | (0.88174, 0.90935) | (0.87726, 0.90535) | (0.90372, 0.92873) |
| | SmolLM 1.7B | (0.92593, 0.94788) | (0.86607, 0.89533) | (0.91622, 0.93956) | (0.91451, 0.93809) |
| | SmolLM2 135M | (0.97290, 0.98574) | (0.95612, 0.97283) | (0.88343, 0.91085) | (0.88792, 0.91483) |
| | SmolLM2 360M | (0.98413, 0.99360) | (0.95789, 0.97424) | (0.91850, 0.94152) | (0.93052, 0.95177) |
| | SmolLM2 1.7B | (0.98607, 0.99484) | (0.94732, 0.96573) | (0.96146, 0.97704) | (0.95967, 0.97564) |
| GB Creative Deepseek V3 | Gemma2 2B | (0.93339, 0.95655) | (0.96111, 0.97849) | (0.97352, 0.98755) | (0.94079, 0.96257) |
| | Gemma2 9B | (0.99003, 0.99789) | (0.97273, 0.98700) | (0.97590, 0.98920) | (0.95201, 0.97148) |
| | Llama3 8B | (0.98403, 0.99450) | (0.93782, 0.96016) | (0.98155, 0.99295) | (0.95201, 0.97148) |
| | Falcon 7B | (0.92752, 0.95193) | (0.96266, 0.97981) | (0.92902, 0.95317) | (0.88827, 0.91858) |
| | SmolLM 135M | (0.92897, 0.95291) | (0.83575, 0.87164) | (0.93486, 0.95775) | (0.85909, 0.89258) |
| | SmolLM 360M | (0.96034, 0.97791) | (0.80343, 0.84221) | (0.95125, 0.97089) | (0.89266, 0.92210) |
| | SmolLM 1.7B | (0.97431, 0.98810) | (0.73181, 0.77557) | (0.95882, 0.97675) | (0.91290, 0.93945) |
| | SmolLM2 135M | (0.91508, 0.94130) | (0.91581, 0.94191) | (0.93191, 0.95534) | (0.85342, 0.88752) |
| | SmolLM2 360M | (0.94750, 0.96794) | (0.90782, 0.93513) | (0.94600, 0.96675) | (0.88262, 0.91335) |
| | SmolLM2 1.7B | (0.97352, 0.98755) | (0.90492, 0.93266) | (0.96034, 0.97791) | (0.92237, 0.94743) |

