# OpenReview forum: "Telescope: Improving Zero Shot Detection of LLM Generated Content With Token Repetition"
_ICLR.cc/2026/Conference — Submitted to ICLR 2026_

### Official Review · Reviewer_qj1u · 2025-10-29

**Soundness:** 2
**Presentation:** 1
**Contribution:** 2
**Rating:** 2
**Confidence:** 4

**Summary:**

This paper investigates the detection of content generated by large language models. The authors propose a new metric called Telescope Perplexity (TPPL), designed to capture a vestigial heuristic, an aversion to token repetition formed and retained during the early training stages of LLMs. The study verifies that this “vestigial heuristic” emerges early and persists across training in several open‑source models (e.g., Pythia, Amber‑7B). It further demonstrates that this signal can effectively distinguish human‑written from machine‑generated text across different architectures, corpora, and model outputs.

**Strengths:**

- Proposes the Vestigial Heuristic hypothesis, tracing the detectability of LLM‑generated text back to early‑stage training biases, and provides empirical validation.
- Includes ablation studies such as adversarial perturbation and transfer tests.

**Weaknesses:**

- The Related Work section provides a biased and incomplete review of detector progress, omitting many significant works (e.g., Fast‑DetectGPT [1]). It also lacks systematic categorization and hierarchical structure. The authors should consult survey papers for a more comprehensive and organized discussion.
- In Section 4.1.1, the authors claim that a major challenge is the lack of datasets generated by more advanced models in the literature—but which specific literature? To my knowledge, many cutting‑edge datasets (e.g., DetectRL [2]) already rely entirely on commercial models. Why not use such datasets instead? Judging from Table 2, both DetectLLM and PPL already achieve strong performance on most datasets, which indirectly suggests that the datasets used here may be relatively easy and requires more stressful testing.
- The method shows a noticeable performance drop (AUROC ≈ 0.88) on structured text such as news. When thresholds are transferred (transfer test), the F‑measure declines, indicating a need for recalibration across domains. Therefore, a systematic cross‑domain transfer comparison is necessary to validate the robustness of the method. However, the paper does not appear to include any corresponding tables, figures, or comparisons with baselines.
- The ICLR paper template used appears to be incorrectly formatted, and the writing is quite disorganized, making it difficult to extract concrete details.

[1] Fast-DetectGPT: Efficient Zero-Shot Detection of Machine-Generated Text via Conditional Probability Curvature, ICLR 2024

[2] DetectRL: Benchmarking LLM-Generated Text Detection in Real-World Scenarios, NeurlPS 2024

**Questions:**

See Weaknesses.

---

> ### Author Response · Authors · 2025-11-20
>
> _"The Related Work section provides a biased and incomplete review of detector progress, omitting many significant works (e.g., Fast‑DetectGPT [1]). It also lacks systematic categorization and hierarchical structure. The authors should consult survey papers for a more comprehensive and organized discussion."_
>
> **Response:**
>
> See Y. Li, Q. Li, L. Cui, W. Bi, Z. Wang, L. Wang, L. Yang, S. Shi, and Y. Zhang. Mage: Machine generated text detection in the wild, 2024. URL https://arxiv.org/abs/2305.13242, or https://arxiv.org/pdf/2305.15047, for why detectGPT and its variants where not included, we can provide results on a subset of benchmarks. We initially had a wider systematic categorization of many more LLM text detectors but had to cut it out due to needing to meet the strict page limits.
>
> ---
>
> _"In Section 4.1.1, the authors claim that a major challenge is the lack of datasets generated by more advanced models in the literature—but which specific literature? To my knowledge, many cutting‑edge datasets (e.g., DetectRL [2]) already rely entirely on commercial models. Why not use such datasets instead? Judging from Table 2, both DetectLLM and PPL already achieve strong performance on most datasets, which indirectly suggests that the datasets used here may be relatively easy and requires more stressful testing."_
>
> **Response:**
>
> DetectRL's many of the same base data sources as HC3 but is very out of date with the most sophisticated model used being GPT-3.5, we did an extensive review of the literature and it still is missing modern LLM generated data, given the rapid pace of change it is likely to be a persistent gap.
>
> ---
>
> _"The method shows a noticeable performance drop (AUROC ≈ 0.88) on structured text such as news. When thresholds are transferred (transfer test), the F‑measure declines, indicating a need for recalibration across domains. Therefore, a systematic cross‑domain transfer comparison is necessary to validate the robustness of the method. However, the paper does not appear to include any corresponding tables, figures, or comparisons with baselines."_
>
> **Response:**
>
> Comprehensive results for transferability can be found in tables 22-25 at the end of the appendix the discussion of which is in 4.3 and 5.4.
>
> ---
>
> _"The ICLR paper template used appears to be incorrectly formatted, and the writing is quite disorganized, making it difficult to extract concrete details."_
>
> **Response:**
>
> We apologize for the incorrect formatting and will correct it. Thank you for bringing this to our attention.
>
> ---
>
> _"Drastically different sampling temperatures are used in the generation of ESL dataset and Ghostbuster replication with GPT-4o mini and Deepseek. Why?"_
>
> **Response:**
>
> These are the default parameters for each model. Given both were selected as a result of being easily accessible via a free chat UI, we use the same parameters.

---

> > ### Comment · Reviewer_qj1u · 2025-11-26
> > **Thanks for your responses**
> >
> > Thank you for the author's reply. I would like to maintain my rating at this stage and look forward to discussing with other reviewers in an open mind.

---

### Official Review · Reviewer_hSV2 · 2025-10-30

**Soundness:** 2
**Presentation:** 2
**Contribution:** 2
**Rating:** 2
**Confidence:** 3

**Summary:**

The paper introduces Telescope Perplexity, a perplexity-style metric that measures the likelihood of repeating the last token in the context window. The authors suggest using this metric to differentiate LLM-generated from human-generated text in a zero-shot setting. The intuition is that LLMs learn early in pre-training to strongly avoid repeating tokens and that this persists throughout the training stages. The authors show empirical results across a range of datasets, including newly generated datasets using recent LLMs. The method is compared to other zero-shot approaches and claims superior performance.

**Strengths:**

- The paper tackles an important problem, differentiating LLM- from human-generated text, using a simple but very interesting approach, focusing on token repetitions.
- The hypothesis and the proposed metric are clearly described.
- The observation that LLMs learn and retain a strong aversion to token repetition early in training appears novel and interesting; however, I am concerned whether the large perplexity values throughout the process are strong evidence to support this claim.
- Extensive empirical evaluation, covering multiple datasets, both legacy and newly generated text, and multiple reference models.

**Weaknesses:**

- **Choice of benchmark and metric**: The reported AUROCs for the proposed method and the existing baselines (including standard perplexity and Binoculars) seem extremely high, close to 0.99 for most datasets. This would either imply that the detection of AI-generated text is a solved problem or that the setup is not realistic.
- **Unclear conceptual link between aversion to repeat tokens and usefulness for detection**: The authors argue that LLMs develop a persistent aversion to token repetition, which leads to high Telescope Perplexity for both human- and LLM-generated text. It remains unclear to me why a reference LLM would assign higher Telescope Perplexity specifically to LLM-generated text? Could the effect instead be driven by confounding variable(s) (e.g., typos, grammatical errors, LLM’s overusage of certain words)?
- **Insufficient details for the newly generated datasets**. Section 4.1.1 does not make it explicit that Claude is also used to generate datasets. I understand that the authors have used models (GPT4o and Deepseek) to generate new datasets based on each of the existing ones (GB Essay, GB News, and GB Creative), however, the dataset of GB News generated with GPT4o mini is possibly missing. A discussion about why it is better to ask the model to rewrite the human-generated text instead of giving it the same question or general topic as the human had when they were writing the text, and if it introduces possible biases in the analysis, would be appreciated.
- **Some claims lack citations**. For example, the authors claim that rewriting is a difficult environment for AI detection models, and that if a reference LLM is perplexed and has never seen anything like it before, that means that other language models likely haven’t seen anything like it in their training data. The implication that all language models have largely overlapping training data holds across present and future models is non-obvious to me.
- **Insufficient details for the “AI humanizer” method**. The “AI Humanizer” has not been released and has not been reproducibly described. Additionally, the paper does not discuss robustness to intentional fine-tuning of the target LLM to minimize Telescope Perplexity (e.g., via DPO/RLHF), which seems a direct threat model that needs to be addressed.
- **Minor**:
    - Appendix organization and writing quality could be improved.
    - Some typos remain (e.g., “behviors” in Section 3.2 and “... calibrated 8.” in Section 9.5)
    - Table 10 mentions bold entries but contains none.
    - Table 14 formatting (double lines) is inconsistent with other tables.

**Questions:**

(also see weaknesses)
- Why is the mean AUROC across reference models (Table 2) chosen over the maximum? If the goal is detecting LLM- versus human-generated text, wouldn’t the best-performing reference model provide more actionable insight?
- What is the motivation for using a logistic regression to determine the threshold for Transferability F1-Score?
- What’s the impact of typos and grammatical errors in the input text?
- Why does Binoculars Score have statistically significantly better performance than Telescope Perplexity in other languages (Table 12)? Could this difference be linked to linguistic properties or method-specific biases?
- While I agree that the omission of exponentiation in Telescope Perplexity does not affect detection performance since the exponential function is monotonic, I am concerned about the large Telescope Perplexity values. Given that the **non-exponentiated** log-likelihood starts around 10 at the beginning of training (Fig. 2) and increases to roughly 12, could these large values be primarily due to noise?
- It looks like the competitor methods show interesting, but opposite trends in performance with respect to text length: for the Detect LLM Text dataset, performance drops sharply around 300 tokens before improving with longer inputs, whereas for the HC3 Plus dataset, performance sharply increases around 200 tokens and then declines (Fig. 2). Could you share any insights into what drives these sharp inflection points, and why the trends differ between the two datasets?

---

> ### Author Response · Authors · 2025-11-20
>
> ---
>
> _"Choice of benchmark and metric ..."_
>
> **Response:**
>
> In finance an AUC of .9 would be considered unbelievably good. For AI detection or plagiarism use of a model with an AUC of .9 could do more harm than good. AUC scores can have different scores that are considered "good" in different circumstances. In the case of LLM text detection, to deploy effectively, you want as high of a score as possible, since being wrong may lead to major consequences.
>
> Averaging across datasets we see Telescope 0.97385 Binoculars 0.90563 Perplexity 0.94108 LRR 0.92667. This increase in performance over existing methods is (in addition to being statistically significant as shown by our 95% confidence intervals) significant for AI detection use cases and may allow for deployment in real world settings.
>
> ---
>
> _"Unclear conceptual link between aversion to repeat tokens and usefulness for detection ... "_
>
> **Response:**
>
> It is a learned behavior during the training process. As for alternate signals we present an ablation with character level swaps creating typo's and grammatical errors and observe no substantive change in performance (see Appendix Section 9.16) we know this is a local effect from section 3.2. Telescope perplexity cannot be properly calculated on single tokens so single word sequences are not the culprit.
>
> ---
>
> _"Insufficient details for the newly generated datasets. ..."_
>
> **Response:**
>
> The Claude GB dataset belongs to the original ghostbusters paper and is not one of our contributions. For the Ghostbusters dataset, the news portion does not contain the required information to generate datasets with new target models, unfortunately.
>
> ---
>
> _"A discussion about why it is better to ask the model to rewrite the human-generated text ...would be appreciated."_
>
> **Response:**
>
> These are two separate tasks, there is a number of evaluations on purely generative tasks that are done as you described, we do include in addition to these a re-writing task as re-writing is a common use case for LLMs.
>
> ---
>
> _"Some claims lack citations..."_
>
> **Response:**
>
> We will include citations for each of these claims.
>
> ---
>
> _"Insufficient details for the "AI humanizer" method. ..."_
>
> **Response:**
>
> This is a tricky subject as our goal is for AI usage to be more transparent not less, contributing to workarounds for other methods that are in use by commercial detectors, for our team, seems more likely to increase harm than reduce it. We are more than happy to include more details where we know such details have already been published.
>
> ---
>
> _"Additionally, the paper does not discuss robustness to intentional fine-tuning ..."_
>
> **Response:**
>
> RL fine tuning is an attack vector that we are aware exists but given its substantial cost and that it is not a form of attack that is currently in use we will leave this as future work. No other LLM detection work considers this either for this reason.
>
> ---
>
> _"Minor: Appendix organization and writing quality could be improved. ..."_
>
> **Response:**
>
> We appreciate the thoroughness of feedback, and will incorporate it in the final paper.
>
> ---
>
> _"... wouldn't the best-performing reference model provide more actionable insight?"_
>
> **Response:**
>
> One of our goals with table 2 is to show robustness to the reference model which is a useful property our technique has. All of the results are available in the appendices, but we would also be happy to show the best performing model in a revision.
>
> ---
>
> _"What is the motivation for using a logistic regression ... ?"_
>
> **Response:**
>
> Logistic regression in this context is a simple way to regress the optimal threshold for a given dataset, we then test that threshold against each other dataset to produce the F1's shown. Logistic regression over a single input variable is equivalent to choosing a classification threshold.
>
> ---
>
> _"What's the impact of typos and grammatical errors in the input text?"_
>
> **Response:**
>
> We present an ablation with character level swaps creating typo's and grammatical errors and observe no substantive change in performance (see Appendix Section 9.16)
>
> ---
>
> _"Why does Binoculars Score ... better performance than Telescope Perplexity in other languages (Table 12)? ..."_
>
> **Response:**
>
> This requires further investigation but the specifics of our method may mean the specific vestigial heuristic we present does not apply well to other languages.
>
> ---
>
> _"... could these large values be primarily due to noise?"_
>
> **Response:**
>
> These are exponentiated values and we will make this more clear in the final paper.
>
> ---
> ... Could you share any insights into what drives these sharp inflection points, and why the trends differ between the two datasets?"_
>
> **Response:**
>
> This is likely because of the overall performance gap between the two datasets for Binoculars, the trend is similar between both, with a sharp increase and then a decline, but for the easier dataset the increase is over more sample lengths and the decrease is less severe.
>
> ---

---

> > ### Comment · Reviewer_hSV2 · 2025-11-26
> >
> > I would like to thank the authors for their rebuttal.
> >
> > > Choice of benchmark and metric ...
> >
> > I am still not convinced if AUC scores of up to 0.999 for some datasets are representative of a realistic setup in practice.
> >
> > > Unclear conceptual link between aversion to repeat tokens and usefulness for detection …
> >
> > The link between aversion to repeat tokens and detection of LLM-generated text remains unclear to me.
> >
> > > Insufficient details for the "AI humanizer" method
> >
> > Given your response, it remains unclear to me whether the details of the “AI humanizer” method will ever be disclosed, even post-publication. If the method indeed leads to workarounds for commercial detectors, have you notified the developers of those systems? In general, revealing such vulnerabilities typically strengthens, rather than weakens, the ecosystem by enabling more robust future detectors.
> >
> > I believe the method should be described in sufficient details for the community to evaluate it. Security through obscurity is unlikely to be a practically effective or scientifically sound method. In fact, withholding details may keep the developers and the community in the dark while motivated adversaries independently discover the method and exploit the same weakness.
> >
> > > Additionally, the paper does not discuss robustness to intentional fine-tuning
> >
> > RL is not the only possible fine-tuning approach. What about other methods?
> >
> > > Could you share any insights into what drives these sharp inflection points…
> >
> > What drives these inflection points remains unclear to me.
> >
> > I will maintain my current score for now.

---

> > > ### Author Response · Authors · 2025-11-27
> > >
> > > Thank you for taking the time to respond.
> > >
> > > *"I am still not convinced if AUC scores of up to 0.999 for some datasets are representative of a realistic setup in practice. Do you have a specific concern with any of the datasets or experiment setup?"*
> > >
> > > Good performance is not necessarily indicative of an issue in the data setup. Especially on datasets where other methods perform poorly, this is one of this papers major contributions.
> > >
> > > ---
> > > *"The link between aversion to repeat tokens and detection of LLM-generated text remains unclear to me."*
> > >
> > > The simple version is that this vestigial heuristic is triggered more strongly by LLM text than human-written text.
> > >
> > > ---
> > > *"Given your response, it remains unclear to me whether the details of the 'AI humanizer' method will ever be disclosed, even post-publication ..."*
> > >
> > > This is a pretty compelling argument. Our original intent was not to release any "AI humanizer" code post-publication. While we still have reservations about potentially contributing to the degradation of existing systems, we will include the code and an in-depth appendix section discussing the "AI humanizer" design in the final paper.
> > >
> > > ---
> > > *"Fine tuning for robustness—RL is not the only possible fine-tuning approach. What about other methods?"*
> > >
> > > Fine-tuning for robustness to a detection method is something that no other work in the field of LLM detection does because it is far too expensive to consider for strong open-source target language models. Not to mention the most important models are closed source, which means that attempting to fine-tune them to test whether or not it could degrade performance is impossible. Attempting to track the performance of LLM detection techniques when fine-tuned using different techniques across different target models represents its own major contribution to the field of LLM text detection.
> > >
> > > ---
> > > *"Could you share any insights into what drives these sharp inflection points?"*
> > >
> > > It may be useful to know that these inflection points coincided with a large jump in the number of samples in each corpus.

---

### Official Review · Reviewer_LrEe · 2025-10-31

**Soundness:** 2
**Presentation:** 2
**Contribution:** 2
**Rating:** 2
**Confidence:** 4

**Summary:**

This paper introduces the “Vestigial Heuristic” hypothesis, proposing that large language models (LLMs) develop an early-stage bias against token repetition while learning bigram patterns, and that this bias persists throughout training. The authors argue this residual heuristic explains why LLMs strongly avoid repetition even when unnecessary. To test this, they propose Telescope Perplexity, a novel metric that measures a model’s likelihood of repeating tokens, providing a focused probe into this aversion. Extensive experiments show Telescope Perplexity effectively distinguishes human-written from LLM-generated text, achieving strong zero-shot detection performance and robustness across modern models and datasets.

**Strengths:**

1) The paper introduces a creative and conceptually fresh hypothesis — the “Vestigial Heuristics” perspective — suggesting that aversions to token repetition arise early during LLM training as a byproduct of bigram modeling and persist into later training stages. However, the empirical evidence is shallow and non-deterministic, leaving the hypothesis weakly supported.
2) The authors provide reasonably careful empirical validation. The experiments test multiple modern LLMs (e.g., GPT-4o Mini, DeepSeek-V3), showing some robustness and competitiveness in zero-shot detection tasks.

**Weaknesses:**

1) The paper introduces Telescope Perplexity as conceptually distinct from standard perplexity, yet the difference (conditioning on the same token rather than the next) seems mathematically minor and somewhat arbitrary. The rationale for why this formulation specifically isolates the repetition heuristic is not rigorously established. Additionally, the calculation of M(s_i|s_{1:i}) is not provided. Given that the scoring models are trained to predict next token, the calculation of the Telescope probability is not obvious.
2) Although the paper reports “extensive testing,” the experiments focus narrowly on zero-shot text detection performance. For example, only three baselines are presented, data are generated by narrow decoding strategies and settings, scenarios under paraphrasing and adversarial attacks are not considered.
3) The review about related work is incomplete given that many perplexity based zero-shot detectors are not compared. For example, DetectGPT, Fast-DetectGPT, LRR, Glimpse, etc.
4) The exceptionally high scores of the simple perplexity baseline suggest that there maybe some distribution bias in the data settings. It would be valuable addition to have some datasets representing paraphrasing and adversarial attacks.

**Questions:**

LN016: What is the indelible mark? If the concept was not well presented, the hypothesis will be vague.

LN055: What is the point of the Vestigial Heuristic? How does it relate to your metric design?

LN100: Hard to understand the formula M(s_i|s_{1:i}). How is the probability calculated using the model M?

LN156: There are plenty of zero-shot detectors using perplexity. For example, DetectGPT, Fast-DetectGPT, LRR, Glimpse, etc.

LN192: Except the raise during the initial stage, no significant patterns from the lines. What is the connect between these results and your heuristics? Do they justify the hypothesis?

LN270: Data from LRMs like gpt-o1 or deepseek-r1 would be valuable addition.

LN280: The settings of temperature with 0.7 and top-p with 0.9 are narrow, which make the text distribution skewed and easy for detection. A wide spectrum of the decoding strategies and settings are expected to simulate real scenarios.

LN338: Which size of SmolLM is used here? I would expect to see the typical setting in the main results, such as falcon7b used by Binoculars or gpt-neo-2.7B by Fast-DetectGPT.

LN344: The baselines are insufficient to support the claim. Additionally, the results seem weird that the perplexity performs better than Binoculars. Is there any deeper reason?

---

> ### Author Response · Authors · 2025-11-20
>
> COMMENT:
> The paper introduces Telescope Perplexity as conceptually distinct from standard perplexity, yet the difference (conditioning on the same token rather than the next) seems mathematically minor and somewhat arbitrary. The rationale for why this formulation specifically isolates the repetition heuristic is not rigorously established.
>
> RESPONSE
> The notation is fairly standard when discussing conditional generation models in machine learning and statistics where ‘s’ represents some ground truth sequence of text that we want the model to generate. M(s_i | s_{1:i-1}) would refer to the probability that the model (in this case a large language model) outputs the i’th token in that sequence while all of the tokens before the i’th token in the sequence are fed into its context window. This is how normal conditional generation for LLM inference works, and all it is saying is that we would like to find the probability that the LLM output is the same as our ground truth sequence. All that we are saying by M(s_i | s_{1:i}) is that instead of feeding all of the tokens before the i’th token into the context window, we instead feed all of the tokens before the i’th token and the i’th token into the context window and then we are trying to find the probability of the i’th token being generated again by the reference model. To compute this, we simply feed a sequence of tokens (s_{1:i}) into the model and then compute the probability that it matches the last token in the sequence from the logits distribution.
>
>
>
> COMMENT:
> Additionally, the calculation of M(s_i|s_{1:i}) is not provided. Given that the scoring models are trained to predict next token, the calculation of the Telescope probability is not obvious.
> Although the paper reports “extensive testing,” the experiments focus narrowly on zero-shot text detection performance. For example, only three baselines are presented, data are generated by narrow decoding strategies and settings, scenarios under paraphrasing and adversarial attacks are not considered.
>
> RESPONSE:
> We consider the primary baselines in the space of zero shot LLM text detection. We also do tests under paraphrasing and adversarial attacks as shown in sections 5 and 9.19. We test on a variety of different decoding strategies from a variety of settings and a variety of target models, such as increasing/ decreasing the temperature in generation and evaluating the efficacy of this detector on certain types of prompt attacks in section 9.15.
>
>
> COMMENT:
> The review about related work is incomplete given that many perplexity based zero-shot detectors are not compared. For example, DetectGPT, Fast-DetectGPT, LRR, Glimpse, etc.
>
> RESPONSE:
> LRR is one of the primary zero shot detectors that we use as a baseline.
>
> We needed to keep the related works section as short as it needed to be or else we wouldn’t have enough space in the main paper to finish discussing our technique and our primary results. The related works section used to be much bigger and contained information about a variety of other detectors, but unfortunately we were forced to cut it down only to detectors that we actively benchmark against to meet the page limit.
>
>
> We discuss in section 6 (Limitations) why we did not benchmark against DetectGPT due to its insane computational requirements that we could not hope to benchmark for the number of reference models and datasets that we test on.
>
> While Fast-DetectGPT is more reasonable in its computational requirements, it still requires us to know the target model to properly use it, but the issue is we don’t know what the set of possible target models is for all of our datasets, and even if we did, in order to compute the Fast-DetectGPT score on all of the potential target models, we would need to inference the language model 4*N times for all of the potential target models. We have so many datasets and test on so many reference models, that this very quickly becomes computationally infeasible.
>
> Glimpse is a technique that enhances black box techniques and turns them into white box techniques. Since this work primarily focuses on zero shot black box techniques where the potential target model is not known ahead of time, we don’t benchmark against glimpse.
>
> There are many other LLM detection techniques that are generated by our code that we don’t include in our results such as entropy because it would cause the already bloated results section to be even harder to navigate, especially since a lot of these techniques don’t perform any better than the ones in the paper.

---

> ### Author Response · Authors · 2025-11-20
>
> COMMENT:
> The exceptionally high scores of the simple perplexity baseline suggest that there maybe some distribution bias in the data settings.
>
>
> RESPONSE:
> We test on a variety of reputable datasets/ target models across a variety of tasks such as rewriting text, news writing, creative writing, and essay writing. If you believe that there is a specific bias in the data distribution and that we are not covering a specific important type of data, please let us know. Yes, perplexity does generally perform very well but only when averaged across reference models. Binoculars for instance far outperforms perplexity in most datasets while using the Falcon 7B reference model, but the chart in the main text of the paper only takes into account an average of all of the reference models tested. Trying to put the enormous amount of data in section 9.19 into the main text is a losing battle. See appendix for full results.
>
> In the main text we attempt to point the reader to those charts because the amount of data that we collected can’t really be condensed into a more compact format without losing information unfortunately.
>
>
> COMMENT:
> It would be valuable addition to have some datasets representing paraphrasing and adversarial attacks.
>
> RESPONSE:
> A very astute point, these are included in 5.2 and 4.1.3 with further analysis and discussed in the appendix.
>
>
>
> Questions:
> LN016: What is the indelible mark? If the concept was not well presented, the hypothesis will be vague.
>
> Please see our answer to your question about line 55, we hope that it answers this question.
>
> LN055: What is the point of the Vestigial Heuristic? How does it relate to your metric design?
>
> The vestigial heuristic is the hypothesis that early on in training, LLMs learn a heuristic to avoid token repetition. Because the model starts off approximating a bigram model at the beginning of training, this is the best it can do, but we argue that this heuristic stays throughout training, and we are able to measure it through measuring the probability of repetition through a reference model. Since this “heuristic” stays throughout training even though it's not strictly necessary, we argue that it is vestigial and indelible.
>
>
> LN100: Hard to understand the formula M(s_i|s_{1:i}). How is the probability calculated using the model M?
>
> Please see our answer to your comments above.
>
>
> LN156: There are plenty of zero-shot detectors using perplexity. For example, DetectGPT, Fast-DetectGPT, LRR, Glimpse, etc.
>
> Could you clarify what question you had regarding this?
>
> LN192: Except the raise during the initial stage, no significant patterns from the lines. What is the connect between these results and your heuristics? Do they justify the hypothesis?
>
> Please see our answer to your question for line 55
>
> LN270: Data from LRMs like gpt-o1 or deepseek-r1 would be valuable addition.
>
> We would agree, but generation using them is significantly more expensive, unfortunately and there aren’t any datasets currently that contain data from these target models.
>
> LN280: The settings of temperature with 0.7 and top-p with 0.9 are narrow, which make the text distribution skewed and easy for detection. A wide spectrum of the decoding strategies and settings are expected to simulate real scenarios.
>
> Please see our answer to your comments above.
>
> LN338: Which size of SmolLM is used here? I would expect to see the typical setting in the main results, such as falcon7b used by Binoculars or gpt-neo-2.7B by Fast-DetectGPT.
>
> We test over every size of SmolLM (135M, 360M, 1.7B) and we also test over every size of SmolLM2 (135M, 360M, 1.7B). If you would like to see specific results for each of the sizes and reference models that we test on, please see section 9.19.
>
>
> LN344: The baselines are insufficient to support the claim. Additionally, the results seem weird that the perplexity performs better than Binoculars. Is there any deeper reason?
>
> Please see our answer to your comments above.

---

> > ### Comment · Reviewer_LrEe · 2025-11-26
> >
> > Thank you for the author's response. I would like to keep my current rating at this stage and am looking forward to engaging in further discussions with other reviewers.

---

### Official Review · Reviewer_EUSN · 2025-11-01

**Soundness:** 1
**Presentation:** 1
**Contribution:** 2
**Rating:** 0
**Confidence:** 4

**Summary:**

The authors claim to have developed a new LLM detection method, Telescope, based on the higher token repetition aversion in LLM-generated texts compared to human-generated texts. The authors test their methods on basic LLM-generated texts against several existing LLM-detection methods, such as Binoculars, DetectLLM, and Perplexity, using the AURoC metric to argue their method's SotA performance. The authors also claim that their method exhibits excellent properties in real-world detection tasks, such as locality and generalization.

**Strengths:**

The authors introduce a new and useful signal for LLM text detection - token repetition aversion, which, to the best of my knowledge, has not been previously documented.The authors introduce a new useful signal for LLM text detection - token repetition aversion, that to the best of my knowledge, has not been documented before.

**Weaknesses:**

- The authors do not seem to be aware, or at least mention, the existence of a generation control parameter called "repeat penalty", applied by default to models upon generation. This parameter penalizes token repetition to prevent model degeneration and can be easily adjusted to avoid detection by the authors' method. Without evaluating their method's robustness against this metric, the detection method's relevance or performance cannot be claimed, especially as sampling methods that automatically optimize repetition penalties are emerging [1]. Moreover, the "Vestigal Heuristic" hypothesis cannot be claimed without accounting for this parameter.

- The authors only evaluated their method's resilience against a single evasion attack, which is insufficient. Evasion attacks are the most problematic current issue in the context of LLM detection, as exemplified by the existing literature on LLM detectors benchmarking, notably the RAID paper [2]. As such, the performance of their method in a real-world setting cannot be evaluated.

- The authors do not seem to be aware of the foundational results in LLM detection, demonstrating that AUROC is not an informative metric with LLM detection [3]. In fact, due to the lack of access to true labels for LLM detector operators, the relevant metric is TPR at a fixed low FPR, for which most recent detectors have been optimized, potentially lowering their AUROC to achieve it. As such, the metric they used to argue the superiority of their method is irrelevant.

- The authors argue that Fine Web is human-written data and is a distinct training corpus. Fine Web is a subsample of Common Crawl, which makes a substantial part of other training model datasets, such as the GPT4 family (80%> as per the GPT4 technical report)

- I strongly suggest that the authors review the paper's wording and the background literature. The original citation for LLM pretraining being useful for downstream tasks (what you term as "understanding" on L34) was originally claimed by the ELMo paper [4], which is the correct citation for such claims. Similarly, the correct canonical citation for LLM texts being undetectable to humans is [5]. Overall, please review the wording of the introduction to ensure that the claims are factual and consistent with prior scientific literature, and that the coverage of existing literature is comprehensive and consistent with the claims.

- Similarly, the code supporting the paper should be shared through an anonymized repository, such as https://anonymous.4open.science/ or an anonymous GitHub repository with PPI removed.

[1] Huang, D., Nguyen, T., Liausvia, F., & Wang, Z. (2025). RAP: A Metric for Balancing Repetition and Performance in Open-Source Large Language Models. North American Chapter of the Association for Computational Linguistics.

[2] Dugan, L., Hwang, A., Trhlik, F., Ludan, J.M., Zhu, A., Xu, H., Ippolito, D., & Callison-Burch, C. (2024). RAID: A Shared Benchmark for Robust Evaluation of Machine-Generated Text Detectors. ArXiv, abs/2405.07940.

[3] Carlini, N., Chien, S., Nasr, M., Song, S., Terzis, A., & Tramèr, F. (2021). Membership Inference Attacks From First Principles. 2022 IEEE Symposium on Security and Privacy (SP), 1897-1914.

[4] Peters, M.E., Neumann, M., Iyyer, M., Gardner, M., Clark, C., Lee, K., & Zettlemoyer, L. (2018). Deep Contextualized Word Representations. NACL 2018

[5] Ippolito, D., Duckworth, D., Callison-Burch, C., & Eck, D. (2019). Automatic Detection of Generated Text is Easiest when Humans are Fooled. Annual Meeting of the Association for Computational Linguistics.

**Questions:**

- Drastically different sampling temperatures are used in the generation of ESL dataset and Ghostbuster replication with GPT-4o mini and Deepseek. Why?

---

> ### Author Response · Authors · 2025-11-19
>
> _"The authors do not seem to be aware, or at least mention, the existence of a generation control parameter called "repeat penalty", applied by default to models upon generation. This parameter penalizes token repetition to prevent model degeneration and can be easily adjusted to avoid detection by the authors' method. Without evaluating their method's robustness against this metric, the detection method's relevance or performance cannot be claimed, especially as sampling methods that automatically optimize repetition penalties are emerging [1]."_
>
> **Response:**
>
> Respectfully, we would like to clarify that repetition penalty is a generation-time sampling control that modifies what tokens get selected during inference. Telescope Perplexity measures the reference model's internal learned probability distribution P(s_i|s_1:i) encoded in its weights from training, which cannot be changed by changing target model generation parameters. In addition this parameter rarely ever actually affects the final output of the target model because modern models almost never actually sample repeated tokens from their probability distributions, so the repetition penalty almost never affects the target model’s generation. We will include a simple experiment to demonstrate this in the final paper.
>
> By default the value is the exact same as it is in training, so it doesn’t really have an effect on our hypothesis that what we are measuring is an instilled heuristic that both the reference and target models learn.
>
> _"The authors only evaluated their method's resilience against a single evasion attack, which is insufficient. Evasion attacks are the most problematic current issue in the context of LLM detection, as exemplified by the existing literature on LLM detectors benchmarking, notably the RAID paper [2]. As such, the performance of their method in a real-world setting cannot be evaluated."_
>
> **Response:**
>
> We evaluated multiple attack categories:
>  - 9 perturbation types from Ghostbusters (Section 4.1.3, Tables 16-17)
> - Custom AI humanizer (Section 9.16, Tables 9-11)
> - Adversarial prompting (Section 9.15)
> - Paraphrasing, word swaps, reordering (Figures 16-18)
>
> We agree comprehensive RAID-style benchmarking would strengthen claims, but RAID in particular uses older language models that are not close to representative of modern use cases.
>
> _"The authors do not seem to be aware of the foundational results in LLM detection, demonstrating that AUROC is not an informative metric with LLM detection [3]. In fact, due to the lack of access to true labels for LLM detector operators, the relevant metric is TPR at a fixed low FPR, ... As such, the metric they used to argue the superiority of their method is irrelevant."_
>
> **Response:**
>
> Only Binoculars argues that TPR @ FPR is a valuable metric. We include an entire section (9.13) to explain why we don’t use it  and how it fails to produce statistically significant results due to extremely large confidence intervals. See Tufts et al. (2025) and section 9.13 in the appendices for further arguments.
>
> _"The authors argue that Fine Web is human-written data and is a distinct training corpus. Fine Web is a subsample of Common Crawl, which makes a substantial part of other training model datasets, such as the GPT4 family (80%> as per the GPT4 technical report)"_
>
> **Response:**
>
> We use Fine Web for a very narrow purpose, it is human written data that, as you say, has been used in many models training. We only use it in 3.3 to show that a model can separate its own full training corpus into human and synthetic data. To reiterate this is the only use it has in the paper so its overlapping with the Common Crawl is irrelevant.
>
>
> _"I strongly suggest that the authors review the paper's wording and the background literature. ... please review the wording of the introduction to ensure that the claims are factual and consistent with prior scientific literature, and that the coverage of existing literature is comprehensive and consistent with the claims."_
>
> **Response:**
>
> We are more than happy to add these citations and adjust the introduction for increased clarity.
>
> _"Similarly, the code supporting the paper should be shared through an anonymized repository, such as https://anonymous.4open.science/ or an anonymous GitHub repository with PPI removed."_
>
> **Response:**
>
> We have provided all code needed in the supplementary materials as a zip file, and we are more than happy to walk you through the process of setting it up on your machine. We will additionally include an anonymous GitHub repository for future submissions.
>
> Question:
> _"Drastically different sampling temperatures are used in the generation of ESL dataset and Ghostbuster replication with GPT-4o mini and Deepseek. Why?"_
>
> **Response:**
>
> These are the default parameters for each model. Given both were selected as a result of being easily accessible via a free chat UI, we use the same parameters.

---

> > ### Comment · Reviewer_EUSN · 2025-11-22
> > **Comments on the rebuttal.**
> >
> > R1: LLM text detection task assumes access only to the generated texts that are potentially LLM generated, without access to the underlying model. Could the authors please comment on how their method is expected to be able to differentiate the repetition learned by the model compared to the one imparted by training (assuming it exists) the repetition penalty in this setting?
> >
> > Additionally, given that the learning of token repetition avoidance has not previously been reported in LLM training literature, could the authors please elaborate as to the expected mechanism for such repetition avoidance learning?
> >
> > R2:
> > - Section 4.1.3 does not contain sufficient description of the attack methods used and does not link to the Tables 16-17, although I acknowledge the inclusion of several variants of the paraphrasing attack.
> > - Custom AI humanizer in 9.1.6 methodology is not sufficiently specified. This is a particularly critical problem given that the reformulation methodology is unorthodox, using a BERT-based model rather than a decoder-based model or a full transformer architecture, as is common for such tasks
> > - Section 9.15 is not linked from the main paper body. Additionally, it is not clear as what the expected result of the adversarial prompt is, given that prior literature focused on prompts performing few-shot learning of a style matching a human one, or deep initialization, neither of which is the case here.
> > - Same as for Tables 16-17
> >
> > Regrading the RAID usage of older models, the benchmark sweep across attacks, text types, sampling parameters is transferable to newer models. However, I do not understand this criticism, given that a detector is expected to be model-agnostic and perform as well for older models as for the newer ones, allowing for a direct comparison with prior detector results.
> >
> > R3: The usage of TPR at fixed FPR is required by the threat model of the LLM detection. A defender does not know in advance the fraction of the LLM text in the dataset or their detector's performance, so they can only set a fixed low FPR. Hence AUROC or F1 metrics are irrelevant in this context, which is consistent with most standard LLM detection benchmarks (eg RAID).
> >
> > R4: This is not clear from the 3.3 section, whose title suggest a focus on generalization. It is now even less clear what the authors attempt to achieve, given that generation from a single model (SmolLM) has not yet been shown to leverage only one type of datasets at a time.
> >
> > R6: Noted for the code, however open repository availability of artifacts is consistent with best practices to allow for later artifacts access and re-use.
> >
> > R7: The temperature has been repeatedly reported as a critical parameter for the detectability benchmarking (cf [5] of the original review), and using temperatures provided by default chat interfaces is not a sufficient reason, especially given that a temperature range is benchmarked elsewhere in the paper.

---

> > > ### Author Response · Authors · 2025-12-02
> > >
> > > > "R1: LLM text detection task assumes access only to the generated texts that are potentially LLM generated, without access to the underlying model.
> > > > Additionally, given that the learning of token repetition avoidance has not previously been reported in LLM training literature, could the authors please elaborate as to the expected mechanism for such repetition avoidance learning?"
> > >
> > > **Response:** Language model learning begins with short range dependencies like bigram models (See L. Choshen, G. Hacohen, D. Weinshall, and O. Abend. The grammar-learning trajectories of neural language models. CoRR, abs/2109.06096, 2021. URL https://arxiv.org/abs/2109.06096) this extreme similarity to bigram models coincides with the increase in Telescope Perplexity to its stable level during training. So then we make the well supported inference that bigram models tend not to repeat tokens (similarly for other short range n-gram models). And voila we've shown how a Language model is likely to have short range (bigram in particular) aversion to repetition.
> > >
> > > ---
> > >
> > > > "R2: Section 4.1.3 does not contain sufficient description of the attack methods used and does not link to the Tables 16-17, although I acknowledge the inclusion of several variants of the paraphrasing attack."
> > >
> > > **Response:** These attack methods are all detailed in the Ghostbusters paper, since we are using their dataset and attack suite. If you would like, we can also add a section in the paper to reiterate exactly how the attacks were generated.
> > >
> > > > "Custom AI humanizer in 9.1.6 methodology is not sufficiently specified. This is a particularly critical problem given that the reformulation methodology is unorthodox ..."
> > >
> > > **Response:** Could you please point us towards citations for these common decoder based methods? Our AI humanizer uses randomly masked words rather than using the original text as context. We do this to maximise the similarity between AI text and human generated text while not fully reconstructing the original text.
> > >
> > > > "Section 9.15 is not linked from the main paper body.... deep initialization, neither of which is the case here."
> > >
> > > **Response:** We will include a link from section 9.15 to the main paper. We do not include a few shot adversarial prompts but we do include a one-shot version with our ESL rephrasing benchmark. We are not aware of any existing reputable datasets composed of deep initialized texts across domains. If the reviewer is aware of such papers we are more than happy to run our method on them.
> > >
> > > > "Same as for Tables 16-17
> > > > Regrading the RAID usage of older models, the benchmark sweep across attacks, text types, sampling parameters is transferable to newer models ... "
> > >
> > > **Response:** Detectors are never completely target model agnostic and oftentime heavily depend on the strength of the target model. Demonstrating performance on older target models is not indicative at all of performance on stronger target models trained using modern LLM training techniques. We would very much prefer to keep our detection results solely on newer target models to be as useful as possible.
> > >
> > > ---

---

> > > > ### Author Response · Authors · 2025-12-02
> > > >
> > > > > "R3: The usage of TPR at fixed FPR is required  ... Hence AUROC or F1 metrics are irrelevant in this context, which is consistent with most standard LLM detection benchmarks (eg RAID)."
> > > >
> > > > **Response:** In order to get reasonable confidence intervals, RAID uses TPR at 5% FPR which isn't ideal since a 5% FPR model would be undeployable in the real world, and therefore that too is insufficient. We maintain that in order to get reasonable confidence intervals while still measuring ability to distinguish between LLM and human written text that AUROC and F1Score are still viable and in many cases are the best metric. And while we acknowledge that AUROC isn't perfect, we believe that it is the best that we can do to benchmark LLM detection performance given the tradeoff between unreasonable confidence intervals at lower FPR values. While the AUROC and F1Score is less interpretable it is still worthwhile to compare between two detectors especially since we can't effectively measure the TPR at the super low FPR needed for deployment.
> > > >
> > > > ---
> > > >
> > > > > "R4: This is not clear from the 3.3 section, whose title suggest a focus on generalization. It is now even less clear what the authors attempt to achieve ..."
> > > >
> > > > **Response:** Perhaps we should change the section name, this section is meant to argue for the generality of a 'vestigial heuristic' not for our specific methods generality across domains like 5.3. We use SmolLM's training data separability to show "that we are not measuring the text's similarity to the training data, but a property learned in training!" This is evidence that method performance is a property of the model, combined with the techniques similar performance across reference models "suggest that the persistent local biases probed by Telescope Perplexity are likely a relatively general characteristic associated with current deep learning approaches to language modeling."
> > > >
> > > > ---
> > > >
> > > > > "R6: Noted for the code, however open repository availability of artifacts is consistent with best practices to allow for later artifacts access and re-use."
> > > >
> > > > **Response:** You've convinced the authors of this, further submission will use this process.
> > > >
> > > > ---
> > > >
> > > > > "R7: The temperature has been repeatedly reported as a critical parameter for the detectability benchmarking (cf [5] of the original review) ..."
> > > >
> > > > ---
> > > >
> > > > > "I would like to thank the authors for their rebuttal.
> > > > > Choice of benchmark and metric ...
> > > > > I am still not convinced if AUC scores of up to 0.999 for some datasets are representative of a realistic setup in practice."
> > > >
> > > > **Response:** Do you have a specific concern with any of the datasets or experiment setup? Good performance is not necessarily indicative of an issue in the data setup. Attacking our results on the account that our method performs too well is unhelpful and not logically sound.
> > > >
> > > > ---
> > > >
> > > > > "Unclear conceptual link between aversion to repeat tokens and usefulness for detection …
> > > > > The link between aversion to repeat tokens and detection of LLM-generated text remains unclear to me."
> > > >
> > > > **Response:** The simple version is that this vestigial heuristic is triggered more strongly by LLM text than human written text.
> > > >
> > > > ---
> > > >
> > > > > "Insufficient details for the "AI humanizer" method
> > > > > Given your response, it remains unclear to me whether the details of the "AI humanizer" method will ever be disclosed, even post-publication. ... revealing such vulnerabilities typically strengthens, rather than weakens, the ecosystem by enabling more robust future detectors.
> > > >
> > > > **Response:** This is a pretty compelling argument. Our original intent was not to release any "AI humanizer" code post publication. While we still have reservations about potentially contributing to the degradation of existing systems we will include the code and an in-depth appendix section discussing the "AI humanizer" design in the final paper.
> > > >
> > > > > "Additionally, the paper does not discuss robustness to intentional fine-tuning
> > > >
> > > > **Response:** Fine tuning for robustness to a detection method is something that no other work in the field of LLM detection does because it is far too expensive to consider for strong open source target language models. Not to even mention the important close source targets. Attempting to track the performance of LLM detection techniques when finetuned using different techniques across different target models represents its own major contribution to the field of LLM text detection.
> > > >
> > > > ---
> > > >
> > > > > "Could you share any insights into what drives these sharp inflection points…
> > > > > What drives these inflection points remains unclear to me."
> > > >
> > > > **Response:** It may be useful to know that these inflection points coincided with a large jump in the number of samples in each corpus.

---

### Comment · Area_Chair_k2Pq · 2025-11-21

Dear Reviewers,

We kindly encourage you to review and respond to the authors’ rebuttals. Your timely feedback is important for ensuring a fair and thorough review process. Thank you for your contributions to ICLR 2026.

AC

---

### Meta-Review · Area_Chair_JRke · 2026-01-06

**Summary:**

This paper introduces Telescope Perplexity as a metric signature for highly effective zero-shot LLM detection. Basically, the idea is interesting, but I do suggest the authors to include more baselines and ablations in the experiment. Although i do not agree with some of the reviewers, but clearly the work needs more efforts to make it sound and complete.
Suggestions for furture improvement: Many baselines are missing, such as: DNA-GPT: Divergent N-Gram Analysis for Training-Free Detection of GPT-Generated Text, ICLR'24
DALD: Improving Logits-based Detector without Logits from Black-box LLMs. NeurIPS'24.
Radar: Robust ai-text detection via adversarial learning. NeurIPS'23.
Detective: Detecting ai-generated text via multi-level contrastive learning. NeurIPS'24.
Human Texts Are Outliers: Detecting LLM-generated Texts via Out-of-distribution Detection. NeurIPS'25.
Additional suggestions, in the rebuttal with openreview, formulas and math terms can be used with latex style (markdown), For example, M(s_i|s_{1:i}), is better to be with $M(s_i|s_{1:i})$.

**Reviewer Concerns:**

1. AUROC vs TPR at low FPR threat model
It was raised by EUSN (explicitly argues AUROC is not informative; low-FPR is required), also
hSV2 (still not convinced AUC ≈ 0.999 is realistic; asks why mean AUROC)
It is highly suggested to report both AUROC and TPR at low FPR

2. Related work omissions. This is raised by three reviewers,
I also list some in the summary.

3. Robustness to intentional fine-tuning to evade detection， raised by hSV2 and EUSN
The authors did not well-addressed it.

**Reviewer Scores:**

all reviewers  are strong negative.

---

### Decision · Program_Chairs · 2026-01-26

Reject